# Break-induced replication is enhanced by a phospho-activated RPA-binding module in Pol32

David Jones [1,3], Rowin Appanah [1,2,3], Luke A. Yates[1], Antony W. Oliver [1] ✉ & Ulrich Rass [1] ✉

Break-induced replication (BIR) facilitates single-ended DNA double-strand break (DSB) repair. Upon homologous recombination-mediated strand invasion into a homologous repair template, BIR is catalysed by a minimal replisome comprising PCNA, DNA polymerase δ (Pol δ), and the Pif1 helicase. Here, we identify an interaction between Pol δ and single-stranded DNA (ssDNA)-binding protein RPA mediated by an RPA-binding module (RBM) within Pol δ subunit Pol32 and RPA subunit Rfa1. Pol32 RBM phosphorylation at Thr256 and Thr257 increases its affinity for Rfa1, while corresponding phospho-mimetic amino-acid substitutions promote BIR efficiency in vivo. This suggests that Pol32 functions as a rheostat whose phosphorylation enhances Pol δ's affinity for RPA-bound BIR intermediates, thereby boosting BIR efficiency. Modelling indicates that Pol32 phospho-RBM-Rfa1 interactions mirror the binding mode of RBMs in Pif1 and the FANCM helicase and BIR antagonist Mph1. This implies a key role for RPA in the dynamic orchestration of the enzymes mediating BIR.

Homologous recombination mediates error-free DNA double-strand break (DSB) repair by harnessing a template with DNA sequence homology across the break site to fill the gap by repair synthesis. At single-ended DSBs, which can be generated by DNA replication fork breakage or telomere erosion, a similar repair process known as break-induced replication (BIR) occurs[1–4]. First, DNA end-resection exposes a 3'-single-stranded DNA (ssDNA) overhang at the break site. Rad51 then forms a nucleoprotein filament on the ssDNA and mediates strand invasion at a repair template with sequence homology. This displaces one of the DNA strands at the repair template to form a displacement loop (D-loop). The invading strand serves as a primer for DNA synthesis, which is mediated by a BIR replisome consisting of DNA polymerase δ (Pol δ), the processivity clamp PCNA, and the helicase Pif1. A double-edged sword, BIR can rescue chromosome replication after replication fork breakage but is prone to mutations, ectopic recombination and chromosome aberrations[5–8]. This suggests that cells must exercise careful control over BIR, and there is evidence that cells limit the extent of BIR-dependent DNA synthesis during the recombinational repair of Cas9 nickase-induced replication fork collapse. This can be achieved through active D-loop processing by structure-specific nucleases and the arrival of a converging DNA replication fork[2,9]. Alternatively, cells may suppress BIR to favour the generation of a two-ended DSB and subsequent repair by error-free homologous recombination[10–12].

In contrast to canonical replication, BIR exposes extensive amounts of ssDNA, both by strand displacement at the D-loop and in the form of the spooled-out nascent strand trailing the progressing D-loop. The nascent strand remains temporarily single-stranded before serving as a template for lagging-strand synthesis. In the cell, ssDNA is quickly covered by the trimeric ssDNA-binding protein RPA consisting of Rfa1/RPA70 (yeast/human), Rfa2/RPA32, and Rfa3/RPA14. The Rfa1/RPA70 N-terminal domain (NTD) harbours a basic-hydrophobic groove that serves as a hub for various client proteins with small acidic-hydrophobic RPA-binding modules (RBMs). RBMs have been identified in several DNA damage response factors and shown to facilitate DNA metabolic processes during DNA replication,

[1]Genome Damage and Stability Centre, School of Life Sciences, University of Sussex, Falmer, Brighton, UK. [2]Present address: Université de Lyon, ENS de Lyon, Université Claude Bernard, CNRS UMR5239, Laboratoire de Biologie et Modélisation de la Cellule, Lyon, France. [3]These authors contributed equally: David Jones, Rowin Appanah. ✉e-mail: Antony.Oliver@sussex.ac.uk; U.W.Rass@sussex.ac.uk

repair, and recombination[13–21]. Recently, an RBM was identified within helicase Pif1, which is required for D-loop progression during BIR[20,22]. Whether there is direct crosstalk between PCNA-Pol δ and RPA during BIR is not known.

*Saccharomyces cerevisiae* Pol δ consists of the catalytic subunit, Pol3, and the accessory subunits Pol31 and Pol32. Pol32 was found to be dispensable for cell viability in budding yeast, but is required for efficient BIR[23]. Thus, *pol32Δ* cells initiate DNA synthesis but cannot sustain BIR over long distances, so Pol32 can be described as a processivity factor for Pol δ in the context of a D-loop[24]. A cryo-EM structure of the Pol δ holoenzyme shows that the Pol32 NTD spanning amino acid residues (aa) 1-115 folds into a globular domain that closely associates with Pol31, which in turn binds the catalytic subunit Pol3[25]. Loss of Pol32 compromises Pol δ function and results in replication defects and cellular replication stress sensitivity, but these phenotypes can be reversed by a Pol31 gain-of-function mutation that enhances the stability of the Pol3-Pol31 complex[26,27]. Conversely, a Pol31 polymorphism that destabilises the interaction of Pol31 with Pol3 renders *POL32* essential in budding yeast[28]. Importantly, the Pol δ-stabilising Pol31 mutation does not suppress the profound BIR defect caused by loss of Pol32 (ref. [26]), highlighting BIR-specific functions uniquely associated with Pol32. Consistently, BIR efficiency is negatively affected by the disruption of PCNA and DNA polymerase α (Pol α) interaction sites within the Pol32 C-terminal domain (CTD) spanning aa 116–350 (ref. [23]). Yet, how Pol32 orchestrates BIR efficiency remains to be fully investigated.

Here, we identify an RBM within the Pol32 CTD. Its phosphorylation enhances Pol32 RBM-RPA interactions, providing the first example of phospho-activated binding to the Rfa1 basic-hydrophobic groove. Corresponding phospho-mimetic amino-acid substitutions boost BIR efficiency in cells. Thus, Pol32 is not a mere enabler of BIR, but serves as an acceptor of post-translational modifications that cells can utilise to dynamically regulate BIR efficiency through modulating the affinity of Pol δ for RPA.

## Results

### A Pol32 C-terminal element promotes BIR
Pol32 is 350 aa in length. The Pol32 CTD is not visible in deposited EM structures[25], adopts an elongated shape[29], and is predicted by AlphaFold[30–32] to be largely disordered (Fig. 1A). A C-terminus-proximal PCNA-interacting protein-box (PIP-box)[33] and a DNA polymerase α interaction motif (DPIM)[34,35] spanning aa 270-309 have previously been identified within the Pol32 CTD. By complementing *pol32Δ* cells with a series of plasmid-borne Pol32 mutants, it has been shown that BIR efficiency is reduced by disruption of the Pol32 PIP-box and large internal deletions that affect the DPIM while leaving the PIP-box intact[23]. To better understand how Pol32 promotes BIR, we truncated Pol32 from the C-terminus in a stepwise manner (Fig. 1B). The truncations were introduced at the endogenous *POL32* locus, and the truncated proteins were expressed at levels comparable to the wild-type protein (Fig. 1C). The impact of these mutations on BIR was determined using a reporter strain[23] for BIR efficiency at a DSB on chromosome V with the only available repair template located at an ectopic site on chromosome XI (Fig. 1D). The DSB is effectively introduced across the cell population upon galactose-induced expression of the HO endonuclease[23] (Supplementary Fig. 1A). BIR-mediated DSB repair can be genetically tracked as it restores a *CAN1* marker at the cut site whilst leading to the loss of a strategically placed *HPH* marker (Fig. 1D and Supplementary Fig. 1B). Confirming earlier work[23,33], the PIP-box truncation (Pol32_1-333) disrupted Pol32-PCNA interactions in yeast two-hybrid assays (Supplementary Fig. 1C) and resulted in a reduction (-5-fold) in BIR efficiency compared to cells harbouring wild-type *POL32* (Fig. 1E). Further truncation of endogenous Pol32 to Gln269, thereby removing the DPIM and disrupting Pol α interactions (Supplementary Fig. 1D), caused a further -2-fold reduction in BIR

efficiency (Fig. 1E). These observations are consistent with the notion that interactions with both PCNA and Pol α are important for the role of Pol32 in BIR[23]. However, removal of a further 20 aa (Pol32_1-249) was required to reduce BIR efficiency to the level seen in *pol32Δ* cells (Fig. 1B, E). Next, we analysed DSB repair in an independent tester strain which is disomic for chromosome III (with one copy harbouring an HO-cut site and the other serving as a repair template)[36]. In this system, DSB repair is highly efficient, and cells overwhelmingly use BIR with a small contribution of DSB repair by gene conversion. In contrast, *pol32Δ* cells show reduced BIR efficiency, an increase in chromosome loss events, and the formation of half-crossovers indicative of failed DSB repair and aberrant processing of BIR intermediates, respectively[36]. We observed similar levels of Pol32-independent BIR, chromosome loss, and half-crossovers in cells harbouring a *POL32* deletion or a truncation of Pol32 at residue 249 (Supplementary Fig. 1E, F). Thus, Pol32_1-249 results in a Pol32-null phenotype for ectopic and allelic interhomolog BIR. In contrast, expression of Pol32_1-249 does not cause cellular replication stress or cold sensitivity, which are the phenotypes that signify replication defects associated with the complete loss of Pol32 and Pol3-Pol31 instability (Fig. 1F).

Accordingly, Pol32_1-249 is a separation-of-function mutant, showing that the C-terminal -100 aa of Pol32 are required for BIR in their entirety, whilst being dispensable for canonical replication functions, which depend upon Pol δ stability. Moreover, this region contains a functional element impacting BIR, approximately spanning aa 250–269 within the Pol32 CTD.

### Pol32 contains an RBM and mediates Pol δ-RPA interactions
AlphaFold predicts a small α-helical feature spanning Pol32 aa 244–261 (marked as Helix in Fig. 1A) that overlaps with the aforementioned functional element relevant for BIR efficiency. Its sequence (DKNNDDLEDLLETTAEDS) is rich in acidic and hydrophobic residues, which are segregated to opposite faces of the α-helix. Such amphipathic helices are often involved in protein-protein interactions and can be found in RBMs of established RPA client proteins[13–19]. Typically, acidic-hydrophobic RBMs engage in interactions with a basic-hydrophobic groove found within the N-terminal OB domain (also known as DBD-F) of Rfa1/RPA70; this has been demonstrated in yeast by structural analyses of the Rfa1 NTD in complex with an RBM found in Ddc2 (the integral partner of the cell cycle checkpoint kinase Mec1)[16,37]. To address the possibility that Pol32 may harbour a hitherto unidentified RBM, we used AlphaFold[38] to generate structural models exploring a potential interaction between Pol32 and Rfa1. Here, a C-terminal section of full-length Pol32 was consistently predicted to associate with the Rfa1 NTD, as supported by analysis with AlphaBridge[39] (Supplementary Fig. 2A). Subsequent truncation of the input sequence to include just aa 237–267 of Pol32 produced highly consistent models (Supplementary Fig. 2B). These show that aa 241–263 of Pol32 adopt an amphipathic α-helical conformation that occupies the groove of Rfa1 (Fig. 2A). However, the trajectory and position of the putative Pol32 RBM differs from Ddc2; it binds with opposite polarity and does not have a kink in its path when bound to Rfa1 (Fig. 2A). Three hydrophobic residues, Leu250, Leu253 and Leu254 (coloured teal, green and orange, respectively) serve to anchor Pol32 into the hydrophobic section of the Rfa1 groove (lined by residues Leu47, Leu60, Ala88, Val97, and Leu99, see Fig. 2A and Supplementary Fig. 2C). By structural comparison, the side chains of these three residues have equivalency to Ile17, Leu18 and Leu21 of Ddc2 (Fig. 2A and Supplementary Fig. 2D; PDB: 8B4J). Reversal of the amino acid sequence of Ddc2 (reading it from the C- to the N-terminus) allows a simpler comparison with Pol32 and suggests that hydrophobic residues at these positions are conserved, as part of an φ-D/E-x-φ-φ motif (where φ denotes a hydrophobic and x any amino acid residue). We note that a semi-conserved region of amino acids follows (or precedes) this motif, consisting of either negatively charged amino acids, and/or

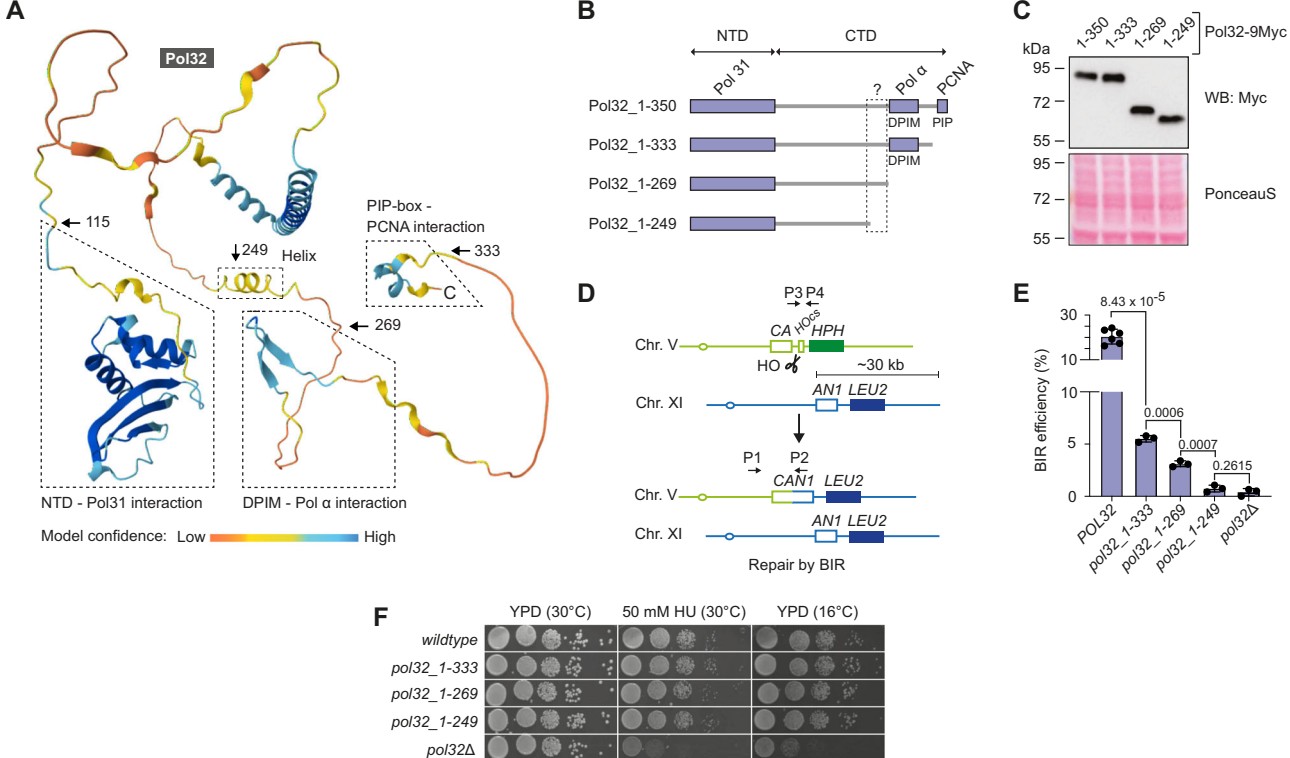

**Fig. 1 | Pol32 contains a BIR-promoting feature spanning amino acids 250–269. A** AlphaFold structural model of Pol32 with functional domains indicated. Arrows demarcate the boundaries of Pol32 C-terminal truncations used in this study. The model is shown in cartoon representation and coloured according to pLDDT (predicted Local Distance Difference Test) score, using a continuous rainbow colour spectrum from red (low confidence) to blue (high confidence). **B** Schematic drawing of full-length and truncated versions of Pol32 summarising the main features. The Pol32 NTD (N-terminal domain) binds Pol31; the DPIM (DNA polymerase α interaction motif) and PIP-box (PIP), embedded in the CTD (C-terminal domain), bind Pol α and PCNA, respectively. The question mark indicates an uncharacterised region identified as functionally significant for BIR in this study. **C** Western blot analysis of the indicated 9Myc-tagged Pol32 constructs expressed from the endogenous *POL32* locus. Representative of *n* = 2 independent experiments. A Ponceau S staining of the western blot membrane serves as a loading control. The positions of size markers (kDa) are indicated. **D** BIR efficiency reporter system with ectopic repair template. An *HPH*-marked HO endonuclease cut site (*HOcs*) replaces the 3′-end of a *CAN1* gene integrated on chromosome V (*CA*). The *LEU2*-marked *AN1* repair template on chromosome XI shares sequence homology with *CA*. Upon galactose-induced HO-mediated DSB formation, BIR restores *CAN1* and must proceed to copy the 30 kb telomere-proximal region of chromosome XI onto chromosome V to produce viable progeny. *HPH* is lost in the process, rendering cells sensitive to hygromycin. PCR with primers P1/P2 and P3/P4 monitors the initiation of BIR DNA synthesis and DSB induction, respectively. **E** BIR efficiency measurements of the indicated strains (mean ± SD; *n* = 3 independent biological repeats, except *n* = 6 for *POL32*). Statistical significance of differences in BIR efficiency between strains was assessed by a two-tailed Welch's *t* test with *p*-values indicated. **F** Dilution spot assays with the indicated strains to assess their replication stress sensitivity on medium containing hydroxyurea (HU) and cold sensitivity. Source data are provided as a Source Data file.

several serine and threonine residues. Within this region, the side chain of Rfa1 Arg44 is poised to interact with Pol32 Asp260, whereas for Ddc2 it sits in proximity to the backbone carbonyl of Ser11 (see Supplementary Fig. 2C, D); as Ser11 of Ddc2 is known to be phosphorylated, this immediately suggests that additional negative charge can be added to this patch via post-translational modification.

Next, we sought to determine whether Pol32 can mediate RPA interactions in the context of the Pol δ holoenzyme. Affinity capture of endogenously TAP-tagged Pol31 co-precipitated the remaining subunits of Pol δ, Pol3 and Pol32, from whole cell extracts (Supplementary Fig. 3A). We found that RPA was enriched in Pol δ precipitates. However, an internal deletion of Pol32 aa 247–258 (hereafter referred to as ΔRBM) diminished Pol δ-RPA interactions (Fig. 2B). These data are therefore consistent with the acidic-hydrophobic motif within Pol32 functioning as a bona fide RBM and serve to confirm the identified Pol δ-RPA interaction.

To measure the affinity of the Pol32 RBM for RPA, we synthesised a fluorescein-labelled peptide derived from Pol32 aa 243–264 and tested binding to purified Rfa1 NTD (aa 1-132)[37] using fluorescence polarisation (FP). We determined a $K_D$ of 22.9 µM for the interaction (Fig. 2C). This is commensurate with affinities that have been

determined for other RBM-RPA interactions[14]. As expected, targeting key hydrophobic interactions indicated by the Pol32 RBM-Rfa1 model by introducing mutations Leu253Arg and Ala258Arg into the Pol32 RBM (hereafter referred to as RBM*) dramatically reduced the apparent binding affinity for the Pol32-derived peptide ($K_D$ not determined; see Fig. 2C).

Perhaps surprisingly, neither the ΔRBM nor the RBM* mutation in the context of full-length Pol32 had apparent effects on BIR efficiency in vivo (Fig. 2D). However, when the RBM was mutated together with the PIP-box (Pol32_1-333 RBM*), cells exhibited a *pol32Δ*-like BIR phenotype (Fig. 2D), while retaining replication competence (Supplementary Fig. 3B). Thus, the Pol32 PIP-box and RBM can function independently to promote BIR efficiency.

A PCR-based BIR assay[23] for the detection of new DNA synthesis initiated at the centromere-proximal side of the HO cut site (see Supplementary Fig. 1B) upon invasion of the repair template showed that Pol32 RBM* was associated with a slight but reproducible delay in the formation of BIR repair intermediates in the first 8 h after DSB induction. While the levels of BIR product formation mediated by Pol32 RBM* were eventually indistinguishable from wild-type, Pol32_1-333 RBM* was associated with a prolonged BIR DNA synthesis defect that

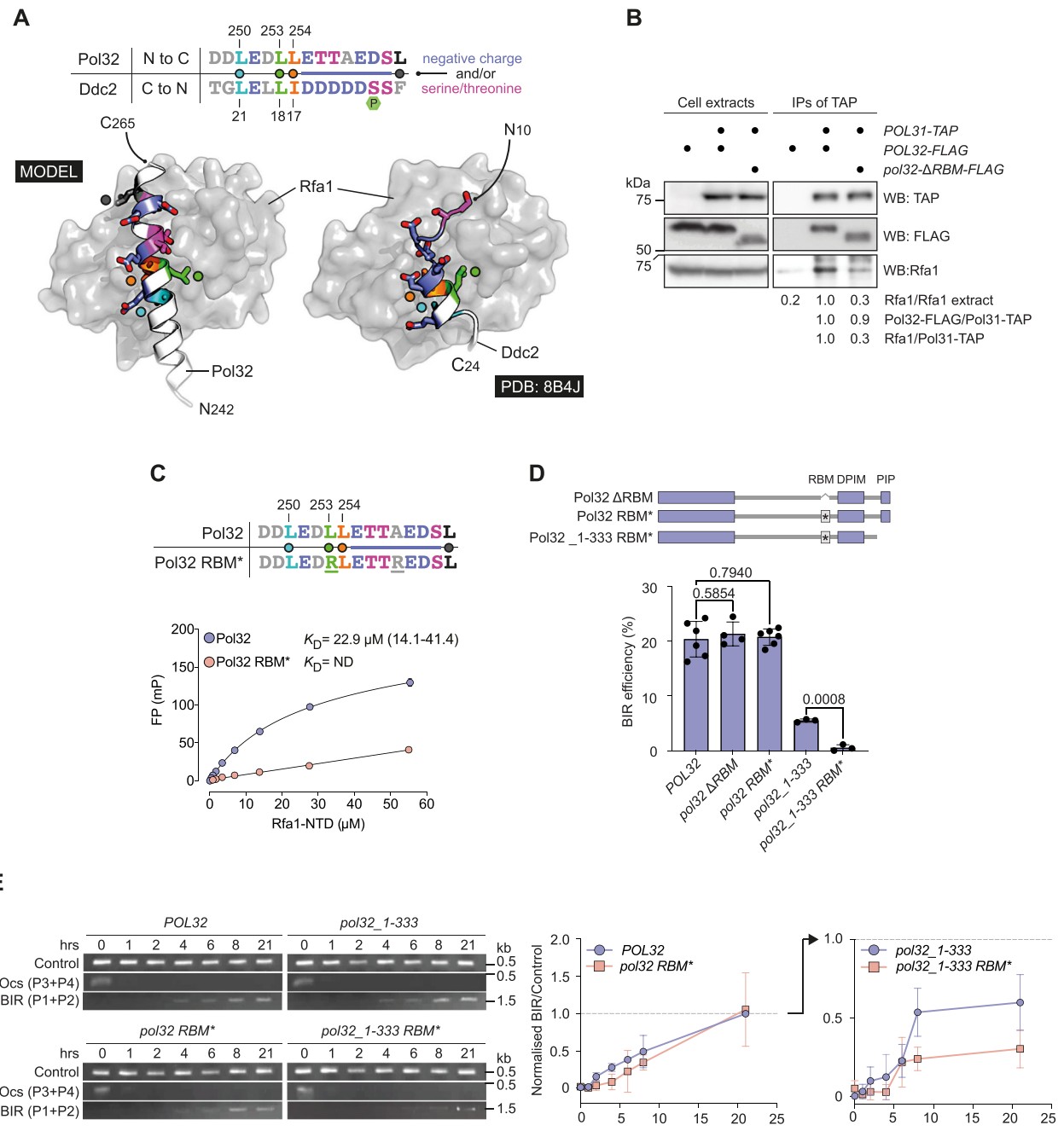

**Fig. 2 | An RBM in Pol32 mediates Pol δ-RPA interactions. A** Schematic molecular cartoons for the AlphaFold model of Pol32 RBM, and X-ray crystal structure of Ddc2 RBM (PDB: 8B4J), in complex with Rfa1 NTD. The side chains for selected amino acids are shown in stick representation, and colour-coded according to the inset amino acid sequence alignment. Rfa1 NTD is represented by the grey coloured molecular surface. A phosphorylation site within Ddc2 is marked with the letter P. **B** Western blot analysis of co-immunoprecipitation experiments carried out with extracts from strains harbouring the indicated Pol31 and Pol32 constructs, with quantification. Representative of $n = 2$ independent experiments. The positions of size markers (kDa) are indicated. **C** FP experiment demonstrating binding between the indicated fluorescein-labelled Pol32 peptides and Rfa1 NTD, and calculated dissociation constant ($K_D$) (mean ± SD, with uncertainty in the mean provided by

95% confidence intervals in brackets; $n = 3$ technical replicates). ND, no deter-mined. **D** Schematic representation of Pol32 RBM mutants and BIR efficiency measured for the indicated strains (mean ± SD; *POL32*, $n = 6$; *pol32 ΔRBM*, $n = 4$; *pol32 RBM\**, $n = 6$; *pol32_1-133*, $n = 3$; *pol32_1-133 RBM\**, $n = 3$ independent biological repeats). Statistical significance of differences in BIR efficiency between strains was assessed by a two-tailed Welch's $t$ test with $p$-values indicated. **E** BIR repair kinetics as determined by PCR assay. Representative example of agarose-gel resolved PCR products generated with primer pairs P1/P2 and P3/P4 as indicated in Fig. 1D (control PCR amplifies an unrelated locus) and quantification (mean ± SD; $n = 3$ independent biological repeats). The positions of size markers (kb) are indicated. Source data are provided as a Source Data file.

grew more pronounced between 8 and 20 h after DSB induction (Fig. 2E). Together, these data indicate that Pol32 RBM-RPA interactions promote BIR.

## The Pol32 RBM is phosphorylated by multiple protein kinases

The newly identified Pol32 RBM contains three potential phosphorylation sites at Thr256, Thr257, and Ser261. Phospho-proteomic experiments have shown that Pol32 can be phosphorylated at all three sites in vivo[40–44]. Whilst fully phosphorylated RBM peptides were observed in a subset of these studies[42–44], phosphorylation at Thr257, either alone or in combination with phosphorylation at Thr256, was more common. Notably, phosphorylation at Ser261 alone was not reported. Thus, the Pol32 RBM appears to be predominantly modified at both threonine residues. While the function and mechanism of Pol32 RBM phosphorylation remain to be determined, phospho-proteomic evidence suggests the involvement of multiple kinases in vivo. These include the acidophilic Dbf4-dependent kinase (DDK) Cdc7 (ref. 44), cyclin-dependent kinase (CDK) Cdc28 (ref. 40), and Polo-like kinase (PLK) Cdc5 (ref. 42). To test direct phosphorylation by these kinases, we purified a recombinant Thioredoxin-Pol32 fusion protein, incorporating Pol32 aa 248-264 (Trx-RBM), and subjected it to in vitro kinase reactions with TAP-tagged versions of Cdc7, Cdc28 (as well as Cdc28-associated cyclins Clb1, 3, and 5), and Cdc5 (as well its downstream kinase Cdc15) (Supplementary Fig. 4A). We also tested the acidophilic casein kinase CKII (Cka2-TAP) and GSK3 kinase ortholog Rim11-TAP, which have predicted[45] consensus phosphorylation target sites at Thr256 and Thr257, respectively. Phosphorylation of Trx-RBM was observed with affinity-captured Cdc7, Cdc5, and CKII, but not Cdc28 or Rim11. The levels of Trx-RBM phosphorylation mediated by Cdc7, Cdc5, and CKII were visibly diminished when a fusion protein with Pol32 T256A and T257A substitutions was used in the kinase assay (Supplementary Fig. 4B), indicating that at least part of the observed kinase activity was directed at the Thr256/257 twin phosphorylation site. Next, we mapped phosphorylation events using mass spectrometry (Fig. 3A), which confirmed the ability of affinity-captured Cdc5, CKII, and Cdc7 to phosphorylate Pol32 RBM residues Thr256, Thr257, and Ser261 in vitro (Fig. 3B and Supplementary Fig. 4C). Here, the greatest number of Pol32 RBM phospho-peptides were generated by Cdc7, which has previously been implicated[46] in the regulation of BIR. In addition, these data confirmed the DDK and PLK-dependent phosphorylation of the Pol32 RBM, which has been observed in vivo, and suggest that Cdc5, Cdc7, and possibly additional acidophilic kinases such as CKII may contribute to establishing its phosphorylation.

## Pol32 RBM phosphorylation boosts RPA affinity and BIR

To address the impact of Pol32 RBM phosphorylation on RPA interactions, we synthesised different versions of the fluorescein-labelled Pol32 RBM peptide and tested interaction with the Rfa1 NTD by FP as before. Here, the pThr257 peptide bound with a $K_D$ of 4.3 µM, a ~5-fold increase in affinity compared to the non-phosphorylated version of the Pol32 peptide. The bis-phosphorylated Pol32 pThr256 pThr257 peptide bound with a $K_D$ of 0.14 µM, an additional ~30-fold increase in affinity over the Pol32 peptide phosphorylated at Thr257 only (and a ~160-fold increase compared to the unphosphorylated peptide). Finally, additional phosphorylation at Ser261 did not have a marked effect on affinity with a Pol32 pThr256 pThr257 pSer261 peptide binding to Rfa1 NTD with a $K_D$ of 0.11 µM (Fig. 3C). Thus, Pol32 RBM bis-phosphorylation at the Thr256/257 twin site significantly strengthens Pol32-RPA interactions.

To rationalise enhanced RPA binding upon Pol32 RBM phosphorylation, we used the functionality of AlphaFold 3 to model the interaction of the phosphorylated RBM (pRBM) with Rfa1 (Fig. 3D and Supplementary Fig. 5). This suggests that phosphorylation of Thr257 breaks the extended alpha-helical conformation that was predicted for the non-phosphorylated Pol32 RBM (see Fig. 2A), positioning the

phosphate so that it sits in proximity to the side chains of Rfa1 Arg91 and -Lys95, and generating a kink in the modelled path of the bound peptide. We validated the importance of this modelled set of contacts by introducing Arg91Ala and Lys95Ala substitutions into Rfa1 NTD (Rfa1-MUT) and then testing interactions by FP. Here, the previously observed enhancement of binding driven by phosphorylation of Thr257 in the Pol32 RBM was neutralised (Supplementary Fig. 6A).

Within the short helical element immediately following pThr257 (aa 258-262, AEDSL), the side chain of Ala258 packs against the face of Leu262 and projects down into a hydrophobic pocket formed by Rfa1 Met49, -Leu47 and the side chain carbon atoms of Rfa1 Asn33 and -Lys58 (Fig. 3D and Supplementary Fig. 5B). Hydrophobic interactions with this so-called side-pocket have previously been observed for RPA client proteins engaged with the Rfa1 NTD hydrophobic groove and serve to enhance the overall interaction strength[14]. This suggests an induced-fit mechanism where Pol32 RBM phosphorylation at Thr257 facilitates a conformational change that allows higher-affinity Rfa1 NTD interactions. In line with the further enhanced affinity of the Thr256/257 bis-phosphorylated RBM for Rfa1 NTD measured in vitro, our modelling also suggests that the additional phosphorylation of Thr256 would be tolerated, picking up potential hydrogen bonds with the side chains of Rfa1 Arg35 and -Arg44. Modelling of the bis-phosphorylated Pol32 RBM showed that the phosphorylated threonines swap partners, presumably to eliminate torsional stress from the polypeptide backbone, but continue to interact with the same basic residue pairings of Arg91/Lys95 and Arg35/Arg44 (Fig. 3D and Supplementary Fig. 5C). In agreement with this hypothesis, the binding affinity of the pThr256/pThr257 peptide was reduced to that of the mono-phosphorylated peptide when titrations were carried out with Rfa1-MUT (Supplementary Fig. 6B). Our models indicate that the short helix that follows the phosphorylation sites would be slightly shifted in register (aa 259–263, EDSLM) with Leu262 still anchored in the hydrophobic side-pocket, but with Ala258 now displaced (Fig. 3D and Supplementary Fig. 5D).

Finally, whilst models of the fully phosphorylated Pol32 RBM bound to Rfa1 NTD were of poorer quality (lower pLDDT (predicted Local Distance Difference Test) score and consistency), they indicated that pSer261 may point towards the solvent and make no apparent interaction with Rfa1 NTD. This is consistent with our biophysical data, where phosphorylation of Ser261 makes a negligible contribution to the overall affinity (Fig. 3C and Supplementary Fig. 5E, F). We therefore conclude that it is phosphorylation at the two sequential threonines 256/257 of the Pol32 RBM that drives a high-affinity interaction with Rfa1 NTD.

Addressing the impact of enhanced Rfa1 NTD interactions mediated by Pol32 pRBM on BIR, we confirmed that phospho-mimetic amino changes Thr256Asp and Thr257Asp within the RBM increase the affinity for Rfa1 (Supplementary Fig. 6C). We then generated versions of the ectopic BIR reporter strain expressing Pol32 T256D T257D, which was present at concentrations comparable to wild-type Pol32 (Fig. 3E). Importantly, Pol32 T256D T257D boosted BIR efficiency as compared to the control strain (Fig. 3F). Like Pol32 ΔRBM and -RBM*, an endogenously expressed Pol32 T256A T257A phospho-null mutant was not associated with a BIR efficiency defect (Fig. 3F). However, as in the case of Pol32_1-333 RBM*, a strain expressing Pol32_1-333 T256A T257A showed a significant drop in BIR efficiency, suggesting that phosphorylation at Thr256/257 is required for BIR efficiency in the absence of the Pol32 PIP-box. Consistently, T256D/T257D continued to enhance BIR efficiency in the context of a Pol32 PIP-box truncation mutant (Fig. 3G). These data are in line with the notion that the phosphorylation of the Pol32 RBM in vivo[40–44] can be deployed by cells to modulate BIR efficiency.

## Pol32 pRBM, Pif1, and Mph1 similarly target Rfa1

Recently, an RBM has been identified within the accessory BIR helicase Pif1 and shown to be required for BIR efficiency[20]. Mph1, which

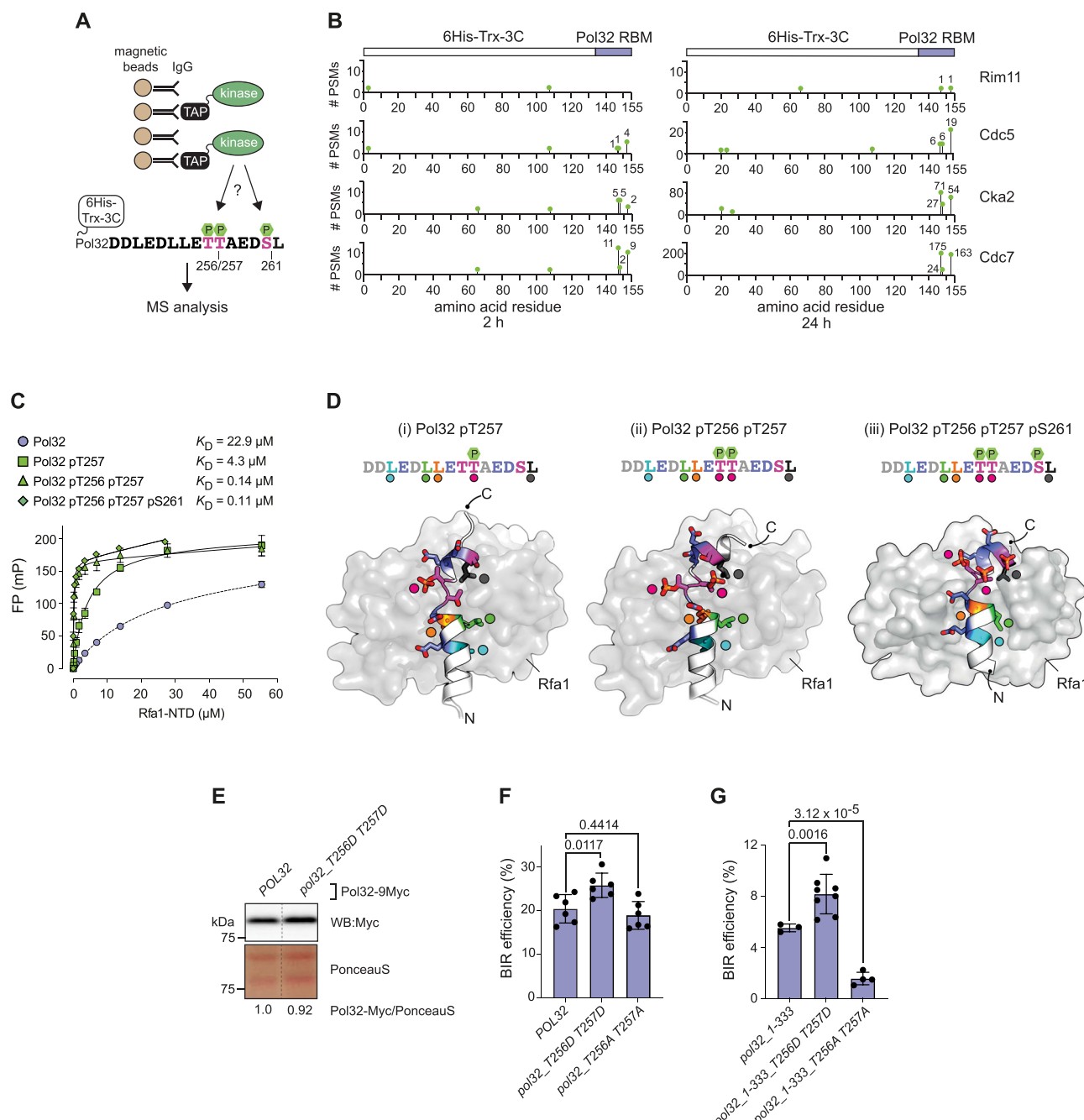

**Fig. 3 | Phosphorylation of Pol32 enhances RPA binding and BIR. A** Schematic of the in vitro kinase assay with a Trx-Pol32 RBM fusion protein followed by mass spectrometry (MS) analysis. **B** Lollipop plots showing phosphorylation at Pol32 RBM residues Thr256, Thr257, and Ser261 and within the thioredoxin tag as determined by MS ($n = 1$) after 2 and 24 h incubation with the indicated affinity-captured kinases. PSM, peptide-spectrum match. **C** Fluorescence polarisation (FP) experiment demonstrating binding between the indicated fluorescein-labelled Pol32 phospho-RBM peptides and Rfa1 NTD, and calculated dissociation constant ($K_D$) (mean ± SD, with uncertainty in the mean provided by 95% confidence intervals of 14.1–41.4 (Pol32), 3.01–6.24 (Pol32 pT257), 0.1–0.19 (Pol32 pT256 pT257), and 0.1–0.13 (Pol32 pT256 pT257 pS261; $n = 3$ technical replicates). The non-phosphorylated Pol32 peptide is included for reference. **D** Schematic molecular cartoons for the AlphaFold models of Pol32 phospho-RBMs in complex with Rfa1 NTD. The side chains for selected amino acids are shown in stick representation, and colour-coded according to the amino acid sequence alignment insets, where

the phosphorylation sites are indicated. Rfa1 NTD is represented by the grey coloured molecular surface. **E** Western blot analysis ($n = 1$) of the indicated 9Myc-tagged Pol32 constructs expressed from the endogenous *POL32* locus. A Ponceau S staining of the western blot membrane was used for quantification. The positions of size markers (kDa) are indicated. **F** BIR efficiency measurements for strains harbouring Pol32 variants with the indicated phospho-mimetic and phospho-null mutations (mean ± SD; $n = 6$ independent biological repeats). Statistical significance of differences in BIR efficiency between strains was assessed by a two-tailed Welch's $t$ test with $p$-values indicated. **G** BIR efficiency measurements for strains harbouring PIP-box truncation variants of Pol32 with the indicated phospho-mimetic and phospho-null mutations (mean ± SD; *pol32_1-133*, $n = 3$; *pol32_1-133 T256D T257D*, $n = 8$; *pol32_1-133 T256A T257A*, $n = 4$ independent biological repeats). Statistical significance of differences in BIR efficiency between strains was assessed by a two-tailed Welch's $t$ test with $p$-values indicated. Source data are provided as a Source Data file.

functions as an antagonist of BIR by dissociating extended D-loops[47–49], has also been shown to bind RPA[50,51], although the details of this interaction and its potential significance to BIR is not known. Inspection of the Mph1 C-terminus revealed an acidic-hydrophobic amino acid sequence (residues 981–993) with close resemblance to the Pol32 RBM. While the Pif1 RBM has been defined as spanning aa 60–74 (ref. 20), Trp75 within Pif1 aligns well with a similar hydrophobic residue, Phe993, at the end of the putative RBM sequence in Mph1 (Fig. 4A). Interestingly, both Pif1 and Mph1 RBMs contain pre-existing negative charges (Asp71 and Asp990, respectively) at the position of the Thr256/257 double phosphorylation site within Pol32, suggesting that both RBMs might behave more similarly to the phosphorylated rather than the non-phosphorylated Pol32 RBM. Indeed, AlphaFold modelling predicts that the Pif1 and Mph1 RBMs bind Rfa1 with the same polarity as Pol32, placing three hydrophobic residues consistent with the φ-D/E-x-φ-φ motif into the Rfa1 groove: Pif1 Leu66, -Leu68 and -Leu69; Mph1 Leu983, -Ile986 and -Leu987 (Fig. 4A and

Supplementary Fig. 7A, B). Pif1 Trp75 and Mph1 Phe993 bind directly into the hydrophobic side-pocket that is occupied by Leu262 in the phosphorylated forms of the Pol32 RBM (Supplementary Fig. 7C, D). Here, the additional phosphorylation of the Pif1 or Mph1 RBMs is not predicted to significantly impact the position of Pif1 Trp75 and Mph1 Phe993, which suggests that both proteins adopt the high-affinity Rfa1 binding confirmation independently of phosphorylation within their RBMs (Supplementary Fig. 7E, F). Consistently, we determined a $K_D$ of 0.32 μM for a fluorescein-labelled peptide derived from Pif1 aa 59–79 with the Rfa1 NTD by FP (Fig. 4B). Thus, Pif1 tightly binds RPA and with similar affinity to Pol32 RBM phosphorylated at Thr256/257. Phosphorylation of Pif1 at Ser70 and Ser72 flanking Asp71 had a moderate impact on Rfa1 NTD binding affinity; improving the $K_D$ to 0.15 μM (Supplementary Fig. 8A). Phospho-null (Ser70Ala/Ser72Ala) or phospho-mimetic (Ser70Asp/Ser72Asp) amino acid substitution in endogenous Pif1 had no significant effect on BIR efficiency in vivo (Supplementary Fig. 8B). However, consistent with a reported

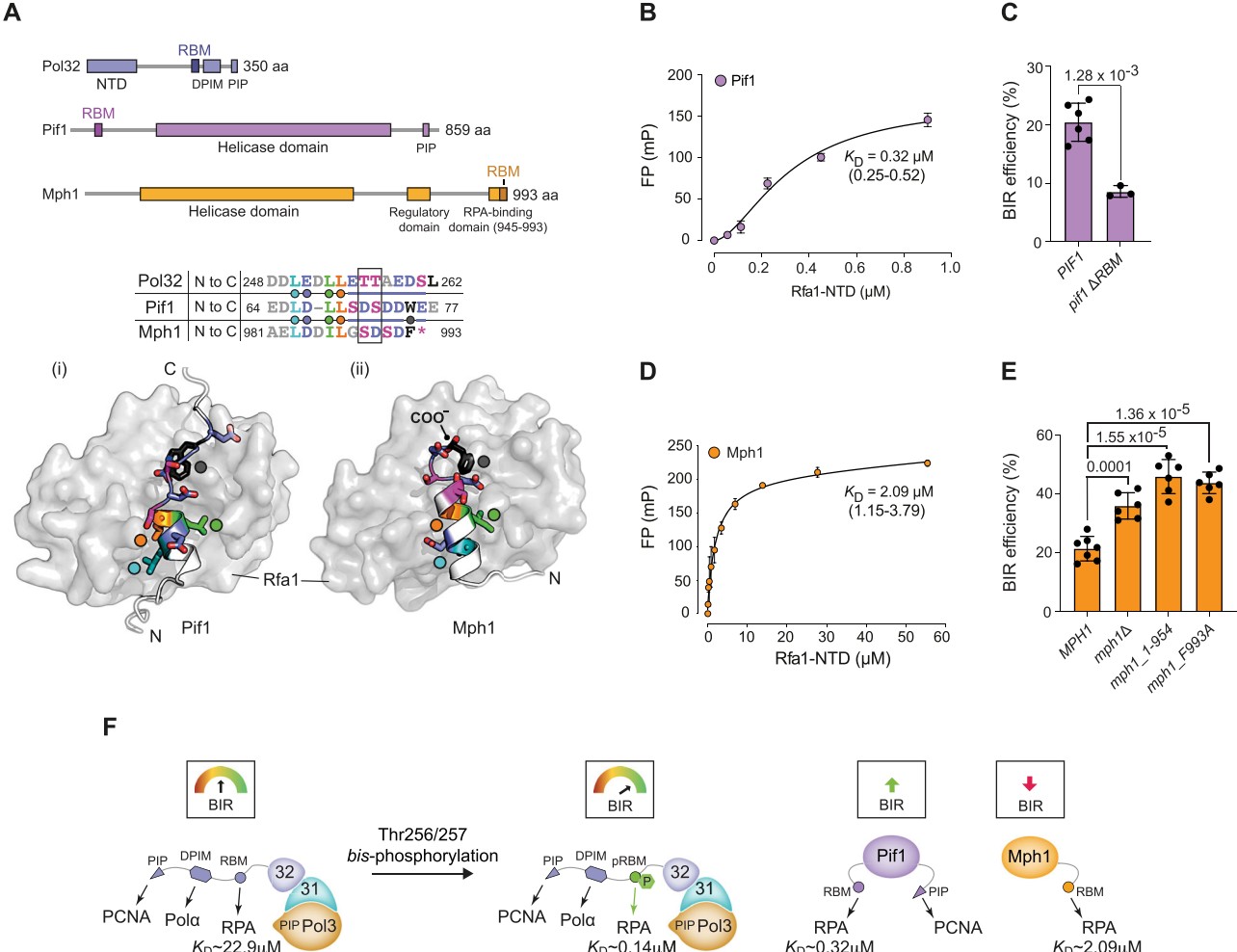

**Fig. 4 | Pif1 and Mph1 bind the Rfa1 NTD like the Pol32 pRBM. A** Schematic representation of Pol32, Pif1, and Mph1 highlighting functional domains and the location of RBMs. **B** Fluorescence polarisation (FP) experiment demonstrating binding between a fluorescein-labelled Pif1 RBM peptide and the Rfa1 NTD, and calculated dissociation constant ($K_D$) (mean ± SD, with uncertainty in the mean provided by 95% confidence intervals in brackets; $n$ = 3 technical replicates). **C** BIR efficiency measurements for strains harbouring wild-type Pif1 or a variant with an internal deletion of the RBM (mean ± SD; *PIF1*, $n$ = 6; *pif1 ΔRBM*, $n$ = 3 independent biological repeats). Statistical significance of differences in BIR efficiency between strains was assessed by a two-tailed Welch's $t$ test with $p$-values indicated. **D** Fluorescence polarisation (FP) experiment demonstrating binding between a fluorescein-labelled Mph1 RBM peptide and the Rfa1 NTD, and calculated dissociation constant ($K_D$) (mean ± SD, with uncertainty in the mean provided by 95% confidence intervals in brackets; $n$ = 3 technical replicates). **E** BIR efficiency measurements for strains harbouring wild-type Mph1 or variants with an RBM deletion or F993A point mutation (mean ± SD; $n$ = 6 independent biological repeats). Statistical significance of differences in BIR efficiency between strains was assessed by a two-tailed Welch's $t$ test with $p$-values indicated. **F** Pol32 RBM phosphorylation allows cells to significantly enhance the affinity of Pol32 for RPA and BIR efficiency. Pif1 and the BIR antagonist Mph1 intrinsically bind RPA with high affinity, and Pol32 RBM phosphorylation may counteract the negative effect of Mph1 on BIR efficiency. Source data are provided as a Source Data file.

dependence of BIR efficiency on the Pif1 RBM[20], an internal deletion of the Pif1 RBM resulted in a significant BIR defect (Fig. 4C).

To validate the Mph1 RBM, we synthesised a fluorescein-labelled peptide derived from Mph1 residues 972 to 993 and determined a $K_D$ of 2.09 μM with Rfa1 NTD by FP (Fig. 4D). Thus, like Pif1, Mph1 contains an RBM with an intrinsic affinity to Rfa1 similar to phosphorylated versions of the Pol32 RBM. As expected, an FP competition experiment between the Mph1 and Pol32 RBM peptides showed that binding is mutually exclusive, confirming that both proteins target the Rfa1 NTD basic-hydrophobic groove (Supplementary Fig. 8C). To test whether RPA interactions may be required for the role of Mph1 in BIR, we truncated the protein to remove the 39 C-terminal aa that have been shown[50] to be required for interactions with RPA. Cells expressing the truncated Mph1_1-954 from the endogenous locus exhibited significantly elevated BIR efficiency, phenocopying a complete deletion of *MPH1* (Fig. 4E). Consistently, BIR intermediate formation was higher in *mph1_1-954* cells compared to *MPH1* wild-type control cells as measured by PCR[23] 6 h after DSB induction (Supplementary Fig. 8D). In contrast, Mph1_1-954 was not associated with the spontaneous mutator phenotype[52] that is caused by loss of Mph1 (Supplementary Fig. 8E). These results demonstrate that Mph1_1-954 remains active but loses the ability to limit BIR efficiency by D-loop dissociation. Finally, we mutated Mph1 Phe993 to alanine and found that this single amino acid change results in significantly elevated BIR efficiency, phenocopying Mph1_1-954 (Fig. 4E). We conclude that the high-affinity RPA binding mode, which relies on RBM contacts within the Rfa1 NTD basic-hydrophobic groove and the hydrophobic side-pocket (ref. 14), is critical for Mph1 to engage and disassemble D-loop intermediates during BIR.

Overall, these findings show RBM-mediated RPA interactions of each BIR replisome component and BIR antagonist Mph1 impact homologous recombination-dependent DNA synthesis, highlighting a critical scaffolding function of RPA bound at recombination intermediates. Pif1 and Mph1 intrinsically bind RPA with high affinity, whereas phosphorylation is required to activate high-affinity RPA binding by the Pol32 RBM, providing a mechanism for the dynamic regulation of BIR efficiency (Fig. 4F).

## A model for the actions of Pol32 in BIR

The existing cryo-EM structure of the budding yeast Pol δ-PCNA-DNA complex does not resolve the Pol32 CTD[53]. We therefore modelled full-length Pol32 *bis*-phosphorylated at Thr256/257 as part of the Pol δ-PCNA-DNA complex and included the Rfa1 NTD (Fig. 5A and Supplementary Fig. 9A). As expected, modelling shows that the Pol δ catalytic subunit Pol 3 employs a noncanonical PIP-box motif embedded in the cysteine-rich metal-binding motif A (CysA/PIP) to bind one subunit of the trimeric PCNA DNA clamp; recapitulating features of the published structure. Pol31 binds to Pol3 and is in turn, bound by the Pol32 NTD. A helix-turn-helix motif (which we refer to as Hook; Pol32 aa 164–218) within the Pol32 CTD bound to one monomer-monomer junction of the PCNA homotrimer (Fig. 5B). This mode of interaction is distinct from that of a canonical PIP-box with the PCNA interdomain connector loop (IDCL), but resembles that previously described[54] for a helix-turn-helix motif within translesion polymerase Rev1 and PCNA. The Pol32 Hook may provide another PCNA anchor point for Pol32, but is not sufficient to support detectable PCNA interactions or BIR efficiency (see Fig. 1E and Supplementary Fig. 1C). Interestingly, the Pol32 PIP-box is consistently engaged with a PCNA IDCL in our models. This suggests that the Pol32 CTD is sufficiently flexible to allow the Pol32 PIP-box to reach one of the two PCNA IDCL binding sites not occupied by Pol3 (Supplementary Fig. 9B). Importantly, we observed a consistently predicted interface between the phosphorylated Pol32 RBM and the Rfa1 NTD in the context of the entire Pol δ-PCNA-DNA-Rfa1 NTD assembly (Fig. 5A and Supplementary Fig. 9A). Moreover, the conformational flexibility offered by the length of the intervening linker

sequence between the Rfa1 NTD and the subsequent RPA subdomains, as well as that of the Pol32 CTD itself, predicts that Pol32 could interact with RPA bound to a displaced ssDNA strand at a D-loop, even while Pol δ is engaged with PCNA at the primer junction during recombination-dependent DNA synthesis (Fig. 5C). As expected, there is no consistent positioning or interaction of the Pol32 DPIM with the Pol δ-PCNA-DNA-Rfa1 NTD assembly. This is in line with the requirement for the DPIM for interactions of Pol32 with Pol α, as shown by our yeast-two-hybrid analysis (Supplementary Fig. 1D) and recently published[55] binary AlphaFold models of the Pol32 DPIM-Pol α interaction. Pol32 DPIM-Pol α interactions come into play late in the BIR process, when the Pol α-primase complex initiates lagging strand synthesis on the nascent strand[22,56–58]. Pol32-Pol α interactions may then support the exchange for Pol δ and enhance BIR efficiency by promoting the conversion of the nascent ssDNA into duplex DNA[23,24]. Taken together, our model is consistent with the independent contributions which we have observed the Pol32 RBM and PIP-box make to BIR efficiency. We propose that multi-point interactions of Pol32 and Pif1 (refs. 20,59) with PCNA and RPA-bound recombination intermediates ensure the proper recruitment and retention of Pol δ and the accessory D-loop helicase Pif1 at BIR intermediates, allowing optimal BIR reactions to take place (Fig. 5D).

## Discussion

We identify a BIR-promoting Pol32 RBM that mediates Pol δ-RPA interactions embedded within the disordered Pol32 CTD. The combination of the Pol32 RBM and the previously described PIP-box and DPIM, which support PCNA and Pol α interactions, respectively[23,33], account for the ability of Pol32 to promote BIR efficiency. Removal of these three protein interaction domains from the Pol32 CTD does not appear to compromise conventional DNA replication. Thus, the Pol32 NTD alone is sufficient to maintain the stability of Pol δ, and tethers to it the CTD, which contributes BIR-specific roles.

The most intriguing feature of the Pol32 RBM is its enhanced affinity for RPA upon *bis*-phosphorylation at Thr256/257. This is reminiscent of Ddc2, whose phosphorylation at both Ser10 and Ser11 promotes RPA interactions. However, these phospho-sites are positioned outside the Rfa1 basic-hydrophobic groove, where pSer11 is able to engage a second Rfa1 molecule[37]. The resulting intermolecular interactions between two Mec1-Ddc2-RPA dimers are thought to promote Mec1-Ddc2 recruitment and the formation of so-called signalling arrays on RPA-coated ssDNA to facilitate DNA damage checkpoint signalling. In the case of Pol32, Thr256/257 phosphorylation modulates the interaction strength with a single Rfa1 NTD molecule. Thus, single or double phosphorylation of the Pol32 RBM increases binding affinity in two ways: firstly, the phosphate groups added at Thr256/257 pickup hydrogen-bond interactions with the basic residue-pairings of Rfa1 Arg91/Lys95 and Arg35/44. Secondly, phosphorylation serves to introduce a kink into the path of the bound RBM peptide, thereby enabling a high-affinity binding modality which positions Leu262 such that it is now able to bind to a hydrophobic side-pocket utilised by other RPA clients independently of phosphorylation. This suggests a dynamic mechanism for the control of BIR efficiency whereby Pol32 phosphorylation stimulates the recruitment and/or retention of Pol δ at BIR-intermediates bound by RPA to promote D-loop DNA synthesis (Fig. 5D).

Disruption of the Pol32 RBM is associated with a slight delay in BIR kinetics, but, unlike removal of the PCNA-binding PIP-box, does not translate into an overt BIR efficiency defect in our reporter strains unless the PIP-box is also removed. PCNA encircles the DNA and acts as a processivity factor for Pol δ. Our structural models of the Pol δ-PCNA-DNA complex indicate that the Pol32 PIP-box binds PCNA, thus providing a physical connection between the processivity clamp and Pol δ at the primer junction. This is consistent with biochemical results[33] demonstrating that the Pol32 PIP-box enhances DNA synthesis by Pol

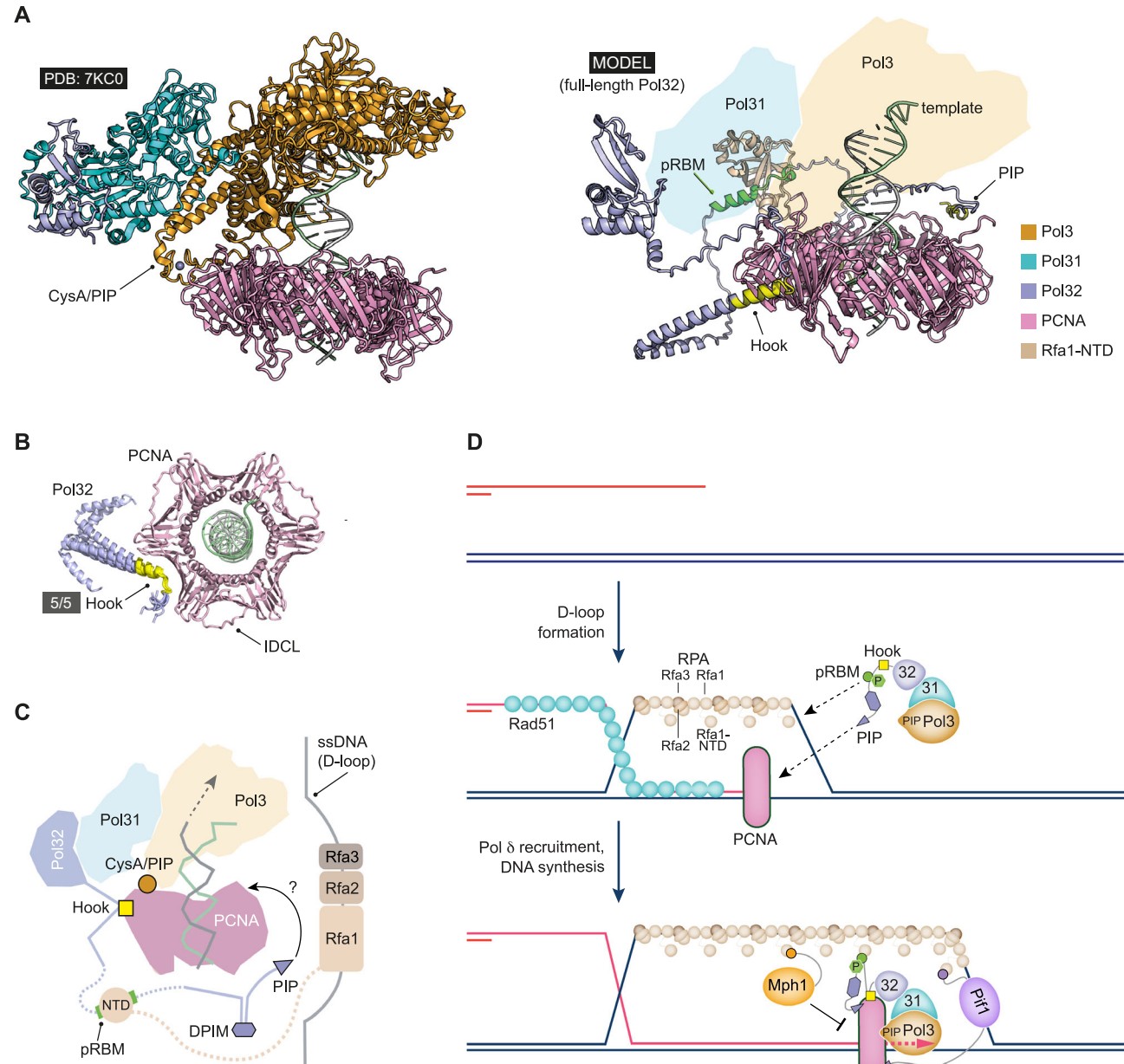

**Fig. 5 | A model for the role of Pol32 in BIR. A** Left, molecular cartoon representation of the cryo-EM structure of yeast Pol δ in complex with PCNA and a primer-template junction (PDB: 7KC0). Please see the associated key for additional information. Right, representative AlphaFold 3 model including full-length Pol32 with key molecular features highlighted. For clarity, molecular cartoons for Pol31 and Pol3 are omitted, but with their relative positions represented by shaded polygons. **B** Molecular cartoon showing superposition of a Pol32 helix-turn-helix motif (Hook) in complex with PCNA, isolated from each of the five generated AlphaFold models. The position of amino acids 151–220 within this motif are highlighted in yellow as those in close proximity to the PCNA monomer-monomer interface. **C** Schematic representing the potential organisation of the Pol32 CTD and its component motifs (Hook, pRBM, DPIM and PIP-box; see main text) with respect to the Pol δ-PCNA-DNA complex. The linker connecting the Rfa1 NTD to the rest of the RPA complex (dotted line) is sufficient to span the distance to the ssDNA in a displacement loop (D-loop). **D** Cartoon illustrating how Pol32 facilitates BIR through a series of protein interaction modules aligned along its CTD. When BIR is initiated at an RPA-bound D-loop, multi-point Pol32-PCNA and phosphorylation-enhanced Pol32-RPA interactions promote Pol δ engagement and continued BIR synthesis to boost BIR efficiency. Competitive interactions between the Pol32 RBM and the Mph1 RBM may help to protect D-loops from Mph1-dependent disassembly by stabilising Pol δ at the primer junction. More details are provided in the main text.

δ-PCNA in vitro. The Pol32 RBM provides physical interactions outside the core BIR replisome, with RPA-bound ssDNA that exists in the immediate vicinity of the D-loop primer-junction. We envisage that such interactions affect Pol δ processivity less directly by promoting the recruitment of Pol δ to D-loops and/or aiding its retention if Pol δ disengages prematurely from PCNA. This model is consistent with the ability of the Pol32 RBM to compensate to some extent for the loss of the PIP-box, when BIR efficiency becomes profoundly dependent on

the RBM and the Thr256/257 phosphorylation sites needed to enhance the interaction strength with RPA. While the functional contribution of the RBM to BIR efficiency is largely masked in HO-based experimental systems when Pol32-PCNA interactions are intact, it is revealed by enhanced BIR efficiency in the presence of Thr256/257 phospho-mimetic mutations, and we anticipate a particular relevance of the RBM for optimal Pol δ recruitment/retention when BIR is in competition with alternative repair pathways at naturally occurring DSBs.

Interestingly, we find that the actions of Mph1, which targets extended D-loops[24,49] and dissociates the nascent strand[47] to restrict BIR efficiency[48,60], depend on a newly defined RBM at the Mph1 C-terminus. The Mph1 RBM binds RPA more avidly than the non-phosphorylated Pol32 RBM, but with similar affinity to the phosphorylated form. It is therefore tempting to speculate that Pol32 RBM phosphorylation may help Pol δ to locally compete with Mph1 for recruitment to D-loops and the primer junction, thus promoting BIR and mitigating D-loop collapse. As a consequence, the risk of chromosome aberrations[5,6] arising from ectopic recombination upon premature release of the nascent strand and invasion at a non-contiguous site with sequence homology[61–64] could be reduced. Conversely, Pol32 RBM dephosphorylation may favour D-loop disassembly, allowing cells to terminate BIR and limit potentially mutagenic[8] BIR-dependent DNA synthesis.

Regulation of the Pol32 RBM phosphorylation status may involve multiple kinases. Ddc2 RBM Ser10 and Ser11 phosphorylation is driven by the constitutively active acidophilic kinase CKII, and its regulation is likely to involve the actions of phosphatases[37]. Consistent with the sequence similarity between the Ddc2 and Pol32 RBMs, we find that CKII can also phosphorylate the Pol32 RBM in vitro and may therefore contribute to the phospho-regulation of Pol32. Several other kinases have been implicated in Pol32 RBM phosphorylation in vivo[40–44]. We confirmed the ability of Cdc5 (PLK) and Cdc7 (DDK) to directly phosphorylate the Pol32 RBM in vitro. Phospho-proteomics suggest that Cdc7 phosphorylates serine and threonine residues with Asp/Glu/pSer/pThr at the plus 1 position[44], and bis-phosphorylation at Thr256/257 within Pol32 conforms with this preference. Interestingly, Cdc7 showed the most robust activity towards the Pol32 RBM in vitro and has previously been reported[46] to promote BIR in vivo. DDK is active from the start of S-phase through to mitosis, while PLK is activated at the G2/M transition. Both kinases act to align cellular processes with cell cycle stage and are known to co-regulate recombination-associated reactions, for example, by coordinating the resolution of recombination intermediates with mitotic entry[65]. Their activity towards Pol32 would suggest that the Pol32 RBM becomes increasingly activated for high-affinity RPA binding as cells progress through S phase, which could promote effective BIR at broken replication forks in S/G2 phase. However, the details of Pol32 phospho-status regulation, which may involve additional kinases as well as phosphatases, remain to be elucidated in future studies.

The reliance of Pol δ, Pif1, and the BIR antagonist Mph1 on RBMs in the context of BIR shows that RPA serves as a critical platform for the orchestration of the BIR machinery (Fig. 5D). Pol32 phosphorylation allows fine-tuning of the affinity of the Pol32 RBM for RPA, identifying one way in which cells might control the stoichiometry or handover of BIR factors at recombination intermediates to regulate D-loop stability and BIR outcomes. More generally, the phospho-activated RBM within Pol32 provides a paradigm for dynamic control of protein association with the Rfa1 NTD basic-hydrophobic groove and hydrophobic pocket to fine-tune DNA repair. Consistent with this notion, phospho-dependent interactions of the Werner syndrome helicase (WRN) with RPA have recently been reported to promote DNA replication fork recovery[66]. With RPA being involved in virtually all DNA transactions during replication, repair, and recombination involving ssDNA, and a growing number of identified Rfa1/RPA70 client proteins[14], pRBM-Rfa1/RPA70 interactions could be broadly relevant in genome maintenance.

## Methods
### Yeast strains
All *S. cerevisiae* strains used in this study were derived from BY4741 (ref. [67]), JRL092 (ref. [23]), AM1003 (ref. [36]), or pJ69-4A (ref. [68]) and are described in Supplementary Table 1. Yeast was grown at 30 °C, or the indicated temperatures, in rich (YP) or minimal media supplemented with either glucose (D), galactose (GAL), or raffinose (RAFF) to a final concentration of 2% (w/v). For solid media, agar was added to a final concentration of 1% (w/v).

### Yeast two-hybrid assay
For yeast two-hybrid experiments[69], strain pJ69-4A was co-transformed with vectors pGBKT7 and pGADT7, each either unmodified or modified to express protein constructs fused to the Gal4-binding or Gal4-activating domain. Logarithmic cultures of the respective strains (Supplementary Table 1) grown in minimal medium lacking tryptophan and leucine were diluted and spotted on control plates (minimal medium lacking tryptophan and leucine) and selective plates (minimal medium lacking tryptophan, leucine, and histidine) containing the histidine biosynthesis inhibitor 3-amino-1,2,4-triazole (3-AT). The interaction between GBD-Stn1 and Ten1 with a GAD-Pmt3-Tpz1 fusion[70] was used as a positive control.

### BIR efficiency measurements
To measure ectopic BIR (JRL092-derived strains), logarithmically growing cells cultured in YPRAFF medium were plated in appropriate dilutions onto YPD or YPGAL medium and grown into colonies. Cell viability as a measure of DSB repair efficiency after HO endonuclease induction was derived by dividing the number of colony-forming units (CFU) on YPGAL by that on YPD. For JRL-derived strains, HO cut efficiency was routinely tested by PCR, and a PCR product spanning the *HOcs* locus was no longer obtained after exposing cells to galactose for 1 h (for example, see Supplementary Fig. 1A). Colonies on YPGAL were always replica plated onto YPD medium supplemented or not with 200 μg/ml hygromycin B. This showed that 2-4% of DSB repair events led to retention of the *HPH* resistance marker, in good agreement with previously determined[23] rates of BIR-independent DSB repair by gene conversion in JRL-derived strains. This fraction was subtracted from the value for DSB repair efficiency to calculate BIR efficiency for a given strain. In addition, BIR survivors were routinely validated using PCR on random colonies to show the presence of a newly formed, contiguous *CAN1* marker at the former *HOcs* locus (see Supplementary Fig. 1B).

To measure allelic BIR (AM1003-derived strains), logarithmically growing cells cultured in YP-raffinose medium were plated in appropriate dilutions onto YPD or YPGAL medium. Colonies on YPGAL were replica-plated onto minimal media lacking adenine and leucine, leucine only, and lacking leucine whilst containing limited amounts of adenine (5 mg/l) to determine the contribution of different DSB repair pathway to cell survival: *ADE+ leu-* colonies arise by BIR, *ADE + LEU+* colonies by gene conversion; half-crossover formation and chromosome loss give rise to white *ade3- leu-* and red *ade1- leu-* colonies on minimal media supplemented with limited amounts of adenine, respectively.

Every strain was tested at least three times and differences in mean values for BIR efficiency were assessed for significance using Welch's *t* test.

### BIR kinetics measurements
PCR analysis in JRL-derived strains was performed on genomic DNA prepared such that final DNA concentrations were equal across samples[23]. HOcs cleavage was confirmed by amplifying genomic DNA with primers P3 (5′-TTGCATGGTTATTTATCTCAATCTC) and P4 (5′-ACGTCAAGACTGTCAAGGA). BIR intermediates were amplified with primers P1 (5′-TTTCCCGACGAGAGTAAATGGCGA) and P2 (5′-GGTGCAAAAGCCGTGAAACC) (see Fig. 1D) using Phusion polymerase (NEB). PCR products were run on 0.9% agarose gels in 0.5 x TBE buffer with ethidium bromide and quantified (signal to noise ratio) using a Gel Doc imaging system (Syngene Ingenius LHR). Ratios were divided by those determined for a control amplified from an independent locus (*CDC5;* using primers 5′-GGTGAAGGTGGATTTGCTC and 5′-ATGTCCAGAATGCTTACCCAT) on chromosome XIII from the same

input and under the same conditions and normalised to the value obtained with wild-type *POL32* at 21 h. Uncropped gels are provided within the Source Data file.

## Immunoprecipitation

Yeast cell cultures were harvested at 4 °C and washed sequentially with an ice-cold solution of 20 mM HEPES-KOH (pH 7.9) and a wash buffer containing 100 mM HEPES-KOH (pH 7.9), 50 mM potassium acetate, 10 mM magnesium acetate and 2 mM EDTA. Cells were then centrifuged and resuspended in wash buffer supplemented with 2 mM sodium glycerophosphate (Johnson Matthey), 2 mM sodium fluoride (Fisher Scientific), 1 mM DTT, 1% (v/v) protease inhibitor cocktail for fungal and yeast extracts (Sigma), and 0.24% (w/v) EDTA-free complete protease inhibitor cocktail (Roche). The mixture was flash frozen in liquid nitrogen and protein immunoprecipitation[71] was carried. In brief, cells were lysed using a mechanical pestle and mortar (Cole-Parmer SPEX Sample Prep 6850) pre-cooled at −80 °C. To each gram of thawed lysate, 0.25 ml of a solution of 50% (v/v) glycerol, 100 mM HEPES-KOH (pH 7.9), 50 mM potassium acetate, 20 mM magnesium acetate, 0.5% Igepal CA-630, 2 mM EDTA supplemented with 2 mM sodium glycerophosphate, 2 mM sodium fluoride, 1 mM DTT, 1% (v/v) protease inhibitor cocktail for fungal and yeast extracts (Sigma), and 0.24% (w/v) EDTA-free complete protease inhibitor cocktail was added. Pierce universal nuclease was added to a final concentration of 1.6 μ/μl, and samples were agitated on a rotating platform at 4 °C for 40 min. Lysates were then clarified by successive centrifugations at $4000 \times g$ and $13{,}000 \times g$.

50 μl analytical samples of the supernatant were mixed with 1.5 x Laemmli buffer (100 μl) and boiled for four minutes (Input samples). The remaining supernatant was incubated with 100 μl TAP-beads (M270 Dynabeads epoxy beads coupled with anti-sheep IgG) at 4 °C for 2 h. The beads were then washed three times with 100 mM HEPES-KOH (pH 7.9), 50 mM potassium acetate, 20 mM magnesium acetate, 2 mM EDTA, 0.1% (v/v) Igepal CA-630. The first wash was supplemented with 2 mM sodium glycerophosphate, 2 mM sodium fluoride, 1 mM DTT, 1% (v/v) protease inhibitor cocktail for fungal and yeast extracts (Sigma), and 0.24% (w/v) EDTA-free complete protease inhibitor cocktail (Roche). After washing, the beads were mixed with 100 μl 1 x Laemmli buffer and boiled for four minutes (IP samples). Input and IP samples were analysed by western blotting.

## Protein kinase assays

TAP-tagged kinases were immunoprecipitated as described above except for lysate resuspensions and wash buffers containing 300 mM potassium acetate. TAP beads were then washed twice with 20 mM Hepes-KOH (pH 7.9), 150 mM NaCl. Radioactive kinase assays: the beads were mixed with 75 μM of peptide substrate, 0.625 mM of ATP and 1.5 μl of ATP [γ-$^{32}$P] (6000 Ci/mmol) in a buffer of 20 mM Hepes-KOH (pH 7.9), 150 mM NaCl, 10 mM MgCl$_2$ and 0.5 mM TCEP and incubated in a 30 °C water bath. The supernatant containing the phosphorylation substrate was then mixed with 1/3 volume of 3 x Laemmeli buffer and boiled for 5 min. Samples were resolved by SDS-polyacrylamide gel electrophoresis, and the gels were dried under vacuum at 80 °C (Gel Dryer Model 583, BioRad). The dried gels were exposed to phosphor-imaging plates and imaged using a Typhoon Imager (Amersham). Uncropped gels and phosphor images are provided within the Source Data file.

For non-radioactive kinase assays followed by mass spectrometric analysis, the TAP beads were mixed with 75 μM of peptide substrate and 2.5 mM of cold ATP in a buffer of 20 mM Hepes-KOH (pH 7.9), 150 mM NaCl, 10 mM MgCl$_2$, 1 x PhosSTOP (Merck) and 0.5 mM TCEP, before incubation in a 30 °C water bath. After 2 or 24 h, the supernatant containing the phosphorylation substrate was immediately frozen at −20 °C.

## TCA protein extraction

Logarithmically growing cells were harvested by centrifugation, washed in deionised water, and resuspended in 300 μl 20% (w/v) tri-chloroacetic acid (TCA). After adding a small quantity of acid-washed glass beads, cells were lysed by shaking vigorously thrice for one minute, with 20 s of cooling on ice between shaking cycles. The supernatant was recovered and combined with a subsequent wash of the remaining cell debris and glass beads with 300 μl 5% (w/v) TCA. Proteins were pelleted by centrifugation at $4000 \times g$ for 10 min and resuspended in 100-200 μl of 1 x Laemmli buffer supplemented with 150 mM Tris base. Samples were boiled for four minutes and cleared by centrifugation at $4000 \times g$ for 10 min before western blotting.

## Western blotting

Protein samples were resolved by SDS-polyacrylamide gel electrophoresis on 5–10% polyacrylamide gels and transferred onto nitrocellulose membranes. Membranes were incubated in TBST containing 5% (w/v) skimmed milk and one of the following antibodies at the indicated dilutions: anti-TAP, consisting of peroxidase anti-peroxidase soluble complex antibody produced in rabbit (Sigma P1291, 1:5000), mouse anti-Myc (Santa-Cruz sc-40, 1:1000), mouse anti-FLAG (Sigma F3165, 1:5000), rabbit anti-Rfa1 (Abcam ab221198, 1:3000). After three washes in TBST, the relevant HRP-conjugated secondary antibody was added (Cell Signalling Technology goat-anti-rabbit 7074, 1:10,000; Cell Signalling Technology horse-anti-mouse 7076, 1:10,000). Finally, membranes were washed thrice in fresh TBST, treated with western blotting reagents, and chemiluminescence was captured using an ImageQuant 800 (Cytiva) western blot imaging system. Uncropped blots are provided within the Source Data file.

## Recombinant protein

For isolation of Rfa1 NTD protein[37] corresponding to amino acids 1 to 132, a 6xHis-GST-Rfa1 NTD construct was expressed overnight in *E. coli* BL21 (DE3) cells in a shaking incubator set at 180 rpm and 20 °C. Cells were harvested by centrifugation and frozen at −80 °C. After resuspension in a buffer containing 50 mM HEPES, 150 mM NaCl, 0.5 mM TCEP (pH 7.5), cells were lysed by sonication, and cell lysates were clarified by centrifugation. 6 x His-GST-Rfa1 NTD was affinity-captured using Ni Sepharose (Cytiva), eluted using imidazole, then applied to a size exclusion chromatography column, equilibrated in 20 mM HEPES (pH 7.5), 150 mM NaCl at 25 °C. Rfa1 NTD ( > 98% purity) was concentrated, flash frozen in liquid nitrogen, and stored at −80 °C. The Rfa1 NTD Arg91Ala and Lys95Ala double mutant (Rfa1-MUT) was expressed and purified in a similar manner, except that Co Sepharose (Cytiva) was used.

## Peptides

His$_6$-tagged Trx-RBM fusions were expressed in *E. coli* strain BL21(DE3). Transformed bacteria were cultured in Turbo Broth (Molecular Dimensions) supplemented with 50 μg/ml carbenicillin, at a temperature of 37 °C, in an orbital-shaking incubator set at 200 rpm. Recombinant protein expression was induced by the addition of 0.2 mM isopropyl β-d-1-thiogalactopyranoside once the optical density at 600 nm had reached 1.5. The temperature of the incubator was then reduced to 20 °C, and the cultures grown for an additional period of -16 h. After this, cells were harvested by centrifugation at $4000 \times g$ for 20 min at 4 °C and then stored at −20 °C. Cell pellets corresponding to 100 ml of cell culture were lysed by thawing on ice and sonication in buffer A (50 mM HEPES pH 7.5, 250 mM NaCl, 0.5 mM TCEP, 10 mM imidazole) supplemented with protease inhibitors (Roche). Insoluble material was removed by high-speed centrifugation at $40{,}000 \times g$ for a period of 1 h at 4 °C. The resulting supernatant was passed through a 5 μm filter and then applied to a batch/gravity column containing 1 ml of TALON resin (TaKaRa Bio). The resin was washed with 5 column volumes of buffer A to remove unbound material, before the addition

of buffer B to elute the retained protein (50 mM HEPES pH 7.5, 250 mM NaCl, 0.5 mM TCEP, 300 mM imidazole).

Fractions containing the Trx-Pol32 RBM fusion proteins were identified by SDS-PAGE, pooled, and then applied to a Superdex S75 16 600 size exclusion chromatography column (Cytiva) pre-equilibrated in buffer C (20 mM HEPES, pH 7.5, 250 mM NaCl, 0.5 mM TCEP). Fractions containing the recombinant protein were again confirmed by SDS-PAGE, pooled and then concentrated to -2.5 mg/ml by use of centrifugal concentrators with a 5 kDa molecular weight cut-off (Sartorius). The purified protein was flash-frozen in aliquots and stored at −80 °C until required.

Purified synthetic peptides (Supplementary Table 2) were purchased from Peptide Protein Research Ltd, Bishops Waltham, UK. All peptides used for fluorescence polarisation are labelled at their N-terminus with Fluorescein, followed by a glycine-tyrosine-glycine linker sequence (GYG).

## Fluorescence polarisation

Fluorescein-labelled peptides (100 nM) were incubated at room temperature with increasing amounts of purified Rfa1 NTD in 20 mM HEPES-NaOH (pH 7.5), 150 mM NaCl, 0.5 mM TCEP in a black 96-well polypropylene plate (VWR). Fluorescence polarisation was measured in a CLARIOstar multimode plate reader (BMG Labtech GmbH). All data were fitted by non-linear regression to a one-site specific binding model, using Prism (v. 10.4, GraphPad Software), allowing calculation of the reported dissociation constants ($K_D$). For the competition experiment, increasing amounts of an unlabelled Mph1 peptide were included in binding reactions containing 7 μM of Rfa1 NTD and 100 nM of the fluorescently labelled Pol32 pThr257 peptide.

## Mass Spectrometric Analysis

For mass spectrometry sample preparation following non-radioactive protein kinase assays, frozen Trx-Pol32 RBM samples were thawed, reduced with 5 mM TCEP for 1.5 h, and alkylated with 10 mM iodoacetamide in the absence of light. Alkylation was then quenched by the addition of 5 mM DTT. The samples were desalted using PD SpinTrao G-25 columns (GE Healthcare) and digested with 2 μg of Trypsin Gold (Promega) at 37 °C. After desalting using C18 tip-columns (Thermo Fisher), samples were concentrated using a vacuum-concentrator. Finally, the samples were reconstituted in 40 μl of a solution of 0.5% acetonitrile and 1% formic acid and optionally subjected to titanium dioxide columns (Thermo Fisher) for phospho-peptide enrichment. LC-MS/MS analysis was undertaken using a Vanquish Neo UHPLC coupled to an Orbitrap Exploris 480 equipped with FAIMS Pro Duo interface (Thermo Fisher Scientific), following a data-dependent MS2 experiment with internal EasyIC (RunStart) as lock mass. In summary, samples were loaded onto a PepMap Neo column (5 μ C18, 300 μm x 5 mm, 100 Å pore size; Thermo Fisher Scientific) at a flow rate of 20 μl/min using 0.1% formic acid in H$_2$O and separated on an analytical EASY-Spray PepMap Neo column (2 μm C18, 75 μm x 500 mm, 100 Å pore size, Thermo Fisher Scientific) with a flow rate of 350 nl/min with a linear gradient of 1 to 35% solvent B (80% (v/v) acetonitrile, 0.1% (v/v) formic acid) over 60 min. Ion source conditions were maintained at 1.75 kV positive voltage, 275 °C ion transfer tube temperature and standard resolution with 3.8 l/min FAIMS carrier gas flow throughout the acquisition. Mass spectrometric data was acquired using the following configuration: MS1Orbitrap resolution was set at 60,000 (*m/z* 200) from a mass range of 375–1200 amu using a normalised automatic gain control (AGC) target of 300% and maximum injection time (maxIT) set to auto; MS2 Orbitrap resolution was set at 15,000 (*m/z* 200) with first acquired mass starting at 120 amu, an isolation window of 2 amu, and AGC set to standard with 40 msec maxIT. Fragmentation was carried out using higher-energy collisional dissociation (HCD) with a normalised collision energy (NCE) of 30. For the FAIMS, two compensation voltages (CV), 45 and 65 V with 2 and 1 sec cycle time,

respectively, for each CV were used with an RF lens of 50%. The following mass filters were also used: MIPS (peptide mode), intensity threshold (8.0E3), charge state (2-6) and dynamic exclusion (30 sec) mass filters were used. Data was analysed using Proteome Discoverer 3.0 (Thermo Fisher Scientific) with Sequest HT and MS Amada 2.0 as the search engines. An in-house modified peptide sequences FASTA file (based on our Pol32 RBM construct) was used to search against. For both Sequest HT and Amada engine precursor and fragment mass limits were fixed at 5 ppm and 0.02 Da. Only the phosphorylation for serine, threonine, and tyrosine residues was set as a dynamic modification. The protein FDR and the peptide spectrum match were both set to 0.01.

## Modelling with AlphaFold and molecular graphics

Molecular models were generated using the AlphaFold Server available online at https://alphafoldserver.com, which currently implements version 3 of AlphaFold[38]. Molecular graphics were generated using PyMOL 3.1 (The PyMOL Molecular Graphics System, Version 3.0, Schrödinger, LLC).

## Reporting summary

Further information on research design is available in the Nature Portfolio Reporting Summary linked to this article.

## Data availability

The datasets generated and/or analysed in the current study are available in the Source Data. All AlphaFold models reported in this manuscript have been deposited in the University of Sussex Research Data Repository (https://sussex.figshare.com) and are available at https://doi.org/10.25377/sussex.31281529. The mass spectrometry proteomics data have been deposited to the ProteomeXchange Consortium (http://www.proteomexchange.org/) via the PRIDE partner repository with the dataset identifier PXD074712. Source data are provided in this paper.

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

## Acknowledgements

This work was supported by UKRI through BBSRC research grant BB/W008505/1 (R.A. and U.R.). U.R acknowledges the support received from the Department for Science, Innovation and Technology through the Academy of Medical Sciences Professorship award AMSPR1\1018. We are grateful to Professor Jim Haber, Brandeis University, for yeast strains and valuable discussions, and Professor Grzegorz Ira, Baylor College of Medicine, and Dr Alessandro Bianchi, University of Sussex, for strains and reagents. We acknowledge the support and expertise provided by Dr Ramón González-Méndez from the Mass Spectrometry Research Facility, School of Life Sciences, University of Sussex. We thank Professor Boris Pfander, TU Dortmund, for helpful discussions, and Professors Tony Carr and Aidan Doherty, University of Sussex, for critical reading of the manuscript.

## Author contributions

D.J. and R.A. generated yeast strains and carried out all yeast experiments. A.W.O. purified Trx-Pol32 RBM fusions, performed molecular modelling, FP measurements, and assisted with the kinase assay design. L.A.Y. purified Rfa1 NTD. U.R. designed the study and wrote the manuscript with contributions from all authors.

## Competing interests

The authors declare no competing interests.
