## [Transparent Peer Review file · Nature Communications]

Break-induced replication is enhanced by a phospho-activated RPA-binding module in Pol32

Corresponding Author: Professor Ulrich (Uli) Rass

Version 0:

Reviewer comments:

Reviewer #1

(Remarks to the Author)

DNA double-strand breaks (DSBs), if not properly repaired, can lead to the loss of chromosomal integrity. DSBs can be repaired through non-homologous end joining (NHEJ), which re-ligates broken ends, or through homologous recombination (HR). HR repair occurs via two principal mechanisms: gene conversion (GC) and break-induced replication (BIR). BIR becomes necessary when only one end shares homology with a donor, such as at stalled or broken replication forks or at dysfunctional telomeres.

Numerous studies have established that Pol32 is not required for gene conversion but is essential for BIR. While DNA synthesis can initiate in the absence of Pol32, it does not proceed efficiently during BIR. How Pol32 regulates BIR remains a central question. In this study, the authors show that the interaction between Pol32 and RPA is critical for BIR. The data also suggest that Pol32 interacts with Mph1 and Pif1 via RPA, leading to a very interesting idea that RPA serves as a critical platform for the orchestration of the BIR machinery. The significance of the proposed RPA-Pol32 interaction is not fully clear because RBM deletion did not have significant impact on Pol32-mediated BIR (Fig. 1D). This finding suggests that other critical domains within Pol32 may be responsible for regulating the interactions between Pol32, RPA, Mph1 and Pif1 (Fig. 5).

Specific Points:

1. In addition to the Pol32-RBM-Rpa1 NTD complex (Figures 2 and 4), Alphafold structural predictions of the entire Pol32-RPA complex would assist in constructing a more comprehensive model (Figure 5).
2. Does the assay (Figure 1D) fully exclude the possibility of GC between the two chromosomes? For instance, did the authors confirm the presence of two copies of LEU2 after DSB induction?
3. The efficiency of DSB induction and cell survival following HO expression in each experiment should be shown. Curiously, the assay becomes more reproducible as cell survival decreases (Figures 1E, 2D, 3B).
4. The explanation of the Pol32-CTD (Figure 2C) is missing. Pol32-CTD expression is significantly low (Fig. 2C) although any other C-terminal deletion mutants were expressed well (Fig. 1C). What region does the Pol32 CTD mutation contain?
5. Ddc2, Mph1, and Pif1 interact with Rfa1 via the same interface (Figure 4), suggesting that these proteins compete with Pol32. Does the deletion of DDC2, MPH1, or PIF1 increase the Pol32-Rfa1 interaction (Figure 2C)? The authors should provide a better explanation of how the interplay between Pol32, MPH1, and PIF1 drives BIR. This would help clarify the role of Pol32 in DNA synthesis during BIR.
6. The authors show that Rim11 is required for BIR. Is Rim11 specifically necessary for BIR but not GC? The phosphorylation of Thr-257 by Rim11 needs to be validated through both in vitro and in vivo experiments. Because Pol32-T257D mutations were tested in all the genetic assays (Fig. 3 and Fig. 4B), T257D peptides should be also evaluated in Fig. 4A.
7. Pif1 appears to interact with the N-terminus of Rfa1 similar to Ddc2 and Mph1 (Fig. 4E). Pif1 peptides would be tested in Fig. 4A.

Reviewer #2

(Remarks to the Author)

This work presents interesting data suggesting a dynamic role of RPA in BIR, where RPA interactions with Pol δ , Mph1, and Pif1 dictate the stability and extension of the recombination D-loop during BIR. The authors identify residues within the Pol32 component of Pol δ that are responsible for interaction with RPA. They also investigate the role of serine/threonine phosphorylation within the same region of Pol32 where it interacts with RPA. While phosphorylation sites were previously described, their role remains unknown. The authors show that Pol32 phosphorylation stimulates the Pol δ -RPA interaction and enhances BIR efficiency.

One of the strongest aspects of this work is the identification of Rim11 kinase as a likely regulator of Pol δ phosphorylation, with rim11 Δ mutants exhibiting the same low BIR phenotype as Pol32 phosphorylation mutants. However, somewhat unexpectedly, the pol32 mutant that abolishes interaction with RPA does not, by itself, show any deficiency in BIR. Finally, the authors also identify the C-terminal part of Mph1 as interacting with RPA and regulating BIR, likely by influencing D-loop stability. They propose an interesting model in which Rim11-mediated phosphorylation shifts Rfa1 interaction from Mph1 (D-loop unwinding) toward Pol δ (D-loop extension), thereby stimulating BIR.

This is an interesting study that provides new mechanistic insights into BIR regulation.

Concerns

The analysis of phenotypes is relatively weak, as it relies on a single BIR assay that is highly inefficient, even in wild-type cells (only ~20% complete BIR). The differences between wild-type and mutants are within a $\pm 5\%$ range for some mutants. Additionally, several interesting predictions can be drawn from the proposed model that should be tested in vivo. A few additional experiments would significantly strengthen this work:

1. Test BIR using another assay. The allelic BIR assay, which is commonly used, would be ideal. Decreased D-loop extension often leads to specific phenotypes, such as half-crossovers, which are expected in rim11 Δ mutants.
2. Measure BIR kinetics in an efficient BIR assay. It would be informative to examine BIR kinetics in mutants such as pol32_RBM, rim11 Δ , and pol32 phosphorylation mutants. In particular, pol32_RBM might exhibit a BIR delay.
3. Examine crossover frequency in the Mph1 C-terminal mutant. The mph1 Δ C (1-954) mutant shows reduced interaction with RPA and stimulates BIR, likely due to loss of its effect on D-loop unwinding. A key prediction is that this mutant should exhibit a higher level of crossover outcomes, which should be tested.
4. Test the genetic interaction between Rim11 and Mph1. If Rim11-mediated phosphorylation of Pol32 enhances Pol δ -RPA interaction and reduces the negative impact of Mph1-RPA on BIR, then rim11 Δ mph1-954 double mutants should not exhibit any defects in BIR. Testing BIR efficiency in this double mutant would directly test the authors' model.

Minor Comments:

Rim11 localization and function: Rim11 is typically localized in the cytoplasm. Is it known to phosphorylate any other nuclear proteins? Is there evidence that Rim11 directly phosphorylates Pol32?

Clarification in Lines 39-41: The work first demonstrated that BIR is limited in the repair of replication fork breakage—PMID: 26273056.

Reviewer #3

(Remarks to the Author)

In the manuscript by Jones and co-workers, the authors report the identification of a novel RPA binding motif (RBM) in the Pol32 subunit of the DNA polymerase Delta using the budding yeast as a model organism. Normally, the Pol32 recruitment to DNA is controlled through its interaction with PCNA via a PCNA-interacting peptide (PIP) located at the C-terminus of Pol32. The importance of the identified RBM in vivo, based on DNA repair by BIR, could only be revealed if the Pol32-PCNA interaction is eliminated from the system by the Pol32 C-terminal deletion involving the PIP, as shown by the experiments in figure 2D. The RPA-Pol31-Pol32 in vivo interaction is detected by protein ColPs (Figure 2C).

The in vitro binding experiments and Alpha-fold modelling support the idea that phosphorylation of some S/T residues within the RBM motif strengthens the RPA-Pol32 interaction. The phosphor-site mutations have an expected effect on BIR in the background where the Pol32-PCNA interaction is lost, but behave in a confusing way in full length Pol32. The authors then claim that Rim11 is the kinase responsible for the phosphorylation of the RBM in Pol32.

Finally, a structural parallel is drawn to the RBMs from Pif1 and Mph1, two other proteins known to be involved in BIR and carrying RBMs.

I have two concerns about this manuscript which I would like to discuss separately:

1. the suitability for publishing it in Nature Communications;
2. the experimental part.

As presented, the focus of the manuscript is on identifying a novel RBM in Pol32. Based on the fact that this is just another RBM added to the several previously reported in different organisms, including some of those in budding yeast, the novelty

of this manuscript doesn't advance the field enough to grant a publication in NC, particularly since the biological significance of this RBM in Pol32 is unclear and its potential regulation through phosphorylation is not studied well enough. Based on the experiments in this manuscript, the RBM plays a secondary role to the PIP in the Pol32 recruitment in BIR: with the Pol32 PIP being functional, the RBM has no effect on the efficiency of BIR. The RBM may play a more important role in some other genome stability aspect independent of BIR, but it is not presented in this work and, therefore, it remains unclear how important it is.

With this said, I could see how the focus of the manuscript could be shifted to a more of a big-picture question, to stress the novelty of identifying extended RBMs where phosphorylation of the residues adjacent to the hydrophobic cores plays a role in genome stability via stimulating the recruitment of the repair proteins to DNA. This would be a major revision and require a lot of additional experiments, such as analysing multiple RBMs, the ones in Pif1 and Mph1 at the very least, demonstrating the phosphorylation dependence on the DNA damage response (or other trigger) in vivo, different forms of peptide binding to RPA similar to those for Pol32 in figure 4A, plus addressing the physiological role of RBM phosphorylation in BIR through analysing the phosphosite mutants in the genetic BIR assays. These experiments would then justify the model proposed by the authors in Figure 5 and might make the story conceptually novel enough for NC to consider. As is, there is very little experimental data to support this model: the phosphorylation of RBM in Pol32 is not important for BIR in vivo; the phosphorylation of RBMs in Pif1 and Mph1 hasn't been addressed experimentally. Also, it is not known what triggers the RBM phosphorylation – cell cycle, DNA damage signalling, etc. Therefore, the model in figure 5 is based on computer modelling and imagination, but not supported by sufficient experimental data.

The experimental part:

Figure 1F. The authors use HU to assay different POL32 alleles. The rest of the paper is focused on BIR, i.e. DSB repair involving activation of the G2/M checkpoint rather than an intra-S phase checkpoint triggered by HU. It might be informative to add a spot test using Phleomycin or a similar drug inducing DSBs rather than stalled forks.

Figure 2C. My understanding is that the C-terminal truncation form of Pol32 used in this experiment doesn't have either RBM or PIP. It would be much more informative to use an RBM-only deletion mutant instead and present a similar experiment later in the paper when the RBM is identified. A separate point – if the authors suggest that the RBM might be phosphorylated to facilitate the repair of DNA damage (Figure 5), then it would make sense to do a pull-down from the cells with and without DNA damage induced.

Figure 3A. The BIR for T257A is very confusing. The allele acts as dominant negative and the authors suggest a somewhat speculative explanation in the discussion. My suggestion would be to look for another RBM in Pol-delta as the mutated one in Pol32 might be blocking the access to an alternative RBM elsewhere in Pol-delta. In Figure 2C, the pulldown of Pol31 brings some RPA even in the absence of the C-terminus of Pol32 (last lane in the blot) suggesting another interaction point between Pol-delta and RPA.

Figure 3C-D. It is very puzzling that the authors didn't consider Chk2 kinases (Rad53 and Dun1) as the candidates for the Pol32 phosphorylation, particularly in the light of their model in Figure 5 and the fact that BIR strongly depends on Rad9 involved in Rad53 activation. The genetics presented in panels C and D is not easy to interpret conclusively, mainly because the T257A mutation in the full length Pol32 context has a hard-to-explain phenotype (Figure 3A, see above). However, T257A behaves much more predictably in Pol32(1-333). This is why the genetics should be done in this background by comparing BIR in Pol32(1-333), Pol32(1-333) Rim11-deletion, Pol32(1-333)-T257A Rim11-deletion, Pol32(1-333) T257D Rim11-deletion. If the genetic experiments still hold Rim11 relevant (it will only work if Rim11 has Pol32 as the only target in BIR), a physical interaction should be addressed between Pol32 (or Pol-delta) and Rim11. As presented, the evidence for the regulation of Pol32 by Rim11 is not convincing and unless more data are added, this part should be excluded from the manuscript.

Figure 4C. The C-terminal deletion (1-953) truncates way too much to make a conclusion about the relevance of the newly identified RBM in Mph1 for BIR. The deleted sequence may contain another functional motif between aa 953 and 982. Even the Mph1 peptide used in the RPA binding assays (Figure 4D) starts from aa971, suggesting that there was no need for such an extended deletion. Therefore, a much shorter truncation (1-982) or a cluster of aa substitution in RBM should be used in the genetic experiments to address the importance of RBM in Mph1.

Version 1:

Reviewer comments:

Reviewer #1

(Remarks to the Author)

In the revision, the authors present additional data examining the contribution of the phospho-activated RPA-binding module (RBM) of Pol32 to break-induced replication (BIR). While these data support the idea that Pol32 can interact with RPA and the RBM contributes to BIR, it remains unclear whether phospho-activated RBM-RPA interaction plays a meaningful role in regulating BIR efficiency.

Although the pol32-RBM mutation diminishes the RPA–Pol32 interaction, it does not measurably affect BIR on its own. If the RPA-binding module (RBM) contributes specifically to BIR through RPA binding, then a mutation in the RBM would be expected to cause a corresponding specific defect. However, an effect of the RBM mutation on BIR efficiency is observed only in Pol32 variants lacking the C-terminal PCNA-binding and Pol alpha–binding domains. These findings suggest that the RBM may contribute only redundantly to Pol32-dependent BIR, potentially independently of its interaction with RPA. As a result, it remains unclear whether the Pol32–RPA interaction plays a critical and specific role in BIR.

Major points

1. The Pol32 RBM mutation significantly reduces the Pol31–RPA interaction (Fig. 2B); however, the RBM mutant does not exhibit a detectable defect in BIR (Fig. 2D). If the interaction with RPA plays a critical role in BIR function, a mutation in the RBM would be expected to cause a corresponding specific defect.
2. The phosphomimetic Pol32-aspartate mutant exhibits increased BIR efficiency compared with wild-type cells (Fig. 3F). However, the corresponding alanine substitution (phospho-null) mutant behaves similarly to wild type (Fig. 3F). These results indicate that phosphorylation at these sites is dispensable for BIR efficiency. It therefore remains possible that the aspartate substitutions enhance Pol32 function through effects unrelated to phosphorylation, rather than faithfully mimicking a phosphorylated state.

Other points:

1. The *in vitro* kinase assays presented in Fig. 3 do not identify the specific kinase(s) responsible for phosphorylation of these residues. The data are also consistent with the possibility that RBM phosphorylation is mediated by a single, as-yet-unidentified kinase that was not tested in these assays. Therefore, the current data do not necessarily provide evidence that multiple kinases phosphorylate the RBM domain of Pol32.
2. The Pol32 RBM mutation significantly reduces the Pol31–RPA interaction (Fig. 2B). The authors could address whether the phosphomimetic pol32-aspartate mutation enhances the RPA–Pol31 interaction, or the pol32-alanine mutation decreases this interaction.
3. Structural models suggest that the RBM is positioned in close proximity to PCNA and Pol31 when Pol δ is bound to the PCNA–DNA complex (Fig. 5A). However, Pol3 and Pol31 (shaded) or ssDNA-bound RPA are not fully incorporated into the current structural model, it remains uncertain whether the RBM of Pol32 is spatially accessible for interaction with RPA once Pol δ is fully engaged with PCNA on DNA (Fig. 5D). Incorporating improved or alternative structural models, along with an expanded discussion, would strengthen the proposed mechanism by which Pol32–RPA interactions contribute to BIR efficiency.

Technical points:

There are no appropriate negative controls in Fig. 2B and supporting raw data. In particular, the POL32 strain should contain TAP-tagged POL31 in addition to POL32-FLAG. In addition, there is no validation shown for the anti-Rfa1 antibody. At least, IP–Western and input (extract) Western samples should be run on the same gel and probed with the anti-Rfa1 antibody to demonstrate specificity and reliability.

RPA and Pol δ are localized to the lagging strand during DNA replication. Therefore, it remains possible that RPA and Pol δ are co-immunoprecipitated indirectly through ongoing lagging-strand DNA synthesis. To confirm a direct interaction between Rfa1 and Pol32, cell extracts used for co-IP experiments should be prepared from G1- or G2/M-arrested cells, where DNA replication is absent.

Reviewer #2

(Remarks to the Author)

The revised manuscript is improved, but a few points still need to be addressed.

(Page 7, lines 181–184) The manuscript states that Pol32 RBM* shows reduced BIR kinetics at 8 h (Fig. 2E), but the data do not clearly support this description. The difference does not appear significant, as the error bars overlap at the 6 h and/or 8 h time points.

More broadly, Pol32 RBM* alone does not show even a weak BIR phenotype in cells. Only when combined with elimination of the Pol32 PIP box does the RBM mutation appear to have an effect. Because the phenotypes supporting the importance of the Pol32–RPA interaction are weak, any statement in the abstract implying a major regulatory function for Pol32–RPA should be substantially toned down. The Discussion could also include some speculation as to why Pol32 RBM* phenotypes are observed only in mutants that also eliminate the PIP box.

Despite these remaining comments, this reviewer finds the revised work very interesting, as it demonstrates that multiple proteins important for BIR interact with RPA, and that these interactions can either stimulate or inhibit the process.

Minor comments:

Line 100: Should this be “~4-fold,” based on the data in the figure?

Question #3 regarding the levels of crossover in the mph1 mutant was not addressed.

Reviewer #3

(Remarks to the Author)

I appreciate all the effort the authors put into addressing the comments on the original submission and the resubmitted manuscript did improve by adding more clarity to the conclusions through providing additional data. Unfortunately, adding more experiments haven't changed the overall message. The newly identified RBM in Pol32 has very little if any relevance to the efficiency of BIR in the context of full length Pol32. This is supported by multiple pieces of data on mutating the RBM in different ways - deleting it completely (figure 1), replacing a couple critical residues with Rs (RBM* mutant, figure 2) or mutating the phosphorylatable threonines into As or Ds (Figure 3). I appreciate that all the in vitro analyses show that the phosphorylation of these threonines improves the Pol32-RPA interaction, yet there is no effect in vivo, unless the Pol32-PCNA interaction is lost. For this reason, I find the title and the second part of the statement in the abstract "Phosphorylation of the Pol32 RBM at Thr256 and Thr257 increases its affinity for Rfa1, while corresponding phospho-mimetic amino-acid substitutions promote BIR efficiency in vivo" misleading as they don't mention the context of broken Pol32-PCNA interaction needed to observe the effect relevant to the Pol32-RPA interaction. One can't deny that the RBM is real and that the reported phosphorylation might be critical for some process involving the Pol32-RPA interaction, but the authors' own experiments prove that BIR is not this process because the Pol32-PCNA interaction is sufficient for the fully efficient BIR.

Investigating the RBM in Pif1 deeper also ruled out the importance of its RBM phosphorylation for BIR in vivo, even though the phosphorylation of S70 and S72 has been previously reported and as the authors discovered, it does increase the RPA binding in vitro.

The newly identified RBM in Mph1 also looks real based on the modelling and in vivo analysis, even though the authors didn't analyse the relevance of the suggested phosphorylation on S989 and S991 by mutagenesis to address its possible role in either RPA binding or in BIR directly (in vivo).

The model presented in Figure 4F is interesting but not justified by the experiments presented in the paper. The phosphorylation of Pol32 RBM is supposed to play a critical molecular role in it regulating the BIR progression, yet the loss of this RBM has no effect on BIR efficiency in vivo. The model in Figure 5D is more appropriate for the manuscript and justified by the presented discoveries.

Nature Communications MS NCOMMS-25-05463

Jones, Appanah et al., "Break-induced replication is regulated by a phospho-activated RPA-binding module in Pol32"

Point-by-point response to reviewers

We thank the referees for their assessments and suggestions. We were pleased they feel "this is an interesting study that provides new mechanistic insights into BIR regulation" (**referee 2**) by the "identification of a novel RPA binding motif (RBM) in the Pol32 subunit of the DNA polymerase Delta" (**referee 3**), whose role "in regulating BIR remains a central question" (**referee 1**). Their valuable comments have formed the basis for a major experimental revision including 11 new main figure panels and 11 new supplementary figure panels. We feel our additional work has significantly strengthened the study and improved our model for phospho-activated Pol32-RPA interactions during BIR-mediated DNA double-strand break repair. Below, please find our point-by-point response to the comments provided, a description of additional experiments, and details of changes made to the figures and the revised MS text (highlighted green in the MS text file).

REVIEWER COMMENTS

Reviewer #1 (Remarks to the Author):

DNA double-strand breaks (DSBs), if not properly repaired, can lead to the loss of chromosomal integrity. DSBs can be repaired through non-homologous end joining (NHEJ), which re-ligates broken ends, or through homologous recombination (HR). HR repair occurs via two principal mechanisms: gene conversion (GC) and break-induced replication (BIR). BIR becomes necessary when only one end shares homology with a donor, such as at stalled or broken replication forks or at dysfunctional telomeres.

Numerous studies have established that Pol32 is not required for gene conversion but is essential for BIR. While DNA synthesis can initiate in the absence of Pol32, it does not proceed efficiently during BIR. How Pol32 regulates BIR remains a central question. In this study, the authors show that the interaction between Pol32 and RPA is critical for BIR. The data also suggest that Pol32 interacts with Mph1 and Pif1 via RPA, leading to a very interesting idea that RPA serves as a critical platform for the orchestration of the BIR machinery. The significance of the proposed RPA-Pol32 interaction is not fully clear because RBM deletion did not have significant impact on Pol32-mediated BIR (Fig. 1D). This finding suggests that other critical domains within Pol32 may be responsible for regulating the interactions between Pol32, RPA, Mph1 and Pif1 (Fig. 5).

Authors' response: We agree with **Referee 1** that the question of how Pol32 regulates BIR remains an important question in the field. In our study we discover a hitherto unknown RPA binding module (RBM) in Pol32 which mediates a novel physical interaction between Pol δ and RPA. What is particularly intriguing is that the Pol32 RBM provides the first example of a phospho-activated RBM whose affinity to the Rfa1 binding groove is increased more than 150-fold by phosphorylation at Thr256 and Thr257. While additional protein interactions between Pol32 and PCNA or Pol α contribute to BIR efficiency, it appears that phospho-activation is unique to the Pol32-

RPA interaction. We therefore propose that the significance of this new Pol32 RBM lies in its potential to act as dynamic regulator of BIR. Consistently, we demonstrate that phospho-mimetic amino acid substitutions T256D/T257D boost BIR efficiency *in vivo*. The corresponding phospho-null substitutions negatively impact BIR efficiency, which becomes apparent in the absence of the Pol32 PIP box (**new Fig. 3G**); moreover, new kinetic analyses of BIR show that mutations within the RBM which lower the affinity for RPA result in slower repair kinetics and a severe repair defect in the absence of the Pol32 PIP-box (**new Fig. 2E**).

Equally important, we define a new RBM in the BIR antagonist Mph1 and show that it interacts with the same binding groove in Rfa1 as those in Pol32 and Pif1. We demonstrate that the Pif1 and Mph1 RBMs bind RPA with high affinity in the non-phosphorylated state (**new Fig. 4B** and **Fig. 4D**), which can be explained by the presence of negatively charged amino acid residues at positions equivalent to the Thr256/257 double phosphorylation site in the Pol32 RBM.

These findings support a mechanism by which cells can manipulate the interplay of the core enzymes involved in BIR at RPA-bound D-loops, thereby allowing dynamic control over D-loop stability and BIR efficiency levels. This model does not require invoking additional, unidentified functional domains within Pol32 for consistency, and adds an important dimension to our understanding of how BIR outcomes are regulated. Beyond BIR, the phospho-regulation of the Pol32 RBM serves as a paradigm for the malleability of RPA interactions in the regulation of DNA metabolic processes, opening the door to future studies on phospho-regulated RPA interactions.

Specific Points:

1. In addition to the Pol32-RBM-Rpa1 NTD complex (Figures 2 and 4), Alphafold structural predictions of the entire PolD-RPA complex would assist in constructing a more comprehensive model (Figure 5).

Authors' response: We thank **Referee 1** for this suggestion and have now performed additional AlphaFold3 (Abramson J et al., Nature 630:493–500 (2024). doi: 10.1038/s41586-024-07487-w) analyses of the Pol32-RPA interaction in the context of the yeast Pol δ holocomplex engaged with PCNA and a primer-template junction to further inform the model presented in **revised Fig. 5**. Specifically, we have modelled full-length Pol32, *bis*-phosphorylated at Thr256 and Thr257 (please also refer to **point 6** below) as part of the Pol δ -PCNA-DNA complex and included the N-terminal domain of Rfa1 (Rfa1 NTD), which is attached to the remainder of the RPA trimer by a long linker. In brief, the modelling revealed several features not captured by an existing Pol δ -PCNA-DNA cryo-EM structure (PDB entry 7KC0; Zheng F et al, Proc Natl Acad Sci U S A 117:30344-30353 (2020). doi: 10.1073/pnas.20176371172020), which does not resolve the unstructured Pol32 C-terminal domain (CTD). Thus, the AlphaFold3 models consistently show multiple interactions of the Pol32 CTD within the Pol δ -PCNA-DNA-Rfa1-NTD assembly: (1) In all predicted models, an as-yet undescribed helix-turn-helix motif at Pol32 amino acid residues 164-218 (which we now refer to as 'Hook') binds a monomer-monomer junction of the PCNA trimer (*i.e.*, not to the interdomain connector loop (IDCL) which is the target of canonical PIP-box interactions). This interaction is reminiscent of a previously described Rev1-PCNA interaction (Sharma NM et al. (2011) J Biol Chem 286:33557-66. doi: 10.1074/jbc.M110.206680). In contrast to the Pol32 PIP-

box, the Pol32 'Hook' is not sufficient to support apparent PCNA interactions by yeast-two-hybrid assay or BIR efficiency. Nonetheless, this new feature is noteworthy as its similarity to Rev1 suggests that it provides an additional PCNA contact point for Pol δ . (2) In 5/5 models, the Pol32 PIP-box is engaged with one of three available IDCLs within the PCNA homotrimer, indicating sufficient Pol32 CTD flexibility to ensure that the Pol32 PIP-box can engage PCNA when Pol δ is bound at the D-loop primer junction. (3) The phosphorylated Pol32-RBM is consistently engaged with the Rfa1 NTD across models, and the flexibility and length of the linker between the Rfa1 NTD and subsequent RPA domains indicates that Pol32 can interact with RPA which is bound to the displaced ssDNA strand at a D-loop while Pol δ -PCNA sits at the primer junction during BIR.

These new insights provide an important refinement to our model (now presented as **revised Fig. 5D**), showing the potential of Pol32 to recruit and stabilize Pol δ at the primer-template junction DNA through two-pronged PCNA and RPA interactions, thereby promoting BIR. The above is described in detail in the revised version of the MS in a dedicated **new Results section** entitled "**A model for the actions of Pol32 in BIR**" (page 13, line 343-page 14, line 379) and illustrated in **new Fig. 5, panels A-C (and new Supplementary Fig. S9)**.

2. Does the assay (Figure 1D) fully exclude the possibility of GC between the two chromosomes? For instance, did the authors confirm the presence of two copies of LEU2 after DSB induction?

Authors' response: Yes, the data shown in **Fig. 1D** does exclude GC. The BIR assay does report on (Pol32-independent) GC, and it has been determined that the GC rate after HO induction in JRL-derived strains is ~2% (Lydeard JR et al., Nature 448:820-3 (2007). doi: 10.1038/nature06047). GC-dependent repair of the HO endonuclease-induced DSB on chromosome V in JRL-derived strains (JRL092 and YRL629-YRL648, and new strains YRL661-YRL666, YRL679, YRL680 in the revised MS) results in the retention of the *HPH* gene on chromosome V. In good agreement with GC rates reported by Jim Haber's lab, we routinely find 2-4% of DSB repair survivors in JRL-derived strains have retained *HPH* (which are consequently resistant to hygromycin B selection), regardless of any specific mutation we have tested. This information is now included in the **Methods section** of the revised version of the MS as detailed in our reply to **point 3** below.

3. The efficiency of DSB induction and cell survival following HO expression in each experiment should be shown. Curiously, the assay becomes more reproducible as cell survival decreases (Figures 1E, 2D, 3B).

Authors' response: The HO endonuclease is a yeast-endogenous enzyme and its efficiency in reliably generating a site-specific DSB across an entire yeast cell population if expressed from a *GAL* promoter is the basis for widely used HO-based BIR reporter assays. Thus, loss of Rad51, which is required for BIR and GC renders JRL-derived strains virtually inviable upon HO induction by galactose in the culture medium (Lydeard JR et al., Nature 448:820-3 (2007). doi: 10.1038/nature06047; Lydeard JR et al., Genes Dev. 24:1133-44. (2010) doi: 10.1101/gad.1922610). Inefficient cutting is not considered a contributing factor to the distribution of repair efficiency data obtained with this assay. To demonstrate the effectiveness of the *GAL*-induced HO

endonuclease in our hands, we have now included in the revised version of the MS a PCR-based validation of DSB formation (**new Supplementary Fig. S1A**), as well as of the *de novo* generation of a contiguous *CAN1* gene through BIR-dependent DSB repair in HO cut survivors (**new Supplementary Fig. S1B**), supplementing **Fig. 1E**. We routinely find that a PCR product spanning the *HOcs* locus can no longer be amplified when the HO endonuclease is induced on YPGAL medium, indicating efficient chromosome V breakage, which is in line with a survival rate after HO induction that tends towards 0% in strains lacking *POL32* (residual survival of 2-4% occurs by GC, generating hygromycin B-resistant colonies which are not included when reporting BIR efficiency).

To clarify, the revised MS text has been amended on **page 4, lines 92-98** to read: “*The impact of these mutations on BIR was determined using a reporter strain²³ for BIR efficiency at a DSB on chromosome V with the only available repair template located at an ectopic site on chromosome XI (Fig. 1D). The DSB is effectively introduced across the cell population upon galactose-induced expression of the HO endonuclease²³ (Supplementary Fig. S1A). BIR-mediated DSB repair can be genetically tracked as it restores a CAN1 marker at the cut site whilst leading to the loss of a strategically placed HPH marker (Fig. 1D and Supplementary Fig. S1B).*”

And additional information has been added to the **Methods section** entitled “**BIR efficiency measurements**” in the revised version of the MS on **page 18, lines 470-484**: “*To measure ectopic BIR (JRL092-derived strains), logarithmically growing cells cultured in YPRAFF medium were plated in appropriate dilutions onto YPD or YPGAL medium and grown into colonies. Cell viability as a measure of DSB repair efficiency after HO endonuclease induction was derived by dividing the number of colony-forming units (CFU) on YPGAL by that on YPD. For JRL-derived strains, HO cut efficiency was routinely tested by PCR, and a PCR product spanning the *HOcs* locus was no longer obtained after exposing cells to galactose for 1 h (for an example, see **Supplementary Fig. S1A**). Colonies on YPGAL were always replica plated onto YPD medium supplemented or not with 200 µg/ml hygromycin B. This showed that 2-4% of DSB repair events led to retention of the HPH resistance marker, in good agreement with previously determined²³ rates of BIR-independent DSB repair by gene conversion in JRL-derived strains. This fraction was subtracted from the value for DSB repair efficiency to calculate BIR efficiency for a given strain. In addition, BIR survivors were routinely validated using PCR on random colonies to show the presence of a new, contiguous *CAN1* marker at the former *HOcs* locus (see **Supplementary Fig. S1B**).*”

4. The explanation of the Pol32-CTD (Figure 2C) is missing. Pol32-CTD expression is significantly low (Fig. 2C) although any other C-terminal deletion mutants were expressed well (Fig. 1C). What region does the Pol32 CTD mutation contain?

Authors' response: The Pol32 Δ CTD mutant is nearly identical to Pol32_1-249. However, following the suggestion by **Referee 3**, we have replaced the Pol32 Δ CTD mutant in IP experiments (**former Fig. 2C**) with the more refined Pol32 Δ RBM mutant, which harbours an internal deletion of the newly identified RBM only rather than a deletion of the entire CTD (**new Fig. 2B** in the revised version of the MS). The Pol32 Δ RBM mutant shows much more similar protein levels to wild-type Pol32 in native protein extracts used for IP experiments as compared to Pol32 Δ CTD. Importantly, after IP, we observe

very similar amounts of recovered Pol δ holoenzyme for complexes containing full-length Pol32 or Pol32 Δ RBM, while the latter (like complexes containing Pol32 Δ CTD in the previous version of the MS) show a severe reduction of co-recovered RPA, validating more elegantly that the novel Pol δ -RPA protein-protein interaction is mediated by the Pol32 RBM. This is referred to in the revised version of the MS on **page 6, line 162-page 7, line 166**:

“However, an internal deletion of Pol32 aa 247 to 258 (hereafter referred to as Δ RBM) diminished Pol δ -RPA interactions (Fig. 2B). These data are therefore consistent with the newly identified acidic-hydrophobic motif within Pol32 functioning as a bona fide RBM and serve to confirm the novel Pol δ -RPA interaction.”

5. Ddc2, Mph1, and Pif1 interact with Rfa1 via the same interface (Figure 4), suggesting that these proteins compete with Pol32. Does the deletion of DDC2, MPH1, or PIF1 increase the Pol32-Rfa1 interaction (Figure 2C)? The authors should provide a better explanation of how the interplay between Pol32, MPH1, and PIF1 drives BIR. This would help clarify the role of Pol32 in DNA synthesis during BIR.

Authors' response: We have previously shown that Ddc2-RPA interactions (Yates LA et al., Proc Natl Acad Sci U S A 120:e2300150120 (2023) doi: 10.1073/pnas.2300150120.) support DNA damage checkpoint signalling upstream of Rad51-mediated strand invasion and do not expect an interplay between Ddc2 and Pol32 at D-loop primer junctions. However, we have performed the suggested experiment by deleting *MPH1* or *PIF1* in strains harbouring endogenously TAP-tagged Pol31 and analysing TAP-IPs for RPA recovery. While we detected slight increases in RPA recovery in the deletion strains (Rfa1/Pol31-TAP signal ratio for control = 1; 1.23 and 1.19 for the *pif1* Δ and *mph1* Δ strains, respectively), we do not think that these minor effects provide relevant information on the interplay of Pol32, Mph1, and Pif1. This is because RPA is estimated to be present in cells at 5-10 x higher concentrations than Pol32, Mph1, or Pif1 (<https://www.yeastgenome.org/>). It is therefore unlikely that the saturation of immunoprecipitated Pol δ complex with RPA would be significantly changed in *pif1* Δ and *mph1* Δ cells regardless of the ability of Mph1 and Pif1 to occupy the same protein binding site on Rfa1 as Pol32. As illustrated in the model in **revised Fig. 5D**, similar considerations apply to the *in vivo* situation at BIR intermediates with a multitude of RPA molecules bound to associated ssDNA regions. Here, RPA provides a platform for the engagement of Pol32, Mph1, and Pif1 through their respective RBMs which bind to the basic-hydrophobic groove of Rfa1 subunits, thereby increasing their local concentration and effecting BIR outcomes. How significant a direct competition for the Rfa1 basic-hydrophobic groove on the same Rfa1 molecule may be, remains an open question for future study; answering this question will likely require fully reconstituted BIR single-molecule studies, which goes beyond the scope of our current study. However, given that the 3'-DNA junction at a D-loop can only be engaged either by Pol δ (promoting further DNA synthesis) or Mph1 (unwinding and D-loop disassembly), it is conceptually straightforward that direct competition between Pol32 and Mph1 for RPA can be relevant locally and affect BIR efficiency (of note, Mph1 does not contain a PIP-box and can therefore not rely on PCNA for recruitment to the primer junction). We articulate this point on **page 15 (lines 405-416)** in the revised version of the MS: *“Mph1 targets extended D-loops^{24,49} and dissociates the nascent strand⁴⁷ to restrict BIR efficiency^{48,60}. We find that these actions of Mph1 depend on a newly defined RBM at the*

Mph1 C-terminus. The Mph1 RBM binds RPA more avidly than the non-phosphorylated Pol32 RBM, but with similar affinity to the phosphorylated form. It is therefore tempting to speculate that Pol32 RBM phosphorylation may help Pol δ to locally resist the actions of Mph1 at the D-loop primer junction. Thus, Pol32 RBM phosphorylation may promote BIR by stabilizing Pol δ at the primer junction and mitigating D-loop collapse. As a consequence, the risk of chromosome aberrations^{5,6} arising from ectopic recombination upon premature release of the nascent strand and invasion at a non-contiguous site with sequence homology⁶¹⁻⁶⁴ could be reduced. Conversely, Pol32 RBM de-phosphorylation may favour D-loop disassembly, allowing cells to terminate BIR and limit potentially mutagenic⁸ BIR-dependent DNA synthesis.”

Like Pol32, Pif1 is a BIR agonist, and unlike Mph1, Pif1 does not require competitive access to the primer junction engaged by Pol δ . To drive D-loop progression, Pif1 is thought to contact the displaced ssDNA strand as shown in our model in **revised Fig. 5D**. As suggested by the Makovets lab, this is likely to be aided by Pif1 RBM-RPA interactions (Kotenko O & Makovets S. EMBO Rep 25:1734-1751 (2024) doi:10.1038/s44319-024-00114-9). Consistently, abrogation of the Pif1 RBM brings down BIR efficiency (**new Fig. 4C** in the revised version of the MS). Coordination of Pif1 and PCNA-bound Pol δ during BIR is mediated by PCNA, to which Pif1 binds with its own PIP-box (Buzovetsky O et al. Cell Rep 21:1707-1714 (2017) doi: 10.1016/j.celrep.2017.10.079). We suggest that “*our model is consistent with the independent contributions which we have observed the Pol32 RBM and PIP-box make to BIR efficiency. We propose that multi-point interactions of Pol32 and Pif1 (refs. 20,59) with PCNA and RPA-bound recombination intermediates ensure the proper recruitment and retention of Pol δ and the accessory D-loop helicase Pif1 at BIR intermediates, allowing optimal BIR reactions to take place (Fig. 5D).*” (**page 14, lines 374-379** in the revised version of the MS).

6. The authors show that Rim11 is required for BIR. Is Rim11 specifically necessary for BIR but not GC? The phosphorylation of Thr-257 by Rim11 needs to be validated through both *in vitro* and *in vivo* experiments. Because Pol32-T257D mutations were tested in all the genetic assays (Fig. 3 and Fig. 4B), T257D peptides should be also evaluated in Fig. 4A.

Authors' response: We thank **Referee 1** for suggesting further investigation of Pol32 RBM phosphorylation by *in vitro* kinase assays. We have now tested Rim11 and were surprised to find that despite the GSK3 family kinase consensus sequence within the Pol32 RBM, Pol32 is a poor phosphorylation substrate (**new Supplementary Fig. S4**). This suggested that (1) Rim11 impacts BIR through a phosphorylation target other than Pol32, and that (2) Pol32 RBM phosphorylation *in vivo* (refs. 40-44 in the revised version of the MS) is mediated by a set of implicated kinases which remain to be tested *in vitro*.

Regarding point (1), we considered the characterized phosphorylation targets of Rim11, which include the meiosis-specific protein Ime1 (Rubin-Bejerano I et al., Mol Cell Biol 24:6967-79 (2004). doi: 10.1128/MCB.24.16.6967-6979.2004), the cytosolic Pah1 phosphatidate phosphatase (Khondker S et al., J Biol Chem 298:102221 (2022). doi: 10.1016/j.jbc.2022.102221), and the Ume6 subunit of the Ume6-Rpd3 histone deacetylase complex (Malathi K et al., Mol Cell Biol 17:7230-6 (1997). doi: 10.1128/MCB.17.12.7230). Interestingly, Pif1 is a deacetylation target of Ume6-Rpd3,

providing a potential link to BIR (Ononye OE et al., J Biol Chem 295:15482-15497 (2020). doi: 10.1074/jbc.RA120.015164). Indeed, we found that deletion of *UME6* reduces BIR efficiency similarly to a deletion of *RIM11* (**Figure A herein**), providing a potential explanation as to why Rim11 loss negatively affects BIR efficiency. While the role of Rim11 in BIR warrants further investigation, for the purposes of the current study the new data clearly indicate that Rim11 is not the kinase responsible for the phosphorylation of the Pol32 RBM.

Fig. A: *Left*, Rim11 has been implicated in the phosphorylation of a limited number of potential (light blue) and verified (dark blue/yellow) targets. Among them, Ume6-Rpd3 targets Pif1 whose acetylation status impacts Pif1 activity. This connects the Rim11 kinase with Pif1-dependent BIR, providing a potential explanation for the observation that loss of *RIM11* results in a BIR efficiency defect. *Right*, BIR efficiency measurements for the indicated JRL-derived strains revealing a BIR defect upon loss of *UME6* that is similar to the one observed upon *RIM11* loss.

(2) To address which kinase might directly phosphorylate Pol32 RBM, we first considered the kinases that have been linked to Pol32 phosphorylation at residues within the RBM *in vivo*. Phospho-proteomic studies consistently show that the Pol32 RBM can be phosphorylated at Thr257, and also at Thr256 and Ser264 by multiple potential kinases (refs. 40-44 in the revised version of the MS): (i) Dbf4-dependent kinase (DDK) Cdc7, (ii) cyclin-dependent kinase (CDK) Cdc28, and (iii) Polo-like kinase (PLK) Cdc5. Given the acidic-hydrophobic amino acid sequence surrounding Thr257, we also considered the acidophilic casein kinase CKII, although there is currently no available *in vivo* evidence to support CKII-dependent Pol32 RBM phosphorylation. We used Pol32 RBM fragments spanning amino acid residues 248 to 264 fused to thioredoxin in *in vitro* kinase assays with TAP-tagged versions of Cdc7, Cdc5, Cdc28, and CKII. Affinity-captured Cdc7, Cdc5, and CKII were able to phosphorylate the RBM fragment, but not Cdc28 (nor Rim11) (**new Supplementary Fig. 4A, B**). Phosphorylation levels dropped when Cdc7, Cdc5, and CKII were offered a version of the RBM fragment harbouring amino acid substitutions T256A/T257A (**new Supplementary Fig. 4B**). Next, we performed mass spectrometry phosphorylation analysis of the wild-type Pol32 RBM peptide to reveal that while all affinity-purified kinases exerted detectable kinase activity on the thioredoxin fusion, the Pol32 RBM was only phosphorylated by Cdc7, Cdc5, and CKII at Thr256, Thr257, and Ser264 (**new Fig. 3A, B and new Supplementary Fig. 4C**). The strongest activity was exhibited by the acidophilic kinases CKII and Cdc7. Interestingly, Cdc7 has been previously reported to promote BIR efficiency (Lyderad JR et al., Genes Dev 24:1133-44 (2010). doi: 10.1101/gad.1922610). These extensive new data confirm the DDK and PLK-dependent phosphorylation of the Pol32 RBM observed *in vivo* and suggest that Cdc5, Cdc7, and possibly additional acidophilic kinases such as CKII, contribute to establishing the ‘phospho-code’ on Pol32. The experiments are

described in an entirely new section in **Results** (please see section “**The Pol32 RBM can be phosphorylated by multiple protein kinases.**”; page 7, line 190-page 8, line 219) and in the **Discussion** on page 16 (line 417-page 17, line 436), as follows: “Regulation of the Pol32 RBM phosphorylation status may involve multiple kinases. Ddc2 RBM Ser 10 and Ser11 phosphorylation is driven by the constitutively active acidophilic kinase CKII and its regulation is likely to involve the actions of phosphatases.³⁷ Consistent with the sequence similarity between the Ddc2 and Pol32 RBMs, we find that CKII can also phosphorylate the Pol32 RBM in vitro and may therefore contribute to the phospho-regulation of Pol32. Several other kinases have been implicated in Pol32 RBM phosphorylation in vivo.⁴⁰⁻⁴⁴ We confirmed the ability of Cdc5 (PLK) and Cdc7 (DDK) to directly phosphorylate the Pol32 RBM in vitro. Phospho-proteomics suggest that Cdc7 phosphorylates serine and threonine residues with Asp/Glu/pSer/pThr at the plus 1 position⁴⁴, and bis-phosphorylation at Thr256/257 within Pol32 conforms with this preference. Interestingly, Cdc7 showed the most robust activity towards the Pol32 RBM in vitro and has previously been reported⁴⁶ to promote BIR in vivo. DDK is active from the start of S-phase through to mitosis, while PLK is activated at the G2/M transition. Both kinases act to align cellular processes with cell cycle stage and are known to co-regulate recombination-associated reactions, for example by coordinating the resolution of recombination intermediates with mitotic entry⁶⁵. Their activity towards Pol32 would suggest that the Pol32 RBM becomes increasingly activated for high-affinity RPA binding as cells progress through S phase, which could promote effective BIR at broken replication forks or exogenous DNA breaks in S/G2 phase. However, the details of Pol32 phospho-status regulation, which may involve additional kinases as well as phosphatases, remain to be elucidated in future studies.”

(3) To validate that Pol32 RBM phospho-mimetic mutations mediate an increase in affinity for the Rif1 NTD as expected from our experiments with the phosphorylated Pol32 RBM peptides, we have now performed an FP experiment with a Pol32 T256D T257D peptide. The threonine to aspartate substitutions significantly increase the affinity of the Pol32 RBM for Rfa1 NTD as compared to the non-phosphorylated wild-type peptide, improving the K_D from 22.9 μ M to 2.93 μ M. We agree with **Referee 1** that this is a valuable control for the link between increased Pol32 RBM-RPA affinity and enhanced BIR efficiency. This new data is presented in new **Supplementary Fig. S6C** and referred to in the revised version of the MS on **page 10, line 270-page 11, line 283**: “Addressing the impact of enhanced Rfa1 NTD interactions mediated by Pol32 pRBM on BIR, we confirmed that phospho-mimetic amino changes Thr256Asp and Thr257Asp within the RBM increase the affinity for Rfa1 (**Supplementary Fig. 6C**). We then generated versions of the ectopic BIR reporter strain expressing Pol32 T256D T257D, which was present at concentrations comparable to wild-type Pol32 (**Fig. 3E**). Importantly, Pol32 T256D T257D boosted BIR efficiency as compared to the control strain, identifying a first Pol32 mutant that improves BIR (**Fig. 3F**). Like Pol32 Δ RBM and -RBM*, an endogenously expressed Pol32 T256A T257A phospho-null mutant was not associated with a BIR efficiency defect (**Fig. 3F**). However, as in the case of Pol32_1-333 RBM*, a strain expressing Pol32_1-333 T256A T257A showed a significant drop in BIR efficiency, suggesting that phosphorylation at Thr256/257 is required for BIR efficiency in the absence of the Pol32 PIP-box. Consistently, T256D/T257D continued to enhance

BIR efficiency in the context of a Pol32 PIP-box truncation mutant (Fig. 3G). These data are in line with the notion that the phosphorylation of the Pol32 RBM in vivo⁴⁰⁻⁴⁴ can be deployed by cells to modulate BIR efficiency.”

7. Pif1 appears to interact with the N-terminus of Rfa1 similar to Ddc2 and Mph1 (Fig. 4E). Pif1 peptides would be tested in Fig. 4A.

Authors’ response: We have now tested a Pif1 RBM peptide with Rfa1 NTD in a fluorescence resonance assay as shown in former Fig. 4A. The measured K_D is 0.32 μ M, showing that the Pif1 RBM binds Rfa1 with high affinity more akin to the phosphorylated Pol32 RBM and much tighter than the non-phosphorylated form of the Pol32 RBM (**new Fig. 4B** in the revised version of the MS). This is consistent with our AlphaFold modelling showing that Trp75 of the non-phosphorylated Pif1 RBM reaches into a well-defined hydrophobic ‘side-pocket’ on the surface of Rfa1 NTD, mirroring an interaction that makes important contributions to the strength of RBM-Rfa1 NTD interactions for various other RPA client proteins (Wu Y *et al.*, *Elife* 12:e81639 (2023) doi: 10.7554/eLife.81639). Additional AlphaFold modelling shows that the ‘side-pocket’ interaction is not significantly altered by Pif1 RBM phosphorylation (**new Supplementary Fig. S7E**). Consistently, phosphorylation of Ser70 and Ser72 within the Pif1 RBM results in only a minor change in affinity for the Rfa1 NTD with an apparent K_D of 0.15 μ M (**new Supplementary Fig. S8A** in the revised version of the MS). In contrast, the Pol32 RBM requires phosphorylation to facilitate the conformational change that enables an equivalent ‘side-pocket’ interaction mediated by Leu262 for high-affinity interactions. The relevant revisions read (**page 11, line 290-page12, line 310; page 13, lines 336-341**):

“While the Pif1 RBM has been defined as spanning aa 60 to 74 (ref. 20), Trp75 within Pif1 aligns well with a similar hydrophobic residue, Phe993, at the end of the putative RBM sequence in Mph1 (**Fig. 4A**). Interestingly, both Pif1 and Mph1 RBMs contain pre-existing negative charges (Asp71 and Asp990, respectively) at the position of the Thr256/257 double phosphorylation site within Pol32, suggesting that both RBMs might behave more similarly to the phosphorylated rather than the non-phosphorylated Pol32 RBM. Indeed, AlphaFold modelling predicts that the Pif1 and Mph1 RBMs bind Rfa1 with the same polarity as Pol32, placing three hydrophobic residues consistent with the ϕ -D/E-x- ϕ - ϕ motif into the Rfa1 groove: Pif1 Leu66, -Leu68 and -Leu69; Mph1 Leu983, -Ile986 and -Leu987 (**Fig. 4A and Supplementary Fig. S7A, B**). Pif1 Trp75 and Mph1 Phe993 bind directly into the hydrophobic ‘side-pocket’ that is occupied by Leu262 in the phosphorylated forms of the Pol32 RBM (**Supplementary Fig. S7C, D**). Here, the additional phosphorylation of the Pif1 or Mph1 RBMs is not predicted to significantly impact the position of Pif1 Trp75 and Mph1 Phe993, which suggests that both proteins adopt the high-affinity Rfa1 binding conformation independently of phosphorylation within their RBMs (**Supplementary Fig. S7E, F**). Consistently, we determined a K_D of 0.32 μ M (95% CI, 0.25-0.52 μ M) for a fluorescein-labelled peptide derived from Pif1 aa 59 to 79 with the Rfa1 NTD by FP (**Fig. 4B**). Thus, Pif1 tightly binds RPA and with similar affinity to Pol32 RBM phosphorylated at Thr256/257. Phosphorylation of Pif1 at Ser70 and Ser72 flanking Asp71 had a moderate impact on Rfa1 NTD binding affinity; improving the K_D to 0.15 μ M (95% CI, 0.13-0.16 μ M; **Supplementary Fig. S8A**).”

“Overall, these findings show RBM-mediated RPA interactions of each BIR replisome component and BIR antagonist Mph1 impact homologous recombination-dependent DNA synthesis, highlighting a critical scaffolding function of RPA bound at recombination intermediates. Pif1 and Mph1 intrinsically bind RPA with high affinity, whereas phosphorylation is required to activate high-affinity RPA binding by the Pol32 RBM, providing a mechanism for the dynamic regulation of BIR efficiency (Fig. 4F).”

Reviewer #2 (Remarks to the Author):

This work presents interesting data suggesting a dynamic role of RPA in BIR, where RPA interactions with Pol δ , Mph1, and Pif1 dictate the stability and extension of the recombination D-loop during BIR. The authors identify residues within the Pol32 component of Pol δ that are responsible for interaction with RPA. They also investigate the role of serine/threonine phosphorylation within the same region of Pol32 where it interacts with RPA. While phosphorylation sites were previously described, their role remains unknown. The authors show that Pol32 phosphorylation stimulates the Pol δ -RPA interaction and enhances BIR efficiency.

One of the strongest aspects of this work is the identification of Rim11 kinase as a likely regulator of Pol δ phosphorylation, with rim11 Δ mutants exhibiting the same low BIR phenotype as Pol32 phosphorylation mutants. However, somewhat unexpectedly, the pol32 mutant that abolishes interaction with RPA does not, by itself, show any deficiency in BIR. Finally, the authors also identify the C-terminal part of Mph1 as interacting with RPA and regulating BIR, likely by influencing D-loop stability. They propose an interesting model in which Rim11-mediated phosphorylation shifts Rfa1 interaction from Mph1 (D-loop unwinding) toward Pol δ (D-loop extension), thereby stimulating BIR.

This is an interesting study that provides new mechanistic insights into BIR regulation.

Authors' response: We thank **Referee 2** for their positive assessment of our work. We are happy to report that the puzzling discrepancy between the Pol32 T257A phosphorylation mutant and Pol32 Δ RBM has been resolved in the revised version of the MS. Prompted by the referees' comments, we have expanded our work on Pol32 RBM phosphorylation (explained in more detail in response to **Referee 1 (point 6)** above). Briefly, we have performed *in vitro* kinase assays and found that, despite the presence of a predicted consensus phosphorylation site, Pol32 RBM is a poor substrate for Rim11 (**new Supplementary Fig. S4**; Rim11 likely impacts BIR through phosphorylation of the Ume6-Rpd3 histone deacetylase, please see **Fig. A** above). Focussing instead on the kinases that have been linked to *bis*-phosphorylation of Pol32 residues Thr256 and Thr257 (refs. 40-44 in the revised MS), we now demonstrate that Dbf4-dependent kinase (DDK) Cdc7 and Polo-like kinase (PLK) Cdc5 are capable of directly phosphorylating the Pol32 RBM (**new Fig. 3A, B**). The new data is described in the **Results** section entitled “**The Pol32 RBM can be phosphorylated by multiple protein kinases.**” (please see **page 7, line 190-page 8, line 219**). Importantly, the Pol32 T256A T257A *bis*-phosphorylation site mutant is not associated with a BIR efficiency defect, consistent with the phenotype of Pol32 Δ RBM. However, upon deletion of the Pol32 PIP-

box (Pol32_1-333 T256A T257A), cells show a significant drop in BIR efficiency as compared to cells expressing Pol32_1-333, indicating that phosphorylation at Thr256/257 *in vivo* allows cells to modulate BIR efficiency. Of note, Cdc7, which has previously been implicated in BIR efficiency by the Haber lab (Lydeard JR et al., *Genes Dev.* 24:1133-44. (2010) doi: 10.1101/gad.1922610), showed the strongest activity towards the Pol32 RBM *in vitro*. This might indicate that Pol32 high-affinity RBM binding is increasingly activated as cells transit S phase, potentially promoting effective BIR at broken replication forks or exogenous DNA breaks in S/G2 phase (please see **Discussion, page 16, line 417-page 17, line 436** in the revised MS).

Concerns

The analysis of phenotypes is relatively weak, as it relies on a single BIR assay that is highly inefficient, even in wild-type cells (only ~20% complete BIR). The differences between wild-type and mutants are within a ±5% range for some mutants. Additionally, several interesting predictions can be drawn from the proposed model that should be tested *in vivo*. A few additional experiments would significantly strengthen this work:

1. Test BIR using another assay. The allelic BIR assay, which is commonly used, would be ideal. Decreased D-loop extension often leads to specific phenotypes, such as half-crossovers, which are expected in *rim11Δ* mutants.

Authors' response: While the JRL-based system (developed by Jim Haber and co-workers: Lydeard JR et al., *Nature* 448:820-3 (2007) doi: 10.1038/nature06047), which we use throughout this MS, is robust and consistently identifies significant differences in BIR efficiency in the tested mutants, we agree that independent validation of the key findings presented in **Fig. 1** with an alternative assay can strengthen our data further. As suggested, we have now obtained the AM1003-based system from Prof Grzegorz Ira and have constructed additional reporter strains to measure allelic BIR as described by the Malkova lab (Deem *et al.*, *Genetics* 179:1845-60. (2008) doi: 10.1534/genetics.108.087940). AM1003 is a haploid background with a second copy of chromosome III (*MATa* and truncated distal to the *HOcs* locus), which increases the efficiency of BIR repair off the uncleavable *MATα-inc* allele on the full-length copy of chromosome III (please see **new Supplementary Fig. S1E**; technical details added to the **Methods** section titled "**BIR efficiency measurements**", **page 18** of the revised MS). The two copies of chromosome III can be genetically distinguished because *HML* on the truncated copy is replaced by *ADE1* while *HML* and *HMR* on the full-length chromosome III are replaced by *ADE3* and *HPH* (which provides resistance to hygromycin), respectively. Deem and co-workers (and other authors since) have shown that approx. 80% of cells repair the DSB ensuing from galactose-induced HO endonuclease expression by BIR (generating *ADE+ leu-* cells) in this disomic system, while less than 10% repair by gene conversion (GC; *ADE+ LEU+* survivors). In the presence of mutations which negatively affect BIR efficiency, chromosome loss and half-crossovers occur; this results in retention of the full-length copy of chromosome III only (the colonies are *ade- leu-* and red because loss of *ADE1* on the truncated chromosome III disrupts the adenine biosynthesis such that red pigment accumulates) or fusions between the *ADE1*-containing fragment from the HO-endonuclease cleaved chromosome III and the distal part of the full-length copy (*ADE3* is lost; white *ade- leu-* colonies), respectively. In good agreement with Deem and co-workers, we observe that

chromosome loss events and half-crossovers are exceedingly rare in the wild-type, but their frequency is significantly increased in *pol32* Δ -mutant cells. Importantly, and confirming our truncational analysis of *POL32* in the ectopic JRL-based BIR assay system, we find that chromosome loss and half-crossovers occur much more frequency compared to wild-type upon removal of PIP box (*pol32_1-333*) and that further truncation (*pol32_1-249*) – thus removing the PIP and RBM modules – results in a phenotype very similar to that observed for *pol32* Δ cells (**new Supplementary Fig. S1F**). These results, which corroborate our conclusion that both the PIP-box and RBM of Pol32 promote D-loop DNA synthesis and thereby BIR efficiency, are described in the revised version of the MS on **page 4, line 106-page 5, line 115**:

*“Next, we analysed DSB repair in an independent tester strain which is disomic for chromosome III (with one copy harbouring an HO-cut site and the other serving as repair template)³⁶. In this system, DSB repair is highly efficient, and cells overwhelmingly use BIR with a small contribution of DSB repair by gene conversion. In contrast, *pol32* Δ cells show reduced BIR efficiency, a strong increase in chromosome loss events, and the formation of half-crossovers indicative of failed DSB repair and aberrant processing of BIR intermediates, respectively.³⁶ We observed similar levels of Pol32-independent BIR, chromosome loss, and half-crossovers in cells harbouring a *POL32* deletion or a truncation of Pol32 at residue 249 (**Supplementary Fig. S1E and F**). Thus, *Pol32_1-249* results in a Pol32-null phenotype for ectopic and allelic interhomolog BIR.”*

2. Measure BIR kinetics in an efficient BIR assay. It would be informative to examine BIR kinetics in mutants such as *pol32_RBM*, *rim111* Δ , and *pol32* phosphorylation mutants. In particular, *pol32_RBM* might exhibit a BIR delay.

Authors' response: We agree with **Referee 2** that measurements of BIR kinetics provide an excellent possibility to uncover BIR defects not easily detected in an end-point DNA double-strand break repair assay. To assess BIR kinetics, we used the PCR-based assay established by the Haber lab (Lydeard JR et al., Nature 448:820-3 (2007). doi: 10.1038/nature06047; described in a new Methods section on **page 19** titled “**BIR kinetics measurements**” in the revised version of the MS). Using primers for the amplification of the nascent *CAN1* gene that is newly formed through BIR after HO-cut induction, we detected a marked delay in repair kinetics in the *pol32_1-133 RBM** mutant, where inactivation of the RBM is combined with a deletion of the PIP box, compared to a *pol32_1-133* mutant. This is consistent with the loss in BIR efficiency seen for the *pol32_1-133 RBM** mutant. Interestingly, and as anticipated by this referee, we could even detect evidence of perturbed repair kinetics in *pol32_RBM** cells, which corroborates that the Pol32 RBM contributes to promoting more stable D-loop DNA synthesis, even in the context of full-length Pol32. The data obtained with the RBM* mutants is presented in **new Fig. 2E** and described in the text on **page 7, lines 181-188**: “A PCR-based BIR assay²³ for the detection of new DNA synthesis initiated at the centromere-proximal side of the HO cut site (see **Supplementary Fig. S1B**) upon invasion of the repair template showed that Pol32 RBM* was associated with a slight but reproducible delay in the formation of BIR repair intermediates in the first 8 h after DSB induction. While the levels of BIR product formation mediated by Pol32 RBM* were eventually indistinguishable from wild-type, *Pol32_1-333 RBM** was associated with a prolonged BIR DNA synthesis defect that grew more pronounced between 8 and 20 h

after DSB induction (**Fig 2E**). Together, these data indicate that Pol32 RBM-RPA interactions promote BIR.”

3. Examine crossover frequency in the Mph1 C-terminal mutant. The mph1 Δ C (1-954) mutant shows reduced interaction with RPA and stimulates BIR, likely due to loss of its effect on D-loop unwinding. A key prediction is that this mutant should exhibit a higher level of crossover outcomes, which should be tested.

Authors' response: To test whether Mph1 relies on its newly defined RBM for the destabilization of D-loops, we made use of the afore-mentioned (please see response to **point 2** above) PCR-based kinetic assay. We found that deletion of the Mph1 RBM resulted in faster BIR repair kinetics, *i.e.*, an accelerated *de novo* formation of CAN1 repair intermediates through D-loop DNA synthesis after HO-cut induction. This is consistent with the expectation that Mph1-RPA interactions support Mph1-dependent D-loop unwinding, thus facilitating Mph1's role as a BIR antagonist. This data is presented in **new Supplementary Fig. S8D** in the revised version of the MS, and is referred to in the text on **page 12, lines 323-327**:

“Cells expressing the truncated Mph1_1-954 from the endogenous locus exhibited significantly elevated BIR efficiency, phenocopying a complete deletion of MPH1 (**Fig. 4E**). Consistently, BIR intermediate formation was higher in mph1_1-954 cells compared to MPH1 wild-type control cells as measured by PCR²³ 6 h after DSB induction (**Supplementary Fig S8D**).”

4. Test the genetic interaction between Rim11 and Mph1. If Rim11-mediated phosphorylation of Pol32 enhances Pol δ -RPA interaction and reduces the negative impact of Mph1-RPA on BIR, then rim11 Δ mph1-954 double mutants should not exhibit any defects in BIR. Testing BIR efficiency in this double mutant would directly test the authors' model.

Authors' response: This specific experiment is superseded by our new findings that DDK and PLK, which have been implicated *in vivo* (and not Rim11) mediate the direct phosphorylation of the Pol32 RBM *in vitro*. The essential and pleiotropic nature of Cdc7 and Cdc5 preclude the suggested experiment. However, we have extended our analyses of the Pif1 and Mph1 RBMs. Sequence alignment and AlphaFold modelling shows that the Pif1 and Mph1 RBMs contain pre-existing negative charges (Asp71 and Asp990, respectively) equivalent to Pol32 Thr256/257 *bis*-phosphorylation. This allows Pif1 and Mph1 to interact with the Rfa1 NTD similarly to the Pol32 pRBM by engaging a small hydrophobic pocket on the surface of Rfa1 independently of phosphorylation (please see **new Supplementary Fig. S7E, F**). As an example in point, we demonstrate that Pif1 RBM phosphorylation has minor effects on RPA affinity and no significant impact on BIR efficiency (**new Supplementary Fig. S8 A, B**). Because Mph1 requires its own RBM and competitive access to the primer junction engaged by Pol δ during BIR to mediate D-loop disassembly, these results strongly suggest that Pol δ with a phosphorylated Pol32 subunit may be a greater obstacle to Mph1 than the non-phosphorylated form. We refer to these considerations on **page 15** (from **line 405**) in the revised version of the MS:

“Mph1 targets extended D-loops^{24,49} and dissociates the nascent strand⁴⁷ to restrict BIR efficiency^{48,60}. We find that these actions of Mph1 depend on a newly defined RBM at the Mph1 C-terminus. The Mph1 RBM binds RPA more avidly than the non-phosphorylated

Pol32 RBM, but with similar affinity to the phosphorylated form. It is therefore tempting to speculate that Pol32 RBM phosphorylation may help Pol δ to locally resist the actions of Mph1 at the D-loop primer junction. Thus, Pol32 RBM phosphorylation may promote BIR by stabilizing Pol δ at the primer junction and mitigating D-loop collapse. (...) Conversely, Pol32 RBM de-phosphorylation may favour D-loop disassembly (...)."

Minor Comments:

Rim11 localization and function: Rim11 is typically localized in the cytoplasm. Is it known to phosphorylate any other nuclear proteins? Is there evidence that Rim11 directly phosphorylates Pol32?

Authors' response: While there is precedent for the Rim11-dependent phosphorylation of nuclear factors (Rubin-Bejerano I et al., Mol Cell Biol 24:6967-79 (2004). doi: 10.1128/MCB.24.16.6967-6979.2004; Malathi K et al., Mol Cell Biol 17:7230-6 (1997). doi: 10.1128/MCB.17.12.7230), we have now tested the ability of Rim11 to directly phosphorylate Pol32. As detailed above, our new experiments show that under conditions where Rim11 is active, the kinase fails to phosphorylate the Pol32 RBM *in vitro* (**new Supplementary Fig. S4**). Instead, we identify the Pol32 RBM as a direct target of DDK (Cdc7) and PLK (Cdc5) (**new Fig. 3A, B**). Cdc7 has previously been implicated in BIR efficiency (Lydeard JR et al., Genes Dev. 24:1133-44. (2010) doi: 10.1101/gad.1922610).

Clarification in Lines 39-41: The work first demonstrated that BIR is limited in the repair of replication fork breakage—PMID: 26273056.

Authors' response: In the revised version of the MS, we have amended the text to highlight the specific findings reported in PMID: 26273056 (this is ref. 2 in the original and the revised version of the MS) to read (on **page 2, lines 41-44**):

"(...) This can be achieved through active D-loop processing by structure-specific nucleases and the arrival of a converging DNA replication fork.^{2,9} Alternatively, cells may suppress BIR to favour the generation of a two-ended DSB and subsequent repair by error-free homologous recombination.¹⁰⁻¹²"

Reviewer #3 (Remarks to the Author):

In the manuscript by Jones and co-workers, the authors report the identification of a novel RPA binding motif (RBM) in the Pol32 subunit of the DNA polymerase Delta using the budding yeast as a model organism. Normally, the Pol32 recruitment to DNA is controlled through its interaction with PCNA via a PCNA-interacting peptide (PIP) located at the C-terminus of Pol32. The importance of the identified RBM *in vivo*, based on DNA repair by BIR, could only be revealed if the Pol32-PCNA interaction is eliminated from the system by the Pol32 C-terminal deletion involving the PIP, as shown by the experiments in figure 2D. The RPA-Pol31-Pol32 *in vivo* interaction is detected by protein CoIPs (Figure 2C).

The *in vitro* binding experiments and Alpha-fold modelling support the idea that phosphorylation of some S/T residues within the RBM motif strengthens the RPA-Pol32 interaction. The phosphor-site mutations have an expected effect on BIR in the

background where the Pol32-PCNA interaction is lost, but behave in a confusing way in full length Pol32. The authors then claim that Rim11 is the kinase responsible for the phosphorylation of the RBM in Pol32.

Finally, a structural parallel is drawn to the RBMs from Pif1 and Mph1, two other proteins known to be involved in BIR and carrying RBMs.

I have two concerns about this manuscript which I would like to discuss separately:

1. the suitability for publishing it in Nature Communications;
2. the experimental part.

As presented, the focus of the manuscript is on identifying a novel RBM in Pol32. Based on the fact that this is just another RBM added to the several previously reported in different organisms, including some of those in budding yeast, the novelty of this manuscript doesn't advance the field enough to grant a publication in NC, particularly since the biological significance of this RBM in Pol32 is unclear and its potential regulation through phosphorylation is not studied well enough. Based on the experiments in this manuscript, the RBM plays a secondary role to the PIP in the Pol32 recruitment in BIR: with the Pol32 PIP being functional, the RBM has no effect on the efficiency of BIR. The RBM may play a more important role in some other genome stability aspect independent of BIR, but it is not presented in this work and, therefore, it remains unclear how important it is.

With this said, I could see how the focus of the manuscript could be shifted to a more of a big-picture question, to stress the novelty of identifying extended RBMs where phosphorylation of the residues adjacent to the hydrophobic cores plays a role in genome stability via stimulating the recruitment of the repair proteins to DNA. This would be a major revision and require a lot of additional experiments, such as analysing multiple RBMs, the ones in Pif1 and Mph1 at the very least, demonstrating the phosphorylation dependence on the DNA damage response (or other trigger) *in vivo*, different forms of peptide binding to RPA similar to those for Pol32 in figure 4A, plus addressing the physiological role of RBM phosphorylation in BIR through analysing the phosphosite mutants in the genetic BIR assays. These experiments would then justify the model proposed by the authors in Figure 5 and might make the story conceptually novel enough for NC to consider.

As is, there is very little experimental data to support this model: the phosphorylation of RBM in Pol32 is not important for BIR *in vivo*; the phosphorylation of RBMs in Pif1 and Mph1 hasn't been addressed experimentally. Also, it is not known what triggers the RBM phosphorylation – cell cycle, DNA damage signalling, etc. Therefore, the model in figure 5 is based on computer modelling and imagination, but not supported by sufficient experimental data.

Authors' response: While we discover a novel RBM that contributes to the function of Pol32 in BIR, we agree with **referee 3** that important significance and novelty arises from the first description of a mechanism by which RBMs can be activated by phosphorylation for high-affinity PRA binding. In our major experimental revision, we have included a number of experiments to solidify our model presented in **revised Fig. 5**. Specifically, we provide additional evidence for the biological significance of the

Pol32 RBM through BIR kinetic measurements (**new Fig. 2E**), identify the Pol32 RBM as a direct phosphorylation target of Dbf4-dependent kinase (DDK) Cdc7 and Polo-like kinase (PLK) Cdc5 using *in vitro* kinase assays and analysis by mass spectrometry (**new Fig. 3A, B** and **Supplementary Fig. S4**), validate the importance of Pol32 RBM phospho-residue-Rfa1 NTD interactions by mutational analysis of Rfa1 (**new Supplementary Fig. S6A, B**), and investigate the RBMs of Pif1 and Mph1 in more detail (**new Fig. 4 A-C, E**, **new Supplementary Fig. S7E, F** and **new Supplementary Fig. S8A, B, D**).

We used a PCR-based assay (Lydeard JR et al., Nature 448:820-3 (2007). doi: 10.1038/nature06047) to measure BIR kinetics by following the formation of BIR intermediates at the cleaved *HOcs* locus. As expected, there was a marked delay in BIR repair in *pol32_1-133 RBM** cells. Interestingly, we could also detect evidence of delayed repair kinetics in *pol32_RBM** cells, which supports the role of Pol32 RBM in promoting D-loop DNA synthesis, even in the context of full-length Pol32. This data is presented in **new Fig. 2E** and described in the revised version of the MS on **page 7, lines 181-188**:

*“A PCR-based BIR assay²³ for the detection of new DNA synthesis initiated at the centromere-proximal side of the HO cut site (see **Supplementary Fig. S1B**) upon invasion of the repair template showed that Pol32 RBM* was associated with a slight but reproducible delay in the formation of BIR repair intermediates in the first 8 h after DSB induction. While the levels of BIR product formation mediated by Pol32 RBM* were eventually indistinguishable from wild-type, Pol32_1-333 RBM* was associated with a prolonged BIR DNA synthesis defect that grew more pronounced between 8 and 20 h after DSB induction (**Fig 2E**). Together, these data indicate that Pol32 RBM-RPA interactions promote BIR.”*

As explained in more detail in our response to **specific point 4** below, we have turned to *in vitro* phosphorylation assays to investigate the kinase(s) responsible for Pol32 RBM phosphorylation. Despite the presence of a predicted consensus phosphorylation site, Pol32 RBM turned out to be a poor substrate for Rim11 (**new Supplementary Fig. S4**; the impact of Rim11 on BIR may be explained by its role in phosphorylating the Ume6-Rpd3 histone deacetylase, which in turn modifies and alters the activity of Pif1 (Ononye OE et al., J Biol Chem 295:15482-15497 (2020). doi: 10.1074/jbc.RA120.015164); please see our response to **Referee1, point 6**; and **Fig. A** above). We next focussed on the kinases linked to phosphorylation of Pol32 *in vivo* (phospho-proteomic studies in refs. 40-44 in the revised version of the MS). We were able to demonstrate that Cdc7 and Cdc5 are capable of directly phosphorylating the Pol32 RBM (**new Fig. 3A, B**). This suggests a cell-cycle stimulus where phosphorylation-dependent high-affinity Pol32 RBM binding to the Rfa1 NTD is increasingly activated as cells progress through S phase, which would promote BIR at broken replication forks in S/G2 phase (please see **Discussion, page 16, line 417-page 17, line 436** in the revised MS).

Following **Referee 3**'s suggestion, we have further investigated the Pol32, Pif1 and Mph1 RBMs (note that we have validated that the C-terminus of Mph1 contains a *bona fide* RBM in the revised version of the MS, please see our response to **specific point 5** below). Testing a mutant Rfa1 NTD protein with Arg91Ala and Lys95Ala substitutions

and the pThr257 or pThr256 pThr257 (*bis-*)phosphorylated Pol32 RBM, we observed that the effect of phosphorylation in promoting high-affinity binding was neutralized (new **Supplementary Fig. 6A, B**). This provides important experimental validation of the significance of the modelled pRBM-Rfa1 contacts with Rfa1 Arg91 and -Lys95 RBM in the pRBM kinked conformation that allows Leu262 to project into a well-characterized hydrophobic ‘side-pocket’, thus contributing to interaction strength with RPA (please see **revised Fig. 3D**). We further demonstrate that both, the Pif1 and Mph1 RBMs, do not require phosphorylation for high-affinity Rfa1 NTD binding. In contrast to the Pol32 RBM, phosphorylation of the Pif1 RBM peptide (pSer70 and pSer72) had a mild impact on Rfa1 NTD binding affinity (improving the already low K_D determined by FP from 0.32 μ M for the non-phosphorylated form to 0.15 μ M; **new Supplementary Fig. S8A**). Concomitantly, cells expressing Pif1 S70D S72D did not show increased BIR efficiency (**new Supplementary Fig. S8A**). Sequence alignment and AlphaFold modelling shows that the Pif1 and Mph1 RBMs naturally contain phospho-mimetic residues (Asp71 and Asp990, respectively) at positions equivalent to Pol32 pThr256 and pThr257. This implies that the Pif1 and Mph1 RBMs might interact with the Rfa1 NTD in a high-affinity configuration similar the phosphorylated Pol32 RBM, explaining their phosphorylation-independent avid binding of the Rfa1 NTD. In support of this view, we provide modelling showing that the phosphorylation is not required for the Pif1 and Mph1 RBMs to engage the small hydrophobic ‘side-pocket’ on the surface of Rfa1 via Trp75 and Phe993, respectively (**new Supplementary Fig. S7E, F**). Since overall BIR efficiency is determined by the interplay of agonists (such as Pol32) and antagonists (Mph1), we suggest that the dynamic regulation of one effector (Pol32) is sufficient to fine-tune BIR outcomes (**page 15**, from **line 405** of the revised version of the MS):

“Mph1 targets extended D-loops^{24,49} and dissociates the nascent strand⁴⁷ to restrict BIR efficiency^{48,60}. We find that these actions of Mph1 depend on a newly defined RBM at the Mph1 C-terminus. The Mph1 RBM binds RPA more avidly than the non-phosphorylated Pol32 RBM, but with similar affinity to the phosphorylated form. It is therefore tempting to speculate that Pol32 RBM phosphorylation may help Pol δ to locally resist the actions of Mph1 at the D-loop primer junction. Thus, Pol32 RBM phosphorylation may promote BIR by stabilizing Pol δ at the primer junction and mitigating D-loop collapse. (...) Conversely, Pol32 RBM de-phosphorylation may favour D-loop disassembly (...).”

Our extended work on RBMs in the key components of the BIR machinery thus shows they exist in two flavours, either supporting phospho-regulated or constitutive high-affinity Rfa1 NTD binding. This raises the question whether the mechanism of phospho-activated pRBM-Rfa1 NTD that we describe for Pol32 captures a general phenomenon. Based on sequence analysis, we predict it does, since RBMs with potential phosphorylation sites (serines and threonines) or phospho-mimetic residues at positions equivalent to Thr256 and Thr257 in Pol32 exist. Excitingly, whilst our manuscript was under review, a complementary study was published in Nature Communications: Noto A et al., Nat Commun 16:997 (2025) doi:10.1038/s41467-025-55958-z) that showed CK2-mediated phosphorylation of an acidic domain in WRN promoted RPA association in human cells, and this facilitated the recovery of stalled DNA replication forks. The authors observed that phosphorylation of a tandem RBM within the WRN acidic domain (at Ser440 in the first RBM and Ser467 in the second) enabled RPA association in cells, while this was not observed with a non-

phosphorylatable mutant. There is no structural basis for phospho-regulated RBMs in WRN, but this can now be rationalized when considering our study on the phospho-regulated RBM in Pol32. Indeed, when we predict the corresponding phosphorylated RBM in WRN (pSer440) with RPA70-NTD, the structural basis for the interaction is analogous to Pol32 (**Fig. B, panels a and b** herein). Similarly to the Pol32 RBM (**Fig. B, panel c**), the unmodified WRN-RBM adopts a linear α -helix, which is the configuration found in the X-ray crystal structure of this RBM associated with RPA70-NTD (PDB 7XUT) (**Fig. B, panel d**). Upon phosphorylation, the continuous α -helix of the Pol32-RBM is ‘broken’ and engages in new interactions with the Rfa1 NTD groove (as described above, pThr257 or pThr256 in the *bis*-phosphorylated RBM engage with the side chains of Rfa1 Arg91 and -Lys95 and the kinked conformation allows Leu262-hydrophobic ‘side-pocket’ interactions; please see **revised Fig. 3D**). AlphaFold structural predictions of a phosphorylated WRN-RBM with RPA70-NTD suggest a highly related conformational change (**Fig. B, panels c and d**). This is all the more remarkable because Rfa1 and RPA70 only share 20% sequence identity.

Fig. B. Schematic molecular cartoons for AlphaFold3 models of **(a)** Pol32 and **(b)** WRN phospho-RBMs in complex with the Rfa1 or RPA70 NTDs, respectively. The side chains for selected amino acids are shown in stick representation; note how Pol32 Leu262 and WRN Ile438 project into hydrophobic ‘side-pockets’ on the surface of Rfa1/RPA70. **(c, d)** Illustrate the analogous conformational changes in the Pol32 and WRN RBMs upon phosphorylation at Thr256/Thr257 and Ser440, respectively: The linear Pol32 and WRN (PDB:7XUT) RBM α -helix is broken and the kinked pRBM configuration bound to Rfa1/RPA70 allows the positioning of Pol32 Leu262 and WRN Ile438 for higher-affinity, ‘side-pocket’ interactions.

Therefore, our molecular dissection of Pol32 RBM phosphorylation, which enhances RPA-association 150-fold, represents an exemplar case for a phospho-regulated RBM, providing a molecular mechanism which we anticipate will be much more widely applicable. Current evidence supports the view that RBM phosphorylation is deployed by cells to enhance RPA interaction and regulate DNA repair outcomes *in vivo* in at least two pathways: BIR (our study) and fork recovery (please see Noto A et al., Nat Commun 16:997 (2025) doi:10.1038/s41467-025-55958-z.) Thus, while the focus of our

work is the mechanism of DSB repair by BIR, shedding important new light on the enigmatic ability of Pol32 to modulate BIR repair, it provides a paradigm for a hitherto undescribed control mechanism for DNA metabolic processes involving RPA-ssDNA intermediates. This provides the basis for exciting future work in experimentally cataloguing the effects of RBM phosphorylation across RPA client proteins and identifying the biological processes regulated by phospho-dependent Rfa1/RPA70 NTD binding from yeast to human. This broad relevance is now referred to in the **Discussion** in the revised version of the MS (**page 17, lines 441-449**):

“More generally, the phospho-activated RBM within Pol32 provides a paradigm for dynamic control of protein association with the Rfa1 NTD basic-hydrophobic groove and hydrophobic pocket to fine-tune DNA repair. Consistent with this notion, phospho-dependent interactions of the Werner syndrome helicase (WRN) with RPA have recently been reported to promote DNA replication fork recovery.⁶⁶ With RPA being involved in virtually all DNA transactions during replication, repair, and recombination involving ssDNA, and a growing number of identified Rfa1/RPA70 client proteins¹⁴, pRBM-Rfa1/RPA70 interactions could be broadly relevant in genome maintenance.”

The experimental part:

(1) Figure 1F. The authors use HU to assay different POL32 alleles. The rest of the paper is focused on BIR, i.e. DSB repair involving activation of the G2/M checkpoint rather than an intra-S phase checkpoint triggered by HU. It might be informative to add a spot test using Phleomycin or a similar drug inducing DSBs rather than stalled forks.

Authors’ response: The Pol32 N-terminal domain is required for Pol δ complex stability and loss of *POL32* results in cellular sensitivity to a wide range of replication-stress inducing agents including HU, MMS, and 4NQO (e.g., Karras GI & Jentsch S, Cell 141:255-67 (2010) doi: 10.1016/j.cell.2010.02.028). This can be overcome by artificially stabilizing the Pol δ complex in *pol32 Δ* cells; the BIR defect of *pol32 Δ* cells, however, cannot be overcome in this way (Shimada K et al., Life Sci Alliance 4:e202101138 (2021) doi: 10.26508/lsa.202101138). This indicates that BIR requires features of Pol32 outside the N-terminal domain, which we show include the newly identified C-terminal RBM. In **Fig. 1F**, we demonstrate that truncation of the C-terminal features of Pol32 which support BIR (PIP-box, DPIM and RBM; please see **Fig. 1E**) do not sensitize cells to replication stress sensitivity, indicating that Pol δ complex formation is intact and Pol δ remains DNA replication competent. We have now performed a spot assay on phleomycin-containing medium and the results are similar to those obtained with HU; *pol32 Δ* cells show hypersensitivity while Pol32 C-terminal truncation mutations are

Fig. C: Dilution spot assays with the indicated strains assessing phleomycin sensitivity analogously to the experiment with HU shown in **Fig. 1F** in the revised version of the MS.

associated with very little to no apparent sensitivity. This is consistent with DNA damaging drugs such as phleomycin causing replication stress and results showing that *POL32* is not required for DSB repair by gene conversion (Lydeard JR et al., Nature 448:820-3 (2007). doi: 10.1038/nature06047), the preferred pathway of repair at exogenously induced DSBs. The phleomycin data (**Fig. C** herein) is available, but we suggest inclusion in the revised MS would not add significantly to the HU sensitivity assay demonstrating that Pol32 C-terminal deletions affecting BIR do not result in *pol32Δ*-linked replication stress sensitivity.

(2) Figure 2C. My understanding is that the C-terminal truncation form of Pol32 used in this experiment doesn't have either RBM or PIP. It would be much more informative to use an RBM-only deletion mutant instead and present a similar experiment later in the paper when the RBM is identified. A separate point – if the authors suggest that the RBM might be phosphorylated to facilitate the repair of DNA damage (Figure 5), then it would make sense to do a pull-down from the cells with and without DNA damage induced.

Authors' response: We fully concur with **Referee 3** that deploying an RBM-only internal deletion to assess the role of Pol32 in the newly found Pol δ complex-RPA interaction is more elegant than a Pol32 C-terminal tail deletion mutant. In our revisions, we have constructed the relevant strains and repeated the experiment shown in former **Fig. 2C**. Using cells harbouring TAP-tagged Pol31 and FLAG-tagged Pol32 Δ RBM, we demonstrate that an internal deletion of the Pol32 RBM strongly diminishes the recovery of RPA observed with wild-type Pol32 to nearly background levels (**new Fig. 2B** in the revised version of the MS). This confirms our previous conclusion that Pol δ -RPA interactions depend on the newly identified RBM within Pol32 (referred to on **page 6, line 162-page 7, line 166**):

*“However, an internal deletion of Pol32 aa 247 to 258 (hereafter referred to as Δ RBM) diminished Pol δ -RPA interactions (**Fig. 2B**). These data are therefore consistent with the newly identified acidic-hydrophobic motif within Pol32 functioning as a bona fide RBM and serve to confirm the novel Pol δ -RPA interaction.”*

BIR-dependent D-loop DNA synthesis does not require checkpoint activity (Malkova A et al., Mol Cell Biol 25:933-44 (2005) doi: 10.1128/MCB.25.3.933-944.2005; Kimble MT et al., Mol Cell 85:61-77.e6 (2025) doi: 10.1016/j.molcel.2024.10.032.), and there is currently no *in vivo* evidence to suggest that Pol32 phosphorylation is controlled by the DNA damage checkpoint. Instead, we followed the evidence provided by phospho-proteomic studies and now provide new data showing that DDK and PLK can directly phosphorylate the Pol32 RBM (please see our response to **specific point 4** below), suggesting that the alternative stimulus suggested by **Referee 3**, cell-cycle progression, is a driver of Pol32 RBM phosphorylation.

(3) Figure 3A. The BIR for T257A is very confusing. The allele acts as dominant negative and the authors suggest a somewhat speculative explanation in the discussion. My suggestion would be to look for another RBM in Pol-delta as the mutated one in Pol32 might be blocking the access to an alternative RBM elsewhere in Pol-delta. In Figure 2C, the pulldown of Pol31 brings some RPA even in the absence of the C-terminus of Pol32 (last lane in the blot) suggesting another interaction point between Pol-delta and RPA.

Authors' response: We thank **Referee 3** for their thoughtful remarks here, and in the

subsequent **specific point 4**, regarding the T257A BIR phenotype. We also have considered the possibility of an intramolecular dominant negative effect associated with this mutation. Within Pol32, we cannot identify another RBM (**Supplementary Fig. S2**) and our extensive new AlphaFold 3 modelling with the entire Pol δ -PCNA-DNA-Rfa1-NTD assembly (please see **Referee 1, point 1**) exclusively identifies the Pol32 RBM-Rfa1 interaction for Pol δ , which is consistent with a significant drop in RPA recovery by Pol δ pull-down to background levels when the Pol32 RBM is internally deleted (**new Fig. 2B** in the revised version of the MS). However, the comments have prompted us to fully revisit the question of Pol32 phosphorylation and the behaviour of phospho-null mutants. It is true that a negative effect of phospho-null mutations within the Pol32 RBM are unexpected because the Pol32 Δ RBM mutant demonstrates that RBM phosphorylation is not required for BIR efficiency as measured in the genetic end-point assay (**Fig. 2D**). Furthermore, our assessment of kinases implicated by phospho-proteomic studies in Pol32 RBM phosphorylation *in vivo* during this revision (please see our reply to **specific point 4** below) shows that multiple kinases can directly phosphorylate the RBM *in vitro* at Thr256, Thr257, and Ser261. Concentrating solely on Thr257 phosphorylation (which was the first identified and only annotated RBM phospho-site on SGD at the start of this project and conforms to a potential GSK3 kinase target site) has therefore proven too narrow an approach. Strikingly, the puzzling dominant negative effect of the T257A mutation was no longer observed when we combined T257A with T256A (please see **new Fig. 3F** in the revised version of the MS). This indicates a spurious effect associated with Pol32 T257A, potentially as a result of alterations to the normal Pol32 phospho-regulation. In the revised version of the MS, we have therefore eliminated the T257A mutant. Our new experiments are now aligned with the *in vivo* evidence showing phosphorylation of Pol32 being predominantly at Thr256 and Thr257 (see also **specific point 4** below) and resolve the previously puzzling discrepancy between the Pol32 Δ RBM and Pol32 T257A mutants. The data is referred to in the revised version of the MS on **page 10, line 270-page 11, line 283**:

*“Addressing the impact of enhanced Rfa1 NTD interactions mediated by Pol32 pRBM on BIR, we confirmed that phospho-mimetic amino changes Thr256Asp and Thr257Asp within the RBM increase the affinity for Rfa1 (**Supplementary Fig. 6C**). We then generated versions of the ectopic BIR reporter strain expressing Pol32 T256D T257D, which was present at concentrations comparable to wild-type Pol32 (**Fig. 3E**). Importantly, Pol32 T256D T257D boosted BIR efficiency as compared to the control strain, identifying a first Pol32 mutant that improves BIR (**Fig. 3F**). Like Pol32 Δ RBM and -RBM*, an endogenously expressed Pol32 T256A T257A phospho-null mutant was not associated with a BIR efficiency defect (**Fig. 3F**). However, as in the case of Pol32_1-333 RBM*, a strain expressing Pol32_1-333 T256A T257A showed a significant drop in BIR efficiency, suggesting that phosphorylation at Thr256/257 is required for BIR efficiency in the absence of the Pol32 PIP-box. Consistently, T256D/T257D continued to enhance BIR efficiency in the context of a Pol32 PIP-box truncation mutant (**Fig. 3G**). These data are consistent with the notion that the phosphorylation of the Pol32 RBM *in vivo*⁴⁰⁻⁴⁴ can be deployed by cells to modulate BIR efficiency.”*

(4) Figure 3C-D. It is very puzzling that the authors didn't consider Chk2 kinases (Rad53 and Dun1) as the candidates for the Pol32 phosphorylation, particularly in the light of

their model in Figure 5 and the fact that BIR strongly depends on Rad9 involved in Rad53 activation.

Authors' response: The Haber lab has previously demonstrated that Rad9 promotes BIR-dependent survivors after an induced HO-cut. However, the execution of BIR repair by D-loop DNA synthesis mediated by Pol δ is not dependent on Rad9 or checkpoint activation. Thus, BIR is restored in *rad9* Δ cells upon nocodazole arrest and occurs with normal kinetics. It is therefore the Rad9-dependent establishment of cell-cycle arrest – providing time for BIR to occur – not the action of DNA damage response kinases on the BIR machinery that explains the requirement for Rad9 for BIR efficiency (Malkova A et al., Mol Cell Biol 25:933-44 (2005) doi: 10.1128/MCB.25.3.933-944.2005). The mechanistic independence of BIR from checkpoint kinases is consistent with recent data from Lorraine Symington's lab showing that replication-coupled DSBs are repaired by BIR without activation of the checkpoint (Kimble MT et al., Mol Cell 85:61-77.e6 (2025) doi: 10.1016/j.molcel.2024.10.032.). In our renewed efforts to identify *bona fide* Pol32-phosphorylating kinases we have therefore not concentrated on checkpoint kinases but all kinases for which data exists in the literature implicating them in the phosphorylation of Pol32 RBM residues *in vivo*, include Cdc7 (DDK), Cdc5 (PLK), and Cyclin-dependent kinase (CDK) Cdc28 (please see below for details).

(4 cont'd) The genetics presented in [Fig. 3] panels C and D is not easy to interpret conclusively, mainly because the T257A mutation in the full length Pol32 context has a hard-to-explain phenotype (Figure 3A, see above). However, T257A behaves much more predictably in Pol32(1-333). This is why the genetics should be done in this background by comparing BIR in Pol32(1-333), Pol32(1-333) Rim11-deletion, Pol32(1-333)-T257A Rim11-deletion, Pol32(1-333) T257D Rim11-deletion. If the genetic experiments still hold Rim11 relevant (it will only work if Rim11 has Pol32 as the only target in BIR), a physical interaction should be addressed between Pol32 (or Pol-delta) and Rim11. As presented, the evidence for the regulation of Pol32 by Rim11 is not convincing and unless more data are added, this part should be excluded from the manuscript.

Authors' response: Following on from the reply to **specific point 3** above detailing that the previously puzzling reduction in BIR efficiency observed for the Pol32 T257A mutant is not seen for a Pol32 T257A T256A double mutant and thus not a consequence of loss of phosphorylation at these sites but a spurious dominant negative effect, we have extended our analysis of potential kinases mediating Pol32 RBM phosphorylation. We can now rule out Rim11 as the kinase responsible for Pol32 Thr257 phosphorylation and have taken the lead from *in vivo* data (refs. 40-44 in the revised version of the MS) demonstrating phosphorylation of the Pol32 RBM to identify the Pol32 RBM as a direct phosphorylation target of Cdc7 and Cdc5.

Prompted by **Referee 1**, we have taken an *in vitro* kinase assay approach. This showed that Rim11 is not capable of phosphorylating the Pol32 RBM despite the presence of a predicted GSK3 family kinase consensus sequence (**new Supplementary Fig. S4**). However, deletion of *UME6*, a known substrate of Rim11 (Malathi K et al., Mol Cell Biol 17:7230-6 (1997). doi: 10.1128/MCB.17.12.7230) and regulator of Pif1 (Ononye OE et al., J Biol Chem 295:15482-15497 (2020). doi: 10.1074/jbc.RA120.015164) phenocopies the BIR defect of *rim11* Δ cells (please see **Fig. A** above).

To identify *bona fide* Pol32 RBM kinases, we now considered the kinases for which evidence exists that they promote Pol32 phosphorylation at residues within the RBM *in vivo*. The Pol32 RBM contains three potential phosphorylation sites residues Thr256, Thr257, and Ser 261. Phospho-proteomics have shown that the Pol32 RBM is predominantly modified at both threonine residues *in vivo* (ref. 40-44 in the revised version of the MS). These studies have implicated multiple kinases including Cdc7, Cdc5, and Cdc28. We also considered the acidophilic casein kinase II (CKII) given the acidic-hydrophobic nature of the RBM amino acid composition. TAP-tagged versions of these kinases were affinity-captured and tested for activity on Pol32 RBM fragments (aa 248 to 264) fused to thioredoxin. Cdc7, Cdc5, and CKII were able to phosphorylate the Pol32 RBM fragment while Cdc28 (or affinity captured cyclins Clb1, 3, or 5) and Rim11 showed no activity on Pol32 (**new Supplementary Fig. 4A, B**). Phosphorylation levels dropped when Cdc7, Cdc5, and CKII were tested on a version of the RBM fragment with phospho-null mutations T256A/T257A (**new Supplementary Fig. 4B**). To map phosphorylation events, we performed mass spectrometry, revealing that while all affinity-purified kinases exerted detectable kinase activity on the thioredoxin fusion, the Pol32 RBM was only phosphorylated by Cdc7, Cdc5, and CKII (**new Fig. 3A, B and new Supplementary Fig. 4C**). As expected, the acidophilic kinases showed the most robust activity on the Pol32 RBM, and Cdc7 has been reported to be critical for BIR efficiency (Lyderad JR et al., *Genes Dev* 24:1133-44 (2010). doi: 10.1101/gad.1922610). The existing *in vivo* data (refs.40-44), combined with our new *in vitro* data, suggest Cdc5, Cdc7, and possibly additional acidophilic kinases such as CKII, contribute to laying down the appropriate ‘phospho-code’ on Pol32. Our experiments are described in new **Results** section entitled “**The Pol32 RBM can be phosphorylated by multiple protein kinases.**” (starting on **page 7** of the revised version of the MS). We consider the findings in the **Discussion** on **page 16, line 417-page 17, line 436**:

“Regulation of the Pol32 RBM phosphorylation status may involve multiple kinases. Ddc2 RBM Ser 10 and Ser11 phosphorylation is driven by the constitutively active acidophilic kinase CKII and its regulation is likely to involve the actions of phosphatases.³⁷ Consistent with the sequence similarity between the Ddc2 and Pol32 RBMs, we find that CKII can also phosphorylate the Pol32 RBM in vitro and may therefore contribute to the phospho-regulation of Pol32. Several other kinases have been implicated in Pol32 RBM phosphorylation in vivo.⁴⁰⁻⁴⁴ We confirmed the ability of Cdc5 (PLK) and Cdc7 (DDK) to directly phosphorylate the Pol32 RBM in vitro. Phospho-proteomics suggest that Cdc7 phosphorylates Ser and Thr residues with Asp/Glu/pSer/pThr at the plus 1 position⁴⁴, and bis-phosphorylation at Thr256/257 within Pol32 conforms with this preference. Interestingly, Cdc7 showed the most robust activity towards the Pol32 RBM in vitro and has previously been reported⁴⁶ to promote BIR in vivo. DDK is active from the start of S-phase through to mitosis, while PLK is activated at the G2/M transition. Both kinases act to align cellular processes with cell cycle stage and are known to co-regulate recombination-associated reactions, for example by coordinating the resolution of recombination intermediates with mitotic entry⁶⁵. Their activity towards Pol32 would suggest that the Pol32 RBM becomes increasingly activated for high-affinity RPA binding as cells progress through S phase, which could promote effective BIR at broken replication forks in S/G2 phase. However, the details of Pol32 phospho-status regulation, which may involve additional kinases as well as phosphatases, remain to be elucidated in future studies.”

We appreciate the insistence of **Referee 3** on this point, which has resulted in much more clarity and alignment with existing data on the point of Pol32 RBM phosphorylation. We feel that the identification of *bona fide* direct modifier kinases of the Pol32 RBM is an important addition to the revised MS.

(5) Figure 4C. The C-terminal deletion (1-953) truncates way too much to make a conclusion about the relevance of the newly identified RBM in Mph1 for BIR. The deleted sequence may contain another functional motif between aa 953 and 982. Even the Mph1 peptide used in the RPA binding assays (Figure 4D) starts from aa971, suggesting that there was no need for such an extended deletion. Therefore, a much shorter truncation (1-982) or a cluster of aa substitution in RBM should be used in the genetic experiments to address the importance of RBM in Mph1.

Authors' response: In the revised version of MS we validate the *mph1_1-954* mutant by comparison with a new mutant which harbours a single amino acid change: Phe993Ala. Our AlphaFold modelling shows that Mph1 Phe993 binds into the well-defined 'side-pocket' on Rfa1-NTD and equivalent hydrophobic residue-side-pocket interactions strongly enhance the affinity of many RPA clients for Rfa1 (Wu Y *et al.*, Elife 12:e81639 (2023) doi: 10.7554/eLife.81639). Consistent with our interpretation that *mph1_1-954* cells exhibit enhanced BIR efficiency due to disruption of the newly defined Mph1 C-terminal RBM, *mph1-F933A* cells exhibit a comparable increase in BIR efficiency. We conclude that classic RBM-mediated RPA-binding is required for the antagonistic role of Mph1 in BIR. This experiment is referred to in the revised version of the MS on **page 13, lines 330-335:**

"(...), we mutated Mph1 Phe993 to alanine and found that this single amino acid change results in significantly elevated BIR efficiency, phenocopying Mph1_1-954 (Fig. 4E). We conclude that the high-affinity RPA binding mode which relies on RBM contacts within the Rfa1 NTD basic-hydrophobic groove and the hydrophobic 'side-pocket' (ref. 14) is critical for Mph1 to engage and disassemble D-loop intermediates during BIR."

NCOMMS-25-05463A

“Break-induced replication is enhanced by a phospho-activated RPA-binding module in Pol32“

Point-by-point Response to Referees

Reviewer #1 (Remarks to the Author):

In the revision, the authors present additional data examining the contribution of the phospho-activated RPA-binding module (RBM) of Pol32 to break-induced replication (BIR). While these data support the idea that Pol32 can interact with RPA and the RBM contributes to BIR, it remains unclear whether phospho-activated RBM-RPA interaction plays a meaningful role in regulating BIR efficiency.

Although the pol32-RBM mutation diminishes the RPA–Pol32 interaction, it does not measurably affect BIR on its own. If the RPA-binding module (RBM) contributes specifically to BIR through RPA binding, then a mutation in the RBM would be expected to cause a corresponding specific defect. However, an effect of the RBM mutation on BIR efficiency is observed only in Pol32 variants lacking the C-terminal PCNA-binding and Pol alpha–binding domains. These findings suggest that the RBM may contribute only redundantly to Pol32-dependent BIR, potentially independently of its interaction with RPA. As a result, it remains unclear whether the Pol32–RPA interaction plays a critical and specific role in BIR.

Major points

1. The Pol32 RBM mutation significantly reduces the Pol31-RPA interaction (Fig. 2B); however, the RBM mutant does not exhibit a detectable defect in BIR (Fig. 2D). If the interaction with RPA plays a critical role in BIR function, a mutation in the RBM would be expected to cause a corresponding specific defect.

Authors’ response: Our data presented in **Fig. 1E**, **Fig. 2D**, **Fig. 2E**, and **Supplementary Fig. S1F** demonstrate the BIR efficiency defect associated with loss of the Pol32 PIP-box is severely exacerbated upon disruption of the novel Pol32 RBM. This shows that in absence of the Pol32 PIP-box, BIR efficiency is dependent on the Pol32 RBM and thus that Pol32-RPA interactions facilitate successful D-loop DNA synthesis across the 30 kb or 100kb-long chromosomal regions that need to be replicated for a viable BIR outcome in the respective tester strains. A similar effect is observed if we mutate the phosphorylation sites Thr256 and Thr257 within the Pol32 RBM to alanine, which results in significant drop in BIR efficiency in *pol32_1-133_T256A T257A* cells as compared to *pol32_1-133* cells (please see **Fig. 3G**). Thus, the availability of phosphorylation sites at Pol32 Thr256 and Thr257, which are phosphorylated *in vivo* (refs. 40-44 in our MS), result in enhanced D-loop DNA synthesis. Consistently, phospho-mimicking amino acid changes Pol32 T256D and T257D result in significantly enhanced BIR efficiency in *pol32_1-133_T256D T257D* cells as compared to *pol32_1-133* cells (**Fig. 3G**). Importantly, full-length Pol32 T256D T257D-expressing cells (i.e., in the presence of the PIP-box) exhibit BIR efficiency levels that are significantly above wild-type levels (**Fig. 3F**), phenocopying cells in which the novel RBM in BIR antagonist Mph1 is disrupted (**Fig. 4E**). We also show that phosphorylation of the Pol32 RBM increases the affinity to

RPA more than 160-fold, which we rationalize by structural modelling showing that phosphorylation enables a high-affinity binding conformation of the Pol32 RBM on Rfa1 (please compare **Fig. 2A** and **Fig. 3D**). These data show that high-affinity RPA interaction by BIR agonist Pol32 and BIR antagonist Mph1 enhance or diminish BIR efficiency, respectively, strongly suggesting that interactions between the BIR machinery and the Rfa1 basic-hydrophobic groove are utilized to orchestrate D-loop DNA synthesis and disassembly at RPA-bound BIR intermediates.

These molecular insights are not diminished by the observation that disruption of the Pol32 RBM alone does not result in an overt BIR efficiency defect in HO-based BIR genetic assays. However, we realize that a detailed discussion on this point was lacking and, as also requested by **Referee 2**, we have now included a mechanistic model for the different functional contributions of the Pol32 PIP-box and RBM to BIR in the **Discussion** section on **page 15, line 406** to **page 16, line 424**:

“Disruption of the Pol32 RBM is associated with a slight delay in BIR kinetics, but, unlike removal of the PCNA-binding PIP-box, does not translate into an overt BIR efficiency defect in our reporter strains unless the PIP-box is also removed. PCNA encircles the DNA and acts as a processivity factor for Pol δ . Our structural models of the Pol δ -PCNA-DNA complex indicate that the Pol32 PIP-box binds PCNA, thus providing a physical connection between the processivity clamp and Pol δ at the primer junction. This is consistent with biochemical results³³ demonstrating that the Pol32 PIP-box enhances DNA synthesis by Pol δ -PCNA in vitro. The Pol32 RBM provides physical interactions outside the core BIR replisome, with RPA-bound ssDNA that exists in the immediate vicinity of the D-loop primer-junction. We envisage that such interactions affect Pol δ processivity less directly by promoting the recruitment of Pol δ to D-loops and/or aiding its retention if Pol δ disengages prematurely from PCNA. This model is consistent with the ability of the Pol32 RBM to compensate to some extent for the loss of the PIP-box, when BIR efficiency becomes profoundly dependent on the RBM and the Thr256/257 phosphorylation sites needed to enhance the interaction strength with RPA. While the functional contribution of the RBM to BIR efficiency is largely masked in HO-based experimental systems when Pol32-PCNA interactions are intact, it is revealed by enhanced BIR efficiency in the presence of Thr256/257 phospho-mimetic mutations, and we anticipate a particular relevance of the RBM for optimal Pol δ recruitment/recruitment when BIR is in competition with alternative repair pathways at naturally occurring DSBs.”

2. The phosphomimetic Pol32-aspartate mutant exhibits increased BIR efficiency compared with wild-type cells (Fig. 3F). However, the corresponding alanine substitution (phospho-null) mutant behaves similarly to wild type (Fig. 3F). These results indicate that phosphorylation at these sites is dispensable for BIR efficiency. It therefore remains possible that the aspartate substitutions enhance Pol32 function through effects unrelated to phosphorylation, rather than faithfully mimicking a phosphorylated state.

Authors' response: In response to **Specific Point 6** by **Referee 1** in our first revision of the MS, we have demonstrated that the phospho-mimetic aspartate substitution at Pol32 Thr256 and Thr257 enhance the affinity for the Rif1 NTD in line with our data

showing that phosphorylated Pol32 RBM peptides bind Rfa1 with much greater avidity than the non-phosphorylated form. Thus, the threonine to aspartate substitutions significantly increased the affinity of the Pol32 RBM for Rfa1 NTD, improving the K_D from 22.9 μM to 2.93 μM (please see **Supplementary Fig. S6C**). While it is always difficult to entirely rule out additional effects of any given mutation, our data are fully consistent with our interpretation that Pol32 T256D T257D enhances BIR efficiency (**Fig. 3F**) through increased affinity to RPA.

Other points:

1. The *in vitro* kinase assays presented in Fig. 3 do not identify the specific kinase(s) responsible for phosphorylation of these residues. The data are also consistent with the possibility that RBM phosphorylation is mediated by a single, as-yet-unidentified kinase that was not tested in these assays. Therefore, the current data do not necessarily provide evidence that multiple kinases phosphorylate the RBM domain of Pol32.

Authors' response: Our data unequivocally demonstrate that the Pol32 RBM can be phosphorylated by multiple kinases as shown by kinase reactions with radioactive ATP (**Supplementary Fig. S4**) and mass spectrometry (**Fig. 3A, B**). Phosphorylation of the Pol32 RBM by PLK and DDK *in vitro* matches published data showing a dependence of Pol32 RBM phosphorylation on PLK and DDK which occurs *in vivo* (refs. 42 and 44 in our MS), whilst we found no evidence for direct phosphorylation by CDK (ref. 40). DDK, which showed the most robust activity towards the Pol32 RBM in our assays is particularly interesting as Cdc7 has been shown to promote BIR *in vivo* (ref. 46 in our MS). As we point out in the MS text, it is entirely possible that other kinases can phosphorylate the Pol32 RBM; indeed, we show that CKII (like Cdc7 an acidophilic kinase) can phosphorylate the Pol32 RBM; yet *in vivo*, only CDK, PLK, and DDK have been experimentally implicated thus far (ref. 40-44). We acknowledge the possibility of further complexities in Pol32 RBM phosphorylation in the MS text on **page 17, line 453ff**: “However, the details of Pol32 phospho-status regulation, which may involve additional kinases as well as phosphatases, remain to be elucidated in future studies”.

2. The Pol32 RBM mutation significantly reduces the Pol31–RPA interaction (Fig. 2B). The authors could address whether the phosphomimetic pol32-aspartate mutation enhances the RPA–Pol31 interaction, or the pol32-alanine mutation decreases this interaction.

Authors' response: We address enhanced Pol32-RPA interactions upon phosphorylation of the Pol32 RBM directly by quantitative biophysical FP experiments using purified components, showing that phosphorylation at Pol32 residues Thr256 and Thr257 increases the affinity for Rfa1 ~160-fold as compared to the non-phosphorylated Pol32 RBM. These data are robust and conclusive and, in our view, would not be strengthened by further IP experiments using Pol32 with surrogate phospho-mimetic amino acid changes in the Pol32 RBM.

3. Structural models suggest that the RBM is positioned in close proximity to PCNA and Pol31 when Pol δ is bound to the PCNA–DNA complex (Fig. 5A). However, Pol3 and

Pol31 (shaded) or ssDNA-bound RPA are not fully incorporated into the current structural model, it remains uncertain whether the RBM of Pol32 is spatially accessible for interaction with RPA once Pol δ is fully engaged with PCNA on DNA (Fig. 5D). Incorporating improved or alternative structural models, along with an expanded discussion, would strengthen the proposed mechanism by which Pol32–RPA interactions contribute to BIR efficiency.

Authors' response: In revision 1 of our MS (in response to **Specific Point 1** of **Referee 1**), we have greatly extended our structural analyses of the Pol δ -PCNA-DNA complex (**Fig. 5A, B** and **Supplementary Fig. S9A**). This modelling comprises Pol δ in its entirety (*i.e.*, including Pol3 and Pol31, as stated in the MS on **page 13, line 344f.**) bound with PCNA on DNA. The results consistently show that the Pol32 RBM is accessible for the Rfa1 NTD when Pol δ is engaged with PCNA on DNA (**Fig. 5A** and **Supplementary Fig. S9A**); to highlight this key point, which is the most relevant to the current MS, and aid the reader in their interpretation of a rather complex three-dimensional object, Pol3 and Pol31 are shown as 'shaded outlines' in **Fig. 5A** (this is stated in the accompanying figure legend on **page 34, line 1000** of the MS: "*For clarity, molecular cartoons for Pol31 and Pol3 are omitted, but with their relative positions represented by shaded polygons*"). We therefore believe that the aspect of the Pol32-RBM being capable of interacting with the Rfa1-NTD, whilst Pol δ is fully engaged with PCNA on DNA – through simultaneous engagement of the Pol3 CysA/PIP and Pol32-PIP motifs – is comprehensively addressed within the MS text. Whilst we share the desire of **Referee 1** for even more complex modelling including RPA on ssDNA-containing recombination intermediates, it is our opinion that AlphaFold, in its current iteration, cannot deliver a single, globally correct architecture for the Pol δ -PCNA-DNA complex engaged with RPA on a D-loop structure. This is in part due to the long length and structural complexity of a D-loop substrate, and also the lack of sequence-specificity for any of the modelled components that could serve as 'molecular anchors' to provide a defined orientation (a point that is especially true for the RPA trimer that can bind to ssDNA in multiple registers and modes). However, as pointed out in our MS, considering the conformational flexibility afforded by the long linker sequence between the Rfa1 NTD and subsequent RPA subdomains, and the flexibility of the disordered Pol32 C-terminal domain, it is highly plausible that Pol32 within the Pol δ -PCNA-DNA complex could interact with RPA bound to a displaced ssDNA strand at a D-loop, as illustrated in our working models shown in **Fig. 5C** and **D**.

We have followed the recommendation to expand the **Discussion** section in order to address the mechanism by which we envisage Pol32-RPA interactions contribute to BIR efficiency more directly as detailed in our response to **Major point 1** above.

Technical points:

There are no appropriate negative controls in Fig. 2B and supporting raw data. In particular, the POL32 strain should contain TAP-tagged POL31 in addition to POL32-FLAG. In addition, there is no validation shown for the anti-Rfa1 antibody. At least, IP–Western and input (extract) Western samples should be run on the same gel and probed with the anti-Rfa1 antibody to demonstrate specificity and reliability. RPA and Pol δ are localized to the lagging strand during DNA replication. Therefore, it

remains possible that RPA and Pol δ are co-immunoprecipitated indirectly through ongoing lagging-strand DNA synthesis. To confirm a direct interaction between Rfa1 and Pol32, cell extracts used for co-IP experiments should be prepared from G1- or G2/M-arrested cells, where DNA replication is absent.

Authors' response:

- (1) The requested control strain harbouring Pol31-TAP in addition to Pol32-FLAG is the experimental strain shown in the respective middle lanes of the WBs for cell extracts and TAP-IP in **Fig. 2B**. This may not have been entirely clear from the labelling of the figure, and we have re-labelled the WBs shown in **Fig. 2B** in the revised version of the MS to improve clarity.
- (2) The **Source Data file** has been updated to include the raw data for the immunoblot quantifications for **Fig. 2B** in the revised version of the MS.
- (3) The anti-Rfa1 antibody that we are using, Agrisera AS21 4551, is identical to Abcam ab221198. This antibody has been validated by Abcam and is widely used in the literature (for a recent publication by the Diffley, Aguilera, Gómez-González labs using the antibody, please see Núñez-Martín I et al., *Nucleic Acids Res* 53:gkaf707 (2025) doi: 10.1093/nar/gkaf707). To be clear that the anti-Rfa1 antibody is a tried and tested reagent, we have added the Abcam ab221198 specification to the **Reporting summary** for the revised version of the MS.
- (4) As described in the **Methods** section, all IP experiments are conducted in the presence of Pierce Universal Nuclease, which completely digests any DNA and RNA in the extracts, eliminating the possibility of DNA-mediated Pol δ -RPA interactions in our IPs. Furthermore, we show that the Pol32 RBM binds Rfa1 in a direct protein-protein interaction in an *in vitro* system with purified components (please see **Fig. 2C** and **Fig. 3C**), which is in line with the severe reduction of Pol δ -RPA co-immunoprecipitation to background levels when the Pol32 RBM is removed from the Pol δ (**Fig. 2B**), a result that would not be expected if DNA-mediated interactions were involved.

Reviewer #2 (Remarks to the Author):

The revised manuscript is improved, but a few points still need to be addressed. (Page 7, lines 181–184) The manuscript states that Pol32 RBM* shows reduced BIR kinetics at 8 h (Fig. 2E), but the data do not clearly support this description. The difference does not appear significant, as the error bars overlap at the 6 h and/or 8 h time points.

More broadly, Pol32 RBM* alone does not show even a weak BIR phenotype in cells. Only when combined with elimination of the Pol32 PIP box does the RBM mutation appear to have an effect. Because the phenotypes supporting the importance of the Pol32-RPA interaction are weak, any statement in the abstract implying a major regulatory function for Pol32-RPA should be substantially toned down. The Discussion could also include some speculation as to why Pol32 RBM* phenotypes are observed only in mutants that also eliminate the PIP box.

Authors' response: We thank **Referee 2** for their insightful comments. While we observe a slight delay in early BIR kinetics upon mutation of the Pol32 RBM (**Fig. 2E**), we agree that the contribution of the Pol32 RBM to BIR efficiency is only overtly apparent upon additional mutation of the Pol32 PIP-box. In contrast, phospho-mimetic Pol32 RBM mutations enhance BIR efficiency both in the presence or absence of the Pol32 PIP-box (**Fig. 3F and G**). We have therefore followed the recommendation of **Referee 2** to tone down the regulatory implications of Pol32 RBM phosphorylation, instead emphasising the phospho-dependent enhancement of RPA interaction and BIR efficiency observed for full-length Pol32. Accordingly, we have changed the title of the MS from “*Break-induced replication is regulated by a phospho-activated RPA-binding module in Pol32*” to “*Break-induced replication is enhanced by a phospho-activated RPA-binding module in Pol32*”. In the **Abstract**, we have changed the sentence reading “*These results suggest that Pol32 functions as a rheostat regulating BIR efficiency by fine-tuning the affinity of Pol δ for BIR intermediates that are bound by RPA*” to “*These results suggest that Pol32 functions as a rheostat where phosphorylation enhances the affinity of Pol δ for BIR intermediates bound by RPA, thereby boosting BIR efficiency*” in the revised version of the MS.

Secondly, we now highlight the distinction between physical interactions provided by the Pol32 PIP-box, which strengthen the association of Pol δ and its processivity clamp PCNA, and those provided by the Pol32 RBM with RPA-bound ssDNA around the D-loop structure. The former are of immediate relevance to processive DNA synthesis, the latter promote processivity only indirectly by supporting the assembly and retention of the BIR replisome at D-loops, which we propose provides a plausible mechanistic explanation for the apparent dominant role of the Pol32 PIP-box in facilitating BIR outcomes in HO-based BIR reporter assays. We apologize for not being more explicit on this point in the previous version of the MS and have added the following paragraph to the **Discussion** section to lay out our interpretation of the distinct functional roles of the Pol32 RBM and PIP-box (**page 15, line 406 to page 16, line 424**):

“Disruption of the Pol32 RBM is associated with a slight delay in BIR kinetics, but, unlike removal of the PCNA-binding PIP-box, does not translate into an overt BIR efficiency defect in our reporter strains unless the PIP-box is also removed. PCNA encircles the DNA and acts as a processivity factor for Pol δ . Our structural models of the Pol δ -PCNA-DNA complex indicate that the Pol32 PIP-box binds PCNA, thus providing a physical connection between the processivity clamp and Pol δ at the primer junction. This is consistent with biochemical results³³ demonstrating that the Pol32 PIP-box enhances DNA synthesis by Pol δ -PCNA in vitro. The Pol32 RBM provides physical interactions outside the core BIR replisome, with RPA-bound ssDNA that exists in the immediate vicinity of the D-loop primer-junction. We envisage that such interactions affect Pol δ processivity less directly by promoting the recruitment of Pol δ to D-loops and/or aiding its retention if Pol δ disengages prematurely from PCNA. This model is consistent with the ability of the Pol32 RBM to compensate to some extent for the loss of the PIP-box, when BIR efficiency becomes profoundly dependent on the RBM and the Thr256/257 phosphorylation sites needed to enhance the interaction strength with RPA. While the functional contribution of the RBM to BIR efficiency is largely masked in HO-based experimental systems when Pol32-PCNA interactions are intact, it is revealed by

enhanced BIR efficiency in the presence of Thr256/257 phospho-mimetic mutations, and we anticipate a particular relevance of the RBM for optimal Pol δ recruitment/recruitment when BIR is in competition with alternative repair pathways at naturally occurring DSBs.”

Despite these remaining comments, this reviewer finds the revised work very interesting, as it demonstrates that multiple proteins important for BIR interact with RPA, and that these interactions can either stimulate or inhibit the process.

Minor comments:

Line 100: Should this be “~4-fold,” based on the data in the figure?

Authors’ response: We thank **Referee 2** for pointing this out and have changed the MS accordingly.

Question #3 regarding the levels of crossover in the mph1 mutant was not addressed.

Authors’ response: This pertains to a previous question of whether an Mph1 C-terminal mutant is associated with higher crossover frequencies during DSB repair. Crossover frequency measurements require extended or unlimited homology on either side of a DSB, while BIR is measured between limited-homology substrates; thus, the genetic recombination assays deployed in this study do not report on crossover frequency. However, the ability of Mph1 to suppress crossovers (instead promoting synthesis-dependent strand annealing at DSBs) and to restrict BIR outcomes are equally underpinned by Mph1-dependent dissociation of the invading DNA strand from D-loops (Prakash R et al., Genes Dev 23:67-79 (2009) doi: 10.1101/gad.1737809). Our results from revision 1 of the MS showing that disrupting the Mph1 RBM phenocopies *MPH1* loss for elevated BIR efficiency and enhances D-loop stability in a PCR-based BIR assay provide very strong evidence that Mph1-dependent D-loop dissociation relies on Mph1-RPA interactions. While this predicts crossover frequency will be equally affected upon disruption of the Mph1 C-terminal RBM (with less effective D-loop disruption resulting in pathway choice away from synthesis-dependent strand annealing to more crossover outcomes), we feel that an expansion into other DSB repair pathways more peripherally linked to the BIR focus of our MS is not essential to draw the conclusion that the Mph1 RBM promotes the BIR antagonistic, well-established D-loop dissociation function of Mph1.

Reviewer #3 (Remarks to the Author):

I appreciate all the effort the authors put into addressing the comments on the original submission and the resubmitted manuscript did improve by adding more clarity to the conclusions through providing additional data. Unfortunately, adding more experiments haven't changed the overall message. The newly identified RBM in Pol32 has very little if any relevance to the efficiency of BIR in the context of full length Pol32. This is supported by multiple pieces of data on mutating the RBM in different ways - deleting it completely (figure 1), replacing a couple critical residues with Rs (RBM*

mutant, figure 2) or mutating the phosphorylatable threonines into As or Ds (Figure 3). I appreciate that all the in vitro analyses show that the phosphorylation of these threonines improves the Pol32-RPA interaction, yet there is no effect in vivo, unless the Pol32-PCNA interaction is lost. For this reason, I find the title and the second part of the statement in the abstract "Phosphorylation of the Pol32 RBM at Thr256 and Thr257 increases its affinity for Rfa1, while corresponding phospho-mimetic amino-acid substitutions promote BIR efficiency in vivo" misleading as they don't mention the context of broken Pol32-PCNA interaction needed to observe the effect relevant to the Pol32-RPA interaction. One can't deny that the RBM is real and that the reported phosphorylation might be critical for some process involving the Pol32-RPA interaction, but the authors' own experiments prove that BIR is not this process because the Pol32-PCNA interaction is sufficient for the fully efficient BIR.

Authors' response: **Referee 3** is right in pointing out that while the mutation of the Pol32 RBM results in a slight delay in BIR kinetics (**Fig. 2E**), this does not cause an overt BIR efficiency phenotype in our HO-based BIR report systems. However, it is important to consider that the contribution of the Pol32 RBM to BIR efficiency is unmasked upon removal of the Pol32 PIP-box, a condition under which BIR efficiency is dependent on the Pol32 RBM, which shows that Pol32-RPA interactions facilitate the completion of BIR along extended chromosomal regions (30 kb to 100kb) (**Fig. 1E, Fig. 2D, Fig. 2E, and Supplementary Fig. S1F**). Similarly, mutation of the Thr256/257 phosphorylation sites in Pol32 RBM largely abolishes BIR outcomes in *pol32_1-133_T256A T257A* cells (**Fig. 3G**). This is compatible with different functional roles which we envisage for the Pol32 RBM vs. the PIP-box modules in promoting BIR: the Pol32 PIP-box directly connects Pol δ to its processivity clamp PCNA, thereby supporting continuous BIR DNA synthesis; the Pol32 RBM links Pol δ to RPA-bound ssDNA within D-loops, thus only indirectly supporting processivity by promoting BIR replisome recruitment and/or retention at D-loops. This mechanistic difference provides a conceptual model to explain why the presence of the Pol32 PIP-box largely masks the contribution of the newly identified Pol32 RBM to promoting BIR outcomes in HO-based experimental systems. We realize that we were not clear enough on this point and now elaborate on the potential cellular function of the Pol32 RBM for BIR in an extended **Discussion** section in the revised version of the MS (**page 15, line 406 to page 16, line 424**):

“Disruption of the Pol32 RBM is associated with a slight delay in BIR kinetics, but, unlike removal of the PCNA-binding PIP-box, does not translate into an overt BIR efficiency defect in our reporter strains unless the PIP-box is also removed. PCNA encircles the DNA and acts as a processivity factor for Pol δ . Our structural models of the Pol δ -PCNA-DNA complex indicate that the Pol32 PIP-box binds PCNA, thus providing a physical connection between the processivity clamp and Pol δ at the primer junction. This is consistent with biochemical results³³ demonstrating that the Pol32 PIP-box enhances DNA synthesis by Pol δ -PCNA in vitro. The Pol32 RBM provides physical interactions outside the core BIR replisome, with RPA-bound ssDNA that exists in the immediate vicinity of the D-loop primer-junction. We envisage that such interactions affect Pol δ processivity less directly by promoting the recruitment of Pol δ to D-loops and/or aiding its retention if Pol δ disengages prematurely from PCNA. This model is consistent with the ability of the Pol32 RBM to compensate to some extent for the loss

of the PIP-box, when BIR efficiency becomes profoundly dependent on the RBM and the Thr256/257 phosphorylation sites needed to enhance the interaction strength with RPA. While the functional contribution of the RBM to BIR efficiency is largely masked in HO-based experimental systems when Pol32-PCNA interactions are intact, it is revealed by enhanced BIR efficiency in the presence of Thr256/257 phospho-mimetic mutations, and we anticipate a particular relevance of the RBM for optimal Pol δ recruitment/recruitment when BIR is in competition with alternative repair pathways at naturally occurring DSBs.”

With respect to the statement (**Abstract**) “Phosphorylation of the Pol32 RBM at Thr256 and Thr257 increases its affinity for Rfa1, while corresponding phospho-mimetic amino-acid substitutions promote BIR efficiency *in vivo*.”, this is fully supported by the data presented in our MS (please see **Fig. 3F** showing that full-length Pol32 T256D T257D-expressing cells exhibit BIR efficiency levels that are significantly above wild-type levels). Reflecting the phosphorylation-dependent enhancement of Pol32 RBM-RPA interaction and the boost in BIR efficiency observed for Pol32 T256D T257D (in the presence of the PIP-box), and following a recommendation by **Referee 2**, we have changed the title of the MS from “*Break-induced replication is regulated by a phospho-activated RPA-binding module in Pol32*” to “*Break-induced replication is enhanced by a phospho-activated RPA-binding module in Pol32*”.

Investigating the RBM in Pif1 deeper also ruled out the importance of its RBM phosphorylation for BIR *in vivo*, even though the phosphorylation of S70 and S72 has been previously reported and as the authors discovered, it does increase the RPA binding *in vitro*.

Authors’ response: Please note that phosphorylation of Pif1 Ser70 and Ser72 results in an approximate 2-fold increase of an already high affinity of the Pif1 RBM for RPA (**Supplementary Fig. S8**). This is in stark contrast to an approximate 160-fold increase in affinity for the Pol32 RBM (which binds Rfa1 with a moderate affinity in the non-phosphorylated form) upon phosphorylation at Thr256 and Thr265. As we explain in the MS (please refer to section “*Pol32 pRBM, Pif1, and Mph1 similarly target Rfa1*” on **page 11**), this can be rationalized by the presence of a negative charge within Pif1 (Asp71) equivalent to Thr256 in Pol32 that allows Pif1 to engage with Rfa1 in the ‘high-affinity’ mode with Pif1 Trp75 projecting into the Rfa1 hydrophobic ‘side-pocket’ independently of phosphorylation. In contrast, Pol32 RBM phosphorylation is required for ‘side-pocket’ interactions of Pol32 Leu262. Consistently, cells expressing (full-length) Pol32 T256D T257D *in vivo* exhibit a boost in BIR efficiency (**Fig. 3F**) while Pif1 S70D S72D expression is no different from Pif1 wild-type expression in terms of BIR efficiency. This provides strong evidence that Pol32 RBM phosphorylation does promote BIR, and we argue that this is not thrown into question, but rather corroborated by the finding that Pif1 RBM phosphorylation is not required for high-affinity RPA binding nor BIR efficiency.

The newly identified RBM in Mph1 also looks real based on the modelling and *in vivo* analysis, even though the authors didn't analyse the relevance of the suggested phosphorylation on S989 and S991 by mutagenesis to address its possible role in either RPA binding or in BIR directly (*in vivo*).

Authors' response: As explained in our response to the preceding point, our data show that RBMs exist in two flavours and either support phosphorylation-dependent or constitutive high-affinity Rfa1 binding. Like Pif1, Mph1 falls into the latter category (Mph1 Asp990 has equivalency with Pif1 Asp71 allowing high-affinity binding in the non-phosphorylated form, please refer to **Fig. 4A, D** in our MS), for which our analysis of Pif1 serves as an exemplar case.

The model presented in Figure 4F is interesting but not justified by the experiments presented in the paper. The phosphorylation of Pol32 RBM is supposed to play a critical molecular role in it regulating the BIR progression, yet the loss of this RBM has no effect on BIR efficiency in vivo. The model in Figure 5D is more appropriate for the manuscript and justified by the presented discoveries.

Authors' response: As detailed in our response to **Referee 3's** first point above, we now provide an extended discussion to explain why the removal of the PIP-box is required to unmask the contribution of the Pol32 RBM to BIR efficiency. With regard to the model **Fig. 4F**, we disagree that this is not supported by the data. We show in **Fig. 3F** that phospho-mimicking mutations T256D T257D within full-length Pol32 are associated with a significant increase in BIR efficiency over wild-type Pol32, which is fully consistent with the depiction in **Fig. 4F**. Upon request, we can relegate **Fig. 4** panel **F** to the supplementary information, but we feel this would be a loss as this illustration serves as a visual summary of results for a broader audience.